# GENERALIZED SOBOLEV IPM FOR GRAPH-BASED MEASURES

## ABSTRACT

We study the Sobolev IPM problem for measures supported on a graph metric space, where critic function is constrained to lie within the unit ball defined by Sobolev norm. While Le et al. (2025) achieved scalable computation by relating Sobolev norm to weighted $L^p$-norm, the resulting framework remains intrinsically bound to $L^p$ geometric structure, limiting its ability to incorporate alternative structural priors beyond the $L^p$ geometry paradigm. To overcome this limitation, we propose to generalize Sobolev IPM through the lens of *Orlicz geometric structure*, which employs convex functions to capture nuanced geometric relationships, building upon recent advances in optimal transport theory—particularly Orlicz-Wasserstein (OW) and generalized Sobolev transport—that have proven instrumental in advancing machine learning methodologies. This generalization encompasses classical Sobolev IPM as a special case while accommodating diverse geometric priors beyond traditional $L^p$ structure. It however brings up significant computational hurdles that compound those already inherent in Sobolev IPM. To address these challenges, we establish a novel theoretical connection between Orlicz-Sobolev norm and Musielak norm which facilitates a novel regularization for the generalized Sobolev IPM (GSI). By further exploiting the underlying graph structure, we show that GSI with Musielak regularization (GSI-M) reduces to a simple *univariate optimization* problem, achieving remarkably computational efficiency. Empirically, GSI-M is several-order faster than the popular OW in computation, and demonstrates its practical advantages in comparing probability measures on a given graph for document classification and several tasks in topological data analysis.

## 1 INTRODUCTION

Probability measures serve as canonical mathematical representations for diverse objects across various research domains, e.g., documents in natural language processing (Kusner et al., 2015; Yurochkin et al., 2019), persistence diagrams in topological data analysis (Edelsbrunner & Harer, 2008; Le et al., 2025), point clouds in computer vision and graphics (Hua et al., 2018; Wang et al., 2019). To compare such measures, integral probability metrics (IPM) offer a versatile and principled class of metric functions (Müller, 1997). Conceptually, IPM operate by determining an optimal critic function that achieves maximal discrimination between two probability measures. This mathematical elegance and versatility has facilitated the widespread adoption of IPM throughout statistics and machine learning (Sriperumbudur et al., 2009; Gretton et al., 2012; Peyré & Cuturi, 2019; Liang, 2019; Uppal et al., 2019; 2020; Nadjahi et al., 2020; Kolouri et al., 2020).

In this work, we study the Sobolev IPM problem for measures supported on a graph metric space, where critic function is constrained within the unit ball induced by Sobolev norm (Adams & Fournier, 2003).[1] Sobolev IPM has proven fundamental to numerous theoretical analyses, including convergence rates in density estimation and approximation theory for deep architectures (Liang, 2017; 2021; Singh et al., 2018). Although Le et al. (2025) recently pioneered computationally tractable algorithmic approach by relating Sobolev norm to weighted $L^p$-norm, the resulting framework remains intrinsically bound to $L^p$ geometric structure, thereby limiting its ability to incorporate

---

[1]One should distinguish Sobolev IPM problem from Sobolev transport (Le et al., 2022) and generalized Sobolev transport (Le et al., 2025) problems which only constraint on gradient of critic function. See §5 for a further discussion.

alternative structural priors. To overcome this limitation, we propose to generalize Sobolev IPM (GSI) through the lens of *Orlicz geometric structure*, which employs convex functions to capture nuanced geometric relationships, building upon seminal developments in optimal transport (OT) theory—notably Orlicz-Wasserstein (OW) (Sturm, 2011; Kell, 2017; Guha et al., 2023; Altschuler & Chewi, 2023) and generalized Sobolev transport (GST) (Le et al., 2024)—that have demonstrated remarkable effectiveness in advancing machine learning methodologies. More specifically, Altschuler & Chewi (2023) leverage OW to facilitate the development of differential-privacy-inspired methodologies that address long-standing convergence challenges in hypocoercive differential equations. Similarly, Guha et al. (2023) demonstrate that OW substantially enhances Bayesian contraction rates, effectively circumventing limitations inherent to traditional OT with Euclidean ground cost. Although the computational burden of OW is substantial, GST offers a scalable variant suitable for practical use. Moreover, Orlicz geometric structure has found broad applicability across diverse machine learning paradigms (Andoni et al., 2018; Song et al., 2019; Deng et al., 2022; Chamakh et al., 2020; Lorenz & Mahler, 2022). For comprehensive studies of Orlicz functions, see (Adams & Fournier, 2003; Rao & Ren, 1991).

Analogous to the computational challenges inherent in Sobolev IPM, the generalized Sobolev IPM (GSI) poses significant computational obstacles. To overcome these limitations, we establish a novel connection between Orlicz-Sobolev norm and Musielak norm which in turn motivates a *novel regularization* scheme for GSI. Exploiting the underlying graph structure, we further show that GSI with Musielak regularization (GSI-M) reduces to a simple *univariate optimization* problem, yielding substantial computational efficiency and enabling practical deployment at scale. Therefore, our approach helps to mitigate the computational challenges of GSI, and paves the way for its practical applications, even at scale.

**Contribution.** Our contributions are three-fold as follows:

- We leverage a certain class of convex functions corresponding to Orlicz geometric structure to generalize Sobolev IPM beyond $L^p$ geometric structure for graph-based measures. Additionally, we propose a *novel regularization* for the resulting GSI metric that yields an efficient computation by simply solving a univariate optimization problem.

- GSI-M utilizes the Orlicz geometric structure in the same sense as OW/GST for OT problem. We prove that GSI-M is a metric and show its *equivalence* to the original GSI. Moreover, we establish its connections to original/regularized Sobolev IPM, and other transport distances.

- We empirically illustrate that GSI-M is more computationally efficient than OW, and comparable to GST, a scalable variant of OW. We also provide initial evidences on the advantages of GST for document classification and for several tasks in topological data analysis (TDA).

**Organization.** In §2, we review related backgrounds and notations. We describe the generalized Sobolev IPM (GSI) and its novel Musielak regularization in §3. In §4, we prove the metric property for the generalized Sobolev IPM with Musielak regularization (GSI-M) and establish its connection to the original GSI, and other transport distances for graph-based measures. We then discuss related works in §5. In §6, we empirically show the computational efficiency of GSI-M and provide initial evidence of its benefits in document classification and TDA, following by concluding remarks in §7. Proofs for theoretical results and additional materials are deferred to the Appendices.

## 2 Preliminaries

In this section, we introduce notations, and briefly review graph, Orlicz functions, and Sobolev IPM.

**Graph.** We follow the setting for graph-based measures in (Le et al., 2025). We consider a connected, undirected, and physical[2] graph $\mathbb{G}$ with set of nodes and edges $V, E$ respectively, and positive edge lengths $\{w_e\}_{e \in E}$. For continuous graph setting, we regard $\mathbb{G}$ as the set of all nodes in $V$ and all points forming the edges in $E$. Additionally, let $[x, z]$ be the shortest path connecting $x$ and $z$ in $\mathbb{G}$, and equip $\mathbb{G}$ with graph metric $d_{\mathbb{G}}(x, z)$, i.e., the length of $[x, z]$. We assume that there exists a root node $z_0 \in V$ such that for any $x \in \mathbb{G}$, then $[z_0, x]$ is unique, i.e., the uniqueness property of the

---

[2]In the sense that $V$ is a subset of $\mathbb{R}^n$, and each edge $e \in E$ is the standard line segment connecting the two corresponding vertices of edge $e$ in $\mathbb{R}^n$.

shortest paths.[3] Denote $\mathcal{P}(\mathbb{G})$ (resp. $\mathcal{P}(\mathbb{G} \times \mathbb{G})$) as the set of all nonnegative Borel measures on $\mathbb{G}$ (resp. $\mathbb{G} \times \mathbb{G}$) with a finite mass. For $x \in \mathbb{G}$, edge $e \in E$, define the sets $\Lambda(x)$ and $\gamma_e$ as follows:

$$\Lambda(x) := \big\{ y \in \mathbb{G} : x \in [z_0, y] \big\}, \qquad \gamma_e := \big\{ y \in \mathbb{G} : e \subset [z_0, y] \big\}. \tag{1}$$

By a continuous function $f$ on $\mathbb{G}$, we mean that $f : \mathbb{G} \to \mathbb{R}$ is continuous w.r.t. the topology on $\mathbb{G}$ induced by the Euclidean distance. Similar notation is used for continuous functions on $\mathbb{G} \times \mathbb{G}$.

**A family of convex functions.** We consider the set of $N$-functions (Adams & Fournier, 2003, §8.2), which are special convex functions on $\mathbb{R}_+$. Henceforth, a strictly increasing and convex function $\Phi : [0, \infty) \to [0, \infty)$ is called an $N$-function if $\lim_{t \to 0} \frac{\Phi(t)}{t} = 0$ and $\lim_{t \to +\infty} \frac{\Phi(t)}{t} = +\infty$.

**Orlicz functional space.** For $N$-function $\Phi$, and a nonnegative Borel measure $\lambda$ on $\mathbb{G}$, let $L_\Phi(\mathbb{G}, \lambda)$ be the linear hull of the collection of all Borel measurable functions $f : \mathbb{G} \to \mathbb{R}$ satisfying $\int_\mathbb{G} \Phi(|f(x)|)\lambda(\mathrm{d}x) < \infty$. Then, $L_\Phi(\mathbb{G}, \lambda)$ is a normed space with the Luxemburg norm, defined as

$$\|f\|_{L_\Phi} := \inf \left\{ t > 0 \mid \int_\mathbb{G} \Phi\left( \frac{|f(x)|}{t} \right) \lambda(\mathrm{d}x) \leq 1 \right\}. \tag{2}$$

For positive weight function $\hat{w}$ on $\mathbb{G}$, consider the weighted $L_\Phi^{\hat{w}}(\mathbb{G}, \lambda)$ as the linear hull of the set of all Borel measurable functions $f : \mathbb{G} \to \mathbb{R}$ satisfying $\int_\mathbb{G} \hat{w}(x)\Phi(|f(x)|)\lambda(\mathrm{d}x) < \infty$. Then, $L_\Phi^{\hat{w}}(\mathbb{G}, \lambda)$ is a normed space[4] with Musielak norm (Musielak, 2006, §10.2)[5] being defined by

$$\|f\|_{L_\Phi^{\hat{w}}} := \inf \left\{ t > 0 \mid \int_\mathbb{G} \hat{w}(x)\, \Phi\left( \frac{|f(x)|}{t} \right) \lambda(\mathrm{d}x) \leq 1 \right\}. \tag{3}$$

**Sobolev IPM.** For an exponent $1 \leq p \leq \infty$ and its conjugate $p'$,[6] let $W_0^{1,p}(\mathbb{G}, \lambda)$ be the subspace consisting of all functions $f$ in the graph-based Sobolev space $W^{1,p}(\mathbb{G}, \lambda)$ (Le et al., 2022, Definition 3.1) satisfying $f(z_0) = 0$, then Sobolev IPM between measures $\mu, \nu \in \mathcal{P}(\mathbb{G})$ is defined as

$$\mathcal{S}_p(\mu, \nu) := \sup_{f \in W_0^{1,p}(\mathbb{G}, \lambda), \|f\|_{W^{1,p}} \leq 1} \left| \int_\mathbb{G} f(x)\mu(\mathrm{d}x) - \int_\mathbb{G} f(y)\nu(\mathrm{d}y) \right|, \tag{4}$$

where $\|f\|_{W^{1,p}}$ is the Sobolev norm (Adams & Fournier, 2003, §3.1), defined as

$$\|f\|_{W^{1,p}} := \left( \|f\|_{L^p}^p + \|f'\|_{L^p}^p \right)^{\frac{1}{p}}. \tag{5}$$

From the definition of Sobolev IPM (in Equation (4)), it is essentially coupled with the $L^p$ geometric structure. Consequently, unlike optimal transport (OT), where one can adapt to various prior geometric structures by simply modifying the underlying ground cost, it is nontrivial to use Sobolev IPM with other prior geometric structures. In the next section, we will leverage the set of convex $N$-functions to generalize Sobolev IPM beyond the coupled $L^p$ geometric structure.[7]

## 3 GENERALIZED SOBOLEV IPM (GSI)

Sobolev IPM provides a powerful yet rigid framework, essentially coupled with the $L^p$ geometric structure within its definition (Equation (4)). As a result, it is nontrivial to utilize Sobolev IPM with other prior structures, which is in stark contrast to the flexibility of optimal transport (OT) for its adaptivity to diverse prior geometric structures by simply modifying the underlying cost function. In this section, we leverage convex $N$-functions to generalize Sobolev IPM. We first introduce the graph-based Orlicz-Sobolev space (Le et al., 2024) and its Orlicz-Sobolev norm (Rao & Ren, 1991, §9.3), (Adams & Fournier, 2003, §3.1, §8.30). Based on these components, we then describe the definition of the generalized Sobolev IPM (GSI) for graph-based measures.

---

[3]There may exist multiple paths connecting $z_0$ and $x$ in $\mathbb{G}$, but the shortest path $[z_0, x]$ is unique.

[4]The weighted $L_\Phi^{\hat{w}}(\mathbb{G}, \lambda)$ is a specific instance of the Musielak-Orlicz space, where the generalized $N$-function $\bar{\Phi}(x, t) = \hat{w}(x)\Phi(t)$ for all $t > 0$ and $x \in \mathbb{G}$.

[5]See also in (Harjulehto & Hästö, 2019, Definition 3.2.1).

[6]$p' \in [1, \infty]$ satisfying $\frac{1}{p} + \frac{1}{p'} = 1$. If $p = 1$, then $p' = \infty$.

[7]To ease the readers, we further give a brief review for related works and notions in the literature in §D.

**Definition 3.1** (Graph-based Orlicz-Sobolev space (Le et al., 2024)). Let $\Phi$ be an $N$-function and $\lambda$ be a nonnegative Borel measure on graph $\mathbb{G}$. A continuous function $f : \mathbb{G} \to \mathbb{R}$ is said to belong to the graph-based Orlicz-Sobolev space $WL_\Phi^1(\mathbb{G}, \lambda)$ if there exists a function $h \in L_\Phi(\mathbb{G}, \lambda)$ satisfying

$$f(x) - f(z_0) = \int_{[z_0, x]} h(y)\lambda(\mathrm{d}y), \quad \forall x \in \mathbb{G}. \tag{6}$$

Such function $h$ is unique in $L_\Phi(\mathbb{G}, \lambda)$ and is called the generalized graph derivative of $f$ w.r.t. the measure $\lambda$. Henceforth, this generalized graph derivative of $f$ is denoted $f'$.

**Orlicz-Sobolev norm.** $WL_\Phi^1(\mathbb{G}, \lambda)$ is a normed space with the Orlicz-Sobolev norm (Rao & Ren, 1991; Adams & Fournier, 2003), defined as

$$\|f\|_{WL_\Phi^1} := \|f\|_{L_\Phi} + \|f'\|_{L_\Phi}. \tag{7}$$

Additionally, let $WL_{\Phi,0}^1(\mathbb{G}, \lambda)$ be the subspace consisting of all functions $f$ in $WL_\Phi^1(\mathbb{G}, \lambda)$ satisfying $f(z_0) = 0$. Moreover, notice that for $N$-function $\Phi(t) = t^p$ for $1 < p < \infty$, following (Le et al., 2024, Proposition 4.3), we have $WL_\Phi^1(\mathbb{G}, \lambda) = W^{1,p}(\mathbb{G}, \lambda)$ where $W^{1,p}$ is the graph-based Sobolev space,[8] and the $L_\Phi$-norm is equal to the $L^p$-norm (Adams & Fournier, 2003).

**Generalized Sobolev IPM (GSI).** Similar to Sobolev IPM, the GSI is an instance of IPM where its critic function belongs to the graph-based Orlicz-Sobolev space, and is constrained within the unit ball of that space. More concretely, given a nonnegative Borel measure $\lambda$ on $\mathbb{G}$, a pair of complementary $N$-functions $\Phi, \Psi$, the GSI between probability measures $\mu, \nu \in \mathcal{P}(\mathbb{G})$ is defined as

$$\mathcal{GS}_\Phi(\mu, \nu) := \sup_{f \in \mathcal{B}_\Psi} \left| \int_\mathbb{G} f(x)\mu(\mathrm{d}x) - \int_\mathbb{G} f(y)\nu(\mathrm{d}y) \right|, \tag{8}$$

where $\mathcal{B}_\Psi := \left\{ f \in WL_{\Psi,0}^1(\mathbb{G}, \lambda), \|f\|_{WL_\Psi^1} \leq 1 \right\}$ is the unit ball defined by Orlicz-Sobolev norm. For $N$-function $\Psi(t) = t^p$ with $1 < p < \infty$, the GSI turns into Sobolev IPM. Furthermore, notice that the quantity inside the absolute signs is unchanged if $f$ is replaced by $f - f(z_0)$. Thus, we can assume without loss of generality that $f(z_0) = 0$. This is the motivation for our introduction of the Orlicz-Sobolev space $WL_{\Psi,0}^1(\mathbb{G}, \lambda)$, which shares the same sense to Sobolev IPM approach (Le et al., 2025), and the Sobolev GAN approach (Mroueh et al., 2018).

Analogous to the computational challenges inherent in Sobolev IPM, the GSI poses significant computational obstacles. We next draw a novel relation between the Orlicz-Sobolev norm and Musielak norm to form a novel regularization for GSI for efficient computation.

**Weight function.** Hereafter, we consider the weight function, defined as

$$\hat{w}(x) := 1 + \frac{\lambda(\Lambda(x))}{\lambda(\mathbb{G})}, \quad \forall x \in \mathbb{G}. \tag{9}$$

**Theorem 3.2** (Equivalence). *For the length measure $\lambda$ on $\mathbb{G}$, and function $f \in WL_{\Phi,0}^1(\mathbb{G}, \lambda)$, then*

$$\frac{1}{2} \|f'\|_{L_\Phi^{\hat{w}}} \leq \|f\|_{WL_\Phi^1} \leq (1 + \lambda(\mathbb{G})) \|f'\|_{L_\Phi^{\hat{w}}}. \tag{10}$$

Theorem 3.2 implies that the Orlicz-Sobolev norm of a critic function $f \in WL_{\Phi,0}^1(\mathbb{G}, \lambda)$ is equivalent to the Musielak norm of its gradient $f'$, where weight function $\hat{w}$ is given explicitly in Equation (9).

*Remark* 3.3. The novel theoretical finding result in Theorem 3.2 plays the key role for the development of our *regularization* scheme approach, i.e., relaxing the constraint of critic function $f$ within the unit ball induced by Orlicz-Sobolev norm by the Musielak norm constraint of its gradient $f'$ with a specified weight function $\hat{w}$ (Equation (9)).

**Discussion.** Le et al. (2025, Theorem 3.2) established an equivalence between the *Sobolev norm* of a critic function and *weighted $L^p$ norm* of its gradient for the weight function $\hat{w}_{\hat{\mathcal{S}}}(x) = 1 + \lambda(\Lambda(x))$ for all $x \in \mathbb{G}$ by exploiting the closed-form expression of $L^p$-norm and weighted $L^p$-norm. It is *unknown* whether such result is still held for general $N$-functions, beyond the $L^p$ geometric structure,

---

[8]See a review in §D.3.

in Orlicz norm (Equation (2)) and Musielak norm (Equation (3)) for their considered weight function $\hat{w}_{\hat{S}}$. In this work, we leverage the *different* weight function $\hat{w}$ (Equation (9)), formed from our theoretical findings,[9] to establish an equivalence between the *Orlicz-Sobolev norm* of a critic function and *Musielak norm* of its gradient. Notably, to our knowledge, the theoretical finding result in Theorem 3.2 is *novel*. It is in fact nontrivial, and clearly beyond existing results in (Le et al., 2025) for Sobolev IPM and the literature. Unlike the $L^p$ geometric structure exploited in (Le et al., 2025), there is *no* closed-form expression for the Orlicz norm (Equation (2)) and Musielak norm (Equation (3)) for general $N$-functions in our considered GSI problem. In fact, it requires several novel auxiliary theoretical finding results in Appendix A.1, beyond exisiting results in (Le et al., 2025), to establish the *novel* equivalence result in Theorem 3.2. It is also worth noting that the Theorem 3.2 holds true for our considered weight function (Equation 9), and such equivalence result may not exist for any given general weight function.

**Generalized Sobolev IPM with Musielak regularization (GSI-M).** Based on the equivalent relation given by Theorem 3.2, we propose to regularize the GSI (Equation (8)) by relaxing the constraint on the critic function $f$ in the graph-based Orlicz-Sobolev space $WL^1_{\Psi,0}$. More precisely, instead of $f$ belonging to the unit ball $\mathcal{B}_\Psi$ of the Orlicz-Sobolev space, we propose to constraint critic $f$ within the unit ball $\mathcal{B}^{\hat{w}}_\Psi$ of the Musielak norm of $f'$ with $N$-function $\Psi$, and weight function $\hat{w}$. Hereafter, $\mathcal{B}^{\hat{w}}_\Psi$ is defined by

$$\mathcal{B}^{\hat{w}}_\Psi := \left\{ f \in WL^1_{\Psi,0}(\mathbb{G}, \lambda), \|f'\|_{L^{\hat{w}}_\Psi} \leq 1 \right\}. \tag{11}$$

We now formally define the generalized Sobolev IPM with Musielak regularization (GSI-M) between two probability distributions on graph $\mathbb{G}$.

**Definition 3.4** (Generalized Sobolev IPM with Musielak regularization). Let $\lambda$ be a nonnegative Borel measure on $\mathbb{G}$ and a pair of complementary $N$-functions $\Phi, \Psi$. Then, for probability measures $\mu, \nu \in \mathcal{P}(\mathbb{G})$, the generalized Sobolev IPM with Musielak regularization is defined as

$$\widehat{\mathcal{GS}}_\Phi(\mu, \nu) := \sup_{f \in \mathcal{B}^{\hat{w}}_\Psi} \left| \int_\mathbb{G} f(x)\mu(\mathrm{d}x) - \int_\mathbb{G} f(y)\nu(\mathrm{d}y) \right|. \tag{12}$$

**Computation.** We next show that one can compute $\widehat{\mathcal{GS}}_\Phi$ by simply solving a univariate optimization problem, paving ways for its practical applications.

**Theorem 3.5** (GSI-M as univariate optimization problem). *The generalized Sobolev IPM with Musielak regularization $\widehat{\mathcal{GS}}_\Phi(\mu, \nu)$ in Definition 3.4 can be computed as follows:*

$$\widehat{\mathcal{GS}}_\Phi(\mu, \nu) = \inf_{k>0} \frac{1}{k} \left( 1 + \int_\mathbb{G} \hat{w}(x) \, \Phi\left( \frac{k}{\hat{w}(x)} \, |\mu(\Lambda(x)) - \nu(\Lambda(x))| \right) \lambda(dx) \right). \tag{13}$$

Notice that in Equation (13), both the subgraph $\Lambda(x)$ (Equation (1)) and the weight function $\hat{w}(x)$ (Equation (9)) depend on input point $x$ under the integral over $\mathbb{G}$. For practical applications, we next derive an explicit formula for the integral over graph $\mathbb{G}$ in Equation (13) when the input probability measures are supported on nodes $V$ of graph $\mathbb{G}$. This gives an efficient method for computing the GSI-M $\widehat{\mathcal{GS}}_\Phi$. Note that to achieve this result, we use the length measure on graph $\mathbb{G}$ (Le et al., 2022) for the nonnegative Borel measure $\lambda$, i.e., we have $\lambda([x, z]) = d_\mathbb{G}(x, z), \forall x, z \in \mathbb{G}$. We summarize the result in the following theorem.

**Theorem 3.6** (Discrete case). *Let $\lambda$ be the length measure on $\mathbb{G}$, and $\Phi$ be an $N$-function. Suppose that $\mu, \nu \in \mathcal{P}(\mathbb{G})$ are supported on nodes $V$ of graph $\mathbb{G}$.[10] Then we have*

$$\widehat{\mathcal{GS}}_\Phi(\mu, \nu) = \inf_{k>0} \frac{1}{k} \left( 1 + \sum_{e \in E} \left[ \int_0^1 \bar{w}_t(e) \, \Phi\left( \frac{k|\bar{h}(e)|}{\bar{w}_t(e)} \right) w_e \mathrm{d}t \right] \right), \tag{14}$$

*where $\bar{h}(e) := \mu(\gamma_e) - \nu(\gamma_e)$, and $\bar{w}_t(e) := \frac{w_e}{\lambda(\mathbb{G})}t + 1 + \frac{\lambda(\gamma_e)}{\lambda(\mathbb{G})}$ for all edge $e \in E$ and $t \in [0, 1]$.*

---

[9]Especially from the novel auxiliary result in Theorem A.3 in Appendix A.1.

[10]An extension for measures supported in graph $\mathbb{G}$ is discussed in Appendix §E.

From Theorem 3.6, by exploiting the graph structure for the integral over graph $\mathbb{G}$ in Equation (13), one only needs to simply solve the univariate optimization problem to compute the GSI-M $\widehat{\mathcal{GS}}_\Phi$.[11] We further note that for each edge $e$, given a specific popular $N$-function, then the integral w.r.t. scalar $t$ has an explicit form for efficient computation, see Appendix §A.3 for details.[12]

**Special case with closed-form expression.** We illustrate that for a specific $N$-function, the GSI-M can yield a closed-form expression for fast computation, especially for the discrete case when input measures are supported on nodes $V$ of graph $\mathbb{G}$.

**Proposition 3.7** (Closed-form discrete case). *Suppose that $\mu, \nu \in \mathcal{P}(\mathbb{G})$ are supported on nodes $V$ of graph $\mathbb{G}$. For $N$-function $\Phi(t) = \frac{(p-1)^{p-1}}{p^p} t^p$ with $1 < p < \infty$, length measure $\lambda$ on $\mathbb{G}$, then*

$$\widehat{\mathcal{GS}}_\Phi(\mu, \nu) = \left( \sum_{e \in E} \beta_e \, |\mu(\gamma_e) - \nu(\gamma_e)|^p \right)^{\frac{1}{p}},$$
(15)

*where for each edge $e \in E$ of graph $\mathbb{G}$, the scalar number $\beta_e$ is given by*

$$\beta_e := \begin{cases} \lambda(\mathbb{G}) \log \left( 1 + \frac{w_e}{\lambda(\mathbb{G}) + \lambda(\gamma_e)} \right) & \text{if } p = 2, \\ \frac{(\lambda(\mathbb{G}) + \lambda(\gamma_e) + w_e)^{2-p} - (\lambda(\mathbb{G}) + \lambda(\gamma_e))^{2-p}}{(2-p)\lambda(\mathbb{G})^{1-p}} & \text{otherwise.} \end{cases}$$
(16)

The proofs for these theoretical results (in §3) are respectively placed in §B.1 – §B.4.

## 4    PROPERTIES OF GSI WITH MUSIELAK REGULARIZATION (GSI-M)

In this section, we derive the metric property for the GSI-M and establish a relationship for the GSI-M with different $N$-functions. Additionally, we draw connections of the GSI-M to the original GSI, the original Sobolev IPM, the scalable regularized Sobolev IPM (Le et al., 2025), GST, Sobolev transport (ST) (Le et al., 2022), OW (Sturm, 2011), and OT for graph-based measures.[13]

**Theorem 4.1** (Metrization). *The generalized Sobolev IPM with Musielak regularization $\widehat{\mathcal{GS}}_\Phi$ is a metric on the space $\mathcal{P}(\mathbb{G})$ of probability measures on graph $\mathbb{G}$.*

The GSI-M is monotone with respect to the $N$-function $\Phi$ as shown in the next result. Consequently, it may enclose a stronger notion of metrics than Sobolev IPM for comparing graph-based measures.

**Proposition 4.2** (GSI-M with different $N$-functions). *For any two $N$-functions $\Phi_1, \Phi_2$ satisfying $\Phi_1(t) \le \Phi_2(t)$ for all $t \in \mathbb{R}_+$, and $\mu, \nu \in \mathcal{P}(\mathbb{G})$, then we have*

$$\widehat{\mathcal{GS}}_{\Phi_1}(\mu, \nu) \le \widehat{\mathcal{GS}}_{\Phi_2}(\mu, \nu).$$

We next theoretically establish relations between our proposed GSI-M and other related IPM/transport approaches, including original GSI, the original Sobolev IPM, the regularized Sobolev IPM, GST, ST, OW, and OT for graph-based measures. Consequently, we can position the proposed GSI-M within the big research picture of the literature.

**Connection to the original generalized Sobolev IPM.** We next show that the GSI-M is equivalent to the original generalized Sobolev IPM.

**Theorem 4.3** (Relation with original generalized Sobolev IPM). *For a nonnegative Borel measure $\lambda$ on $\mathbb{G}$, $N$-function $\Phi$, $\mu, \nu \in \mathcal{P}(\mathbb{G})$, then*

$$\frac{1}{2} \mathcal{GS}_\Phi(\mu, \nu) \le \widehat{\mathcal{GS}}_\Phi(\mu, \nu) \le (1 + \lambda(\mathbb{G})) \mathcal{GS}_\Phi(\mu, \nu),$$
(17)

*Hence, the generalized Sobolev IPM with Musielak regularization is equivalent to the original GSI.*

---

[11]Rao & Ren (1991, Theorem 13) derived the necessary and sufficient conditions to obtain the infimum for problem (14).

[12]See §C.1 for further implementation notes for $\gamma_e$ in GSI-M including practical preprocessing procedure and exploiting its sparsity to improve computational complexity.

[13]See §D for a review on Sobolev IPM and its scalable regularization, GST, ST, OW, and standard OT.

**Connection to the regularized Sobolev IPM.** Denote $\hat{\mathcal{S}}_p$ for the regularized $p$-order Sobolev IPM.

**Proposition 4.4** (Connection between GSI-M and regularized Sobolev IPM). *Let* $\Phi(t) := \frac{(p-1)^{p-1}}{p^p} t^p$ *with* $1 < p < \infty$, $\hat{c}_1 := \max(1, \lambda(\mathbb{G})^{-1})^{\frac{1-p}{p}}$, *and* $\hat{c}_2 := \min(1, \lambda(\mathbb{G})^{-1})^{\frac{1-p}{p}}$. *Then, for a nonnegative Borel measure* $\lambda$ *on graph* $\mathbb{G}$, *and measures* $\mu, \nu \in \mathcal{P}(\mathbb{G})$, *we have*

$$\hat{c}_1 \, \hat{\mathcal{S}}_p(\mu, \nu) \leq \widehat{\mathcal{GS}}_\Phi(\mu, \nu) \leq \hat{c}_2 \, \hat{\mathcal{S}}_p(\mu, \nu). \tag{18}$$

**Connection to the original Sobolev IPM.** Denote $\mathcal{S}_p$ for the original $p$-order Sobolev IPM.

**Proposition 4.5** (Connection between GSI-M and original Sobolev IPM). *Let* $\Phi(t) := \frac{(p-1)^{p-1}}{p^p} t^p$ *with* $1 < p < \infty$. *For a nonnegative Borel measure* $\lambda$ *on* $\mathbb{G}$, *and measures* $\mu, \nu \in \mathcal{P}(\mathbb{G})$, *then*

$$c_1 \, \mathcal{S}_p(\mu, \nu) \leq \widehat{\mathcal{GS}}_\Phi(\mu, \nu) \leq c_2 \, \mathcal{S}_p(\mu, \nu), \tag{19}$$

*where* $c_1 := \left[ \frac{\min(1, \lambda(\mathbb{G})^{p-1}) \max(1, \lambda(\mathbb{G})^{-1})^{1-p}}{1 + \lambda(\mathbb{G})^p} \right]^{\frac{1}{p}}$; $c_2 := \left[ \min(1, \lambda(\mathbb{G})^{-1})^{1-p} \max(1, \lambda(\mathbb{G})^{p-1}) \right]^{\frac{1}{p}}$.

**Connection to the generalized Sobolev transport.** Denote $\mathcal{GST}_\Phi$ for the GST with $N$-function $\Phi$.

**Proposition 4.6** (Connection between GSI-M and generalized Sobolev transport). *For a nonnegative Borel measure* $\lambda$ *on* $\mathbb{G}$, *then for all measures* $\mu, \nu \in \mathcal{P}(\mathbb{G})$, *we have*

$$\frac{1}{2} \mathcal{GST}_\Phi(\mu, \nu) \leq \widehat{\mathcal{GS}}_\Phi(\mu, \nu) \leq \mathcal{GST}_\Phi(\mu, \nu). \tag{20}$$

**Connection to the Sobolev transport.** Denote $\mathcal{ST}_p$ for the $p$-order Sobolev transport.

**Proposition 4.7** (Connection between GSI-M and Sobolev transport). *For a nonnegative Borel measure* $\lambda$ *on* $\mathbb{G}$, $N$-*function* $\Phi(t) := \frac{(p-1)^{p-1}}{p^p} t^p$ *with* $1 < p < \infty$, *and* $\mu, \nu \in \mathcal{P}(\mathbb{G})$, *then we have*

$$\frac{1}{2} \mathcal{ST}_p(\mu, \nu) \leq \widehat{\mathcal{GS}}_\Phi(\mu, \nu) \leq \mathcal{ST}_p(\mu, \nu). \tag{21}$$

**Connection to the Orlicz-Wasserstein.** Denote $\mathcal{OW}$ for the Orlicz-Wasserstein.

**Proposition 4.8** (Connection between GSI-M and Orlicz-Wasserstein). *Consider the limit case* $\Phi(t) := t$,[14] *and graph* $\mathbb{G}$ *is a tree.*[15] *For a nonnegative Borel measure* $\lambda$ *on* $\mathbb{G}$, $\mu, \nu \in \mathcal{P}(\mathbb{G})$, *then*

$$\frac{1}{2} \mathcal{OW}(\mu, \nu) \leq \widehat{\mathcal{GS}}_\Phi(\mu, \nu) \leq \mathcal{OW}(\mu, \nu). \tag{22}$$

**Connection to the optimal transport.** Denote $\mathcal{W}_1$ for the 1-order Wasserstein.

**Proposition 4.9** (Connection between GSI-M and optimal transport). *Under the same assumptions in Proposition* 4.8, *then for all measures* $\mu, \nu \in \mathcal{P}(\mathbb{G})$, *we have*

$$\frac{1}{2} \mathcal{W}_1(\mu, \nu) \leq \widehat{\mathcal{GS}}_\Phi(\mu, \nu) \leq \mathcal{W}_1(\mu, \nu). \tag{23}$$

The proofs for these theoretical results (in §4) are respectively placed in §B.5 – §B.13. See §5 for further discussions on these finding results to position the proposed GSI-M in the literature.

## 5   RELATED WORKS AND DISCUSSIONS

In this section, we discuss relations between the proposed GSI-M with related works in the literature.

**Sobolev IPM and its regularization (Le et al., 2025).** Propositions 4.5 and 4.4 show that for a certain given $N$-function, GSI-M $\widehat{\mathcal{GS}}$ is provably equivalent to original Sobolev IPM $\mathcal{S}_p$ and its

---

[14] Although $\Phi(t) = t$ is not an $N$-function due to its linear growth, it can be regarded as the limit $p \to 1^+$ for the function $\Phi(t) = t^p$.

[15] To our knowledge, it is *unknown* whether there is a connection between GSI-M and OW for general graphs with cycles. The finding result in Proposition 4.8 only holds for the tree-structured graph.

scalable regularized approach $\hat{\mathcal{S}}_p$ respectively. Additionally, GSI-M can be considered as a variational generalization for Sobolev IPM and its regularization to incorporate more general geometric prior, beyond the $L^p$ structure.

**GST (Le et al., 2024) and ST (Le et al., 2022).** Propositions 4.6 and 4.7 illustrate that GSI-M is provably equivalent to the GST with the same $N$-function $\Phi$, and ST for a specific $N$-function respectively. Similar to GST and ST, GSI-M constraints on gradient of a critic function where GSI-M may be regarded as a weighted variant of GST. However, we emphasize that the weight function (Equation (3)) plays the key role to establish the equivalence between Orlicz-Sobolev norm and Musielak norm (Theorem 3.2) for the proposed Musielak regularization for GSI problem, which constraints critic function within a unit ball of Orlicz-Sobolev norm involving both the critic function and its gradient. Therefore, GSI-M with *the weight function (Equation (3))* provides an efficient regularization approach for the generalized Sobolev IPM problem, ground-based by our theoretical finding results in Theorem 3.2.[16] Additionally, GSI-M and GST have the flexibility to leverage geometric priors beyond the $L^p$ paradigm, while ST is coupled with the $L^p$ structure within its definition, similar to Sobolev IPM.

**OW (Sturm, 2011) and OT.** Propositions 4.8 and 4.9 show that GSI-M is provably equivalent to OW and OT respectively when graph $\mathbb{G}$ is a tree, and $\Phi(t) = t$ (i.e., the limit case of $N$-function). Additionally, both OW and GSI-M are able to employ Orlicz geometric structure with general $N$-function. However, OW is challenging for computation, limits its practical applications while GSI-M is much more efficient in computation, by simply solving a univariate optimization problem.

**Graph-based measures.** We study Sobolev IPM for *two probability measures* supported on the *same* graph, which is also considered in (Le et al., 2025). We distinguish the considered problem with the research lines on computing either kernels (Borgwardt et al., 2020) or distances/discrepancies (Petric Maretic et al., 2019; Xu et al., 2019; Dong & Sawin, 2020; Brogat-Motte et al., 2022; Bai et al., 2025) between *two input graphs* which are possibly *different*. See Appendix §E for a further discussion.

# 6 EXPERIMENTS

In this section, we illustrate that GSI-M is fast for computation, which is several-order faster than OW, and comparable to GST, a scalable variant of OW for graph-based measures. Additionally, we show initial evidences on the advantages of GSI-M to compare probability measures supported on a given graph for document classification, and several tasks in TDA. These empirical results well-support our claimed contributions in §1.

**Document classification.** We consider 4 real-world document datasets: `TWITTER, RECIPE, CLASSIC, AMAZON` where their properties are given in Figure 2. Following (Le et al., 2025), we apply word2vec to map words into vectors in $\mathbb{R}^{300}$, and use probability measures to represent documents where we regard word-embedding vectors in $\mathbb{R}^{300}$ as supports, and corresponding word frequencies as their support mass.

**TDA.** We consider 2 TDA tasks: (i) orbit recognition on the synthesized `Orbit` dataset (Adams et al., 2017) for linked twist map, a discrete dynamical system modeling flow, which is used to model flows in DNA microarrays (Hertzsch et al., 2007), and (ii) object shape image classification on a 10-class subset of `MPEG7` dataset (Latecki et al., 2000) as in (Le et al., 2025). We summarize the dataset properties in Figure 3. We use persistence diagrams (PD) (Edelsbrunner & Harer, 2008) to represent objects of interest for these tasks. Then, we regard each PD as probability measures where its supports are 2-dimensional topological feature data points with a uniform mass.

**Graph.** We use graph $\mathbb{G}_{\text{Log}}$ with 10K nodes, about 100K edges; and graph $\mathbb{G}_{\text{Sqrt}}$ with 10K nodes, about 1M edges (Le et al., 2025, §5) for experiments, except `MPEG7` where these graphs have 1K nodes, about 7K edges for $\mathbb{G}_{\text{Log}}$, and about 32K edges for $\mathbb{G}_{\text{Sqrt}}$ due to its smaller size. These considered graphs empirically satisfy the assumptions in §2, also observed in (Le et al., 2025).[17]

---

[16]It is *unknown* whether there is any connection between our considered GSI problem and a weighted variant of GST with any given general weight function. Note that our proposed GSI-M is theoretically ground-based on the novel finding results in Theorem 3.2 with the weight function $\hat{w}$ (Equation (3)).

[17]See a review for these graphs in Appendix §E.

**$N$-function.** We consider two popular $N$-functions $\Phi$ in applications: $\Phi_1(t) = \exp(t) - t - 1$, and $\Phi_2(t) = \exp(t^2) - 1$. We also examine the limit case: $\Phi_0(t) = t$.

**Optimization algorithm.** We use a second-order solver (i.e., *fmincon* with Trust Region Reflective solver in MATLAB) for solving the *univariate* optimization problem for GSI-M.

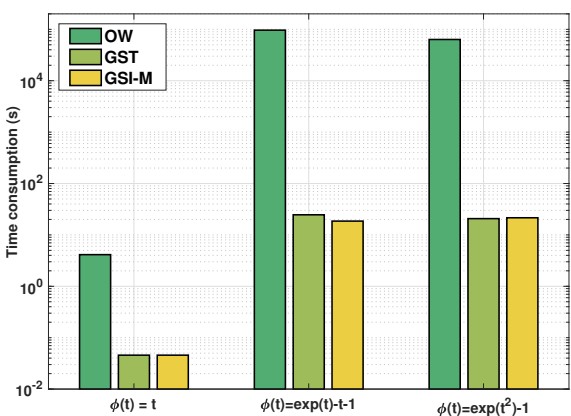

Figure 1: Time consumption on $\mathbb{G}_{\text{Log}}$.

**Classification.** We employ kernelized support vector machine (SVM) for both document classification and TDA tasks. We use kernel $\exp(-\tilde{t}\tilde{d})$ with hyperparameter $\tilde{t} > 0$ and distance $\tilde{d}$ such as GSI-M, GST, and OW for graph-based measures. We follow (Cuturi, 2013) to regularize the Gram matrices of indefinite kernels by adding a sufficiently large diagonal term, and use 1-vs-1 strategy for multi-class classification with SVM. We randomly split each dataset into 70%/30% for training and test with 10 repeats. Generally, we utilize cross validation for hyper-parameters. For SVM regularization hyperparameter, we choose it from $\{0.01, 0.1, 1, 10\}$. For the kernel hyperparameter $\tilde{t}$, we choose it from $\{1/q_s, 1/(2q_s), 1/(5q_s)\}$ where $q_s$ is the $s\%$ quantile of a random subset of distances observed on a training set and $s = 10, 20, \ldots, 90$. For the root node $z_0$ in graph $\mathbb{G}$, we choose it from a random 10-root-node subset of $V$ in $\mathbb{G}$. Note that we include preprocessing procedures, e.g., computing shortest paths, into reported time consumptions.

To illustrate the scale, we note that there are more than 29 million pairs of probability measures in AMAZON, which are required to evaluate distances for kernelized SVM on each run.[18]

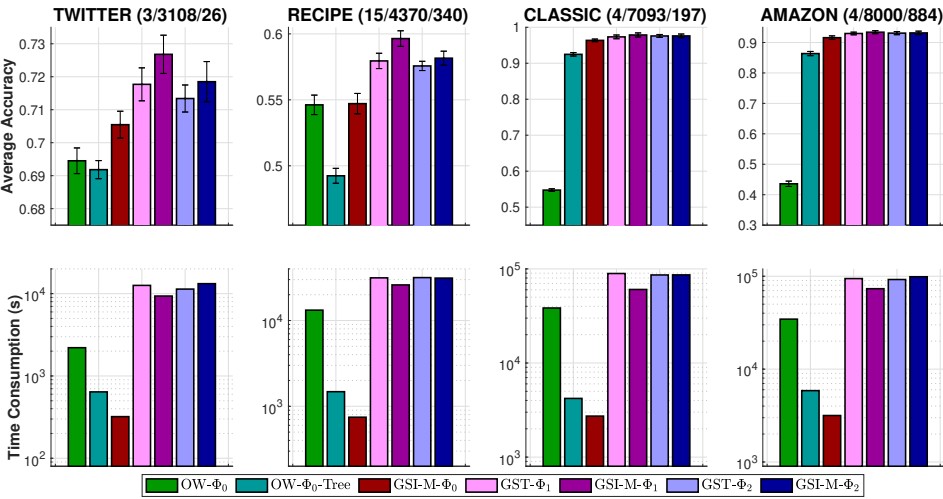

Figure 2: SVM results and time consumption for kernel matrices with graph $\mathbb{G}_{\text{Log}}$. For each dataset, the numbers in the parenthesis are the number of classes; the number of documents; and the maximum number of unique words for each document respectively.

**Results and discussions.** We illustrate the time consumption in Figure 1, and SVM results with corresponding time consumption for document classification and TDA in Figures 2 and 3 respectively, for graph $\mathbb{G}_{\text{Log}}$. Corresponding results for graph $\mathbb{G}_{\text{Sqrt}}$ are placed in Appendix §C.

• **Computational comparision.** We compare the time consumption of GSI-M, GST, and OW with $\Phi_0, \Phi_1, \Phi_2$ on random 10K pairs of measures in AMAZON with graph $\mathbb{G}_{\text{Log}}$, but with 1K nodes and about 7K edges. Figure 1 illustrates that the computation of GSI-M $\widehat{\mathcal{GS}}$ is comparable to GST, and several-order faster than OW. More specifically, $\widehat{\mathcal{GS}}$ is $90\times, 5100\times, 2900\times$ faster than OW for

---

[18]See §C for further details.

$\Phi_0, \Phi_1, \Phi_2$ respectively. For $N$-functions $\Phi_1$ and $\Phi_2$, GSI-M takes less than 22 *seconds* while OW needs at least 17 *hours*, and up to 27 *hours* for the computation.

We remark that following (Le et al., 2024, Remark 4.5), and Proposition 3.7 for the limit $p \to 1^+$, both GSI-M-$\Phi_0$ ($\widehat{\mathcal{GS}}_{\Phi_0}$) and GST-$\Phi_0$ ($\mathcal{GST}_{\Phi_0}$) are equal to the 1-order ST ($\mathcal{ST}_1$), which have closed-form expression for fast computation. Additionally, OW-$\Phi_0$ is equal to the OT with graph metric ground cost (Guha et al., 2023; Le et al., 2024). Consequently, the computations of GSI-M, GST, and OW with the limit case $\Phi_0$ is more efficient than with $N$-functions $\Phi_1, \Phi_2$.

• **Document classification.** We evaluate GSI-M and GST with $\Phi_0, \Phi_1, \Phi_2$, denoted as GSI-M-$\Phi_i$ and GST-$\Phi_0$ for $i = 0, 1, 2$ respectively.[19] For OW, we only use $\Phi_0$ (denoted as OW-$\Phi_0$), but exclude $\Phi_1, \Phi_2$ due to their heavy computations, see Figure 1. We also consider a special case of OW-$\Phi_0$ where it is computed on a random tree extracted from the given graph $\mathbb{G}$, denoted as OW-$\Phi_0$-Tree. We remark that OW-$\Phi_0$-Tree is equal to tree-Wasserstein (Le et al., 2019). Figure 2 illustrates that the performances of GSI-M with all $\Phi$ functions compare favorably to those baselines. Similar to GST, observed in (Le et al., 2024), GSI-M-$\Phi_1$ and GSI-M-$\Phi_2$ improve performances of GSI-M-$\Phi_0$, especially in RECIPE, but their com-

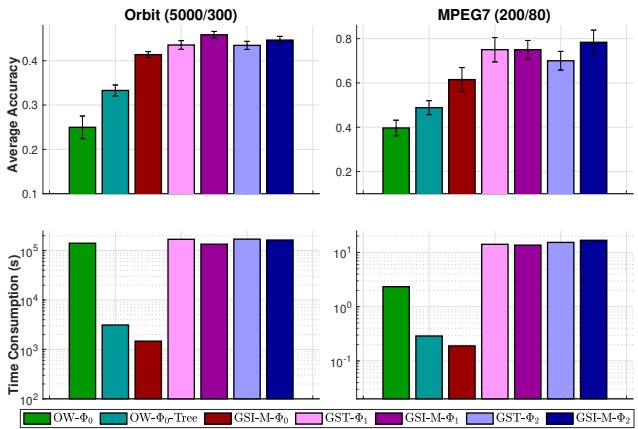

Figure 3: SVM results and time consumption for kernel matrices with graph $\mathbb{G}_{\text{Log}}$. For each dataset, the numbers in the parenthesis are respectively the number of PD; and the maximum number of points in PD.

putational time is several-order slower since GSI-M-$\Phi_0$ has a closed-form expression, following Proposition 3.7 and taking the limit $p \to 1^+$. Therefore, it may imply that Orlicz geometric structure in GSI-M may be helpful for document classification. Additionally, performances of GSI-M compare favorably to those of GST with the same $N$-functions $\Phi_1, \Phi_2$, especially in TWITTER, RECIPE. Results on OW-$\Phi_0$ and OW-$\Phi_0$-Tree also agree with observations in (Le et al., 2024). Performances of OW-$\Phi_0$-Tree are better in CLASSIC, AMAZON, but worse in TWITTER, RECIPE than those of OW-$\Phi_0$. Although OW-$\Phi_0$-Tree only leverages a partial information of $\mathbb{G}$, it forms positive definite kernels while kernels of OW-$\Phi_0$ are indefinite.

• **TDA.** We carry out for the same distances as in document classification. Figure 3 shows that we have similar empirical observations as for document classification. For the same $\Phi$ function, GSI-M and GST are several-order faster OW. Orlicz geometric structure in GSI-M may be also useful for TDA, i.e., performances of GSI-M-$\Phi_1$ GSI-M-$\Phi_2$ compare favorably to those of GSI-M-$\Phi_0$, especially in MPEG7, but they are also several-order slower. Additionally, performances of GSI-M compare favorably to those of GST with the same $N$-functions $\Phi_1, \Phi_2$, especially in MPEG7 for $N$-function $\Phi_2$. Although OW-$\Phi_0$-Tree use a partial information of $\mathbb{G}$, the positive definiteness of its corresponding kernel may help to improve performances of OW-$\Phi_0$, which agrees with observations in (Le et al., 2024).

## 7 CONCLUSION

In this work, we propose the generalized Sobolev IPM (GSI) for graph-based measures by leveraging the set of convex $N$-functions to adopt Orlicz geometric structure for Sobolev IPM beyond its coupled $L^p$ prior. Moreover, we propose a novel regularization for GSI, which is efficient for computation by simply solving a univariate optimization problem. For future works, it is interesting to derive efficient algorithmic approaches for the original Sobolev IPM and its generalization (without regularization), and go beyond the Sobolev geometric structure, e.g., employing more advanced yet challenging geometric structure for IPM such as critic function within unit ball of Besov norm (i.e., Besov IPM) for applications.

---

[19]GSI-M-$\Phi_0$ is equal to GST-$\Phi_0$.

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

# APPENDIX

In this appendix, we provide further theoretical results in §A, and the detailed proofs for theoretical results in the main manuscript and additional results in §B. Then, we describe further experiment experimental details and empirical results in §C, following by brief reviews for related notions for the development of our work and further discussions in §D and §E respectively.

## A   FURTHER THEORETICAL RESULTS

### A.1   AUXILIARY RESULTS

**Lemma A.1.** *Let $\Phi$ be an $N$-function and $f \in WL^1_{\Phi,0}(\mathbb{G}, \lambda)$. Then for any nonnegative weight function $\hat{w}$ and $t > 0$, we have*

$$\int_{\mathbb{G}} \hat{w}(x) \, \Phi\left(\frac{1}{t} \, |f(x)|\right) \lambda(\mathrm{d}x) \leq \frac{1}{\lambda(\mathbb{G})} \int_{\mathbb{G}} \Phi\left(\frac{\lambda(\mathbb{G})}{t} \, |f'(y)|\right) \bar{\lambda}_{\hat{w}}(\Lambda(y))\lambda(\mathrm{d}y), \qquad (24)$$

*where we write $\bar{\lambda}_{\hat{w}}$ for measure $\lambda$ weighted by function $\hat{w}$, i.e., $\int \bar{\lambda}_{\hat{w}}(\mathrm{d}x) := \int \hat{w}(x)\lambda(\mathrm{d}x)$.*

The proof is placed in Appendix §B.14.

**Lemma A.2.** *Let $w_0$ be any positive weight function such that $w_0(x) \geq \lambda(\Lambda(x))$ for all $x \in \mathbb{G}$. Then if function $f \in WL^1_{\Phi,0}(\mathbb{G}, \lambda)$ and scalar $t > 0$ satisfying $\int_{\mathbb{G}} w_0(x) \, \Phi\left(\frac{|f'(x)|}{t}\right) \lambda(\mathrm{d}x) \leq \lambda(\mathbb{G})$, we must have*

$$t \geq \frac{\|f\|_{L_\Phi}}{\lambda(\mathbb{G})}. \qquad (25)$$

The proof is placed in Appendix §B.15.

**Theorem A.3.** *Let $w_0$ be any positive weight function satisfying $w_0(x) \geq \lambda(\Lambda(x))$ for all $x \in \mathbb{G}$. For every function $f \in WL^1_{\Phi,0}(\mathbb{G}, \lambda)$, then*

$$\|f\|_{L_\Phi} \leq \lambda(\mathbb{G}) \, \|f'\|_{L_\Phi^{w_0/\lambda(\mathbb{G})}}. \qquad (26)$$

The proof is placed in Appendix §B.16.

**Lemma A.4.** *Let $\Phi$ be an $N$-function. For any two positive weight functions $w_1, w_2$ such that $w_1(x) \geq w_2(x)$ for all $x \in \mathbb{G}$, and any Borel measurable function $f$ on $\mathbb{G}$, then*

$$\|f\|_{L_\Phi^{w_1}} \geq \|f\|_{L_\Phi^{w_2}}. \qquad (27)$$

The proof is placed in Appendix §B.17.

**Lemma A.5.** *For $N$-function $\Phi$, scalar $t > 0$, and scalar $k \geq 1$, then*

$$\Phi(kt) \geq k\Phi(t). \qquad (28)$$

The proof is placed in Appendix §B.18.

**Lemma A.6.** *For $N$-function $\Phi$, positive weight function $\hat{w}$ on $\mathbb{G}$, $k > 0$, then we have*

$$\|kf\|_{L_\Phi^{\hat{w}}} = k \, \|f\|_{L_\Phi^{\hat{w}}}. \qquad (29)$$

The proof is placed in Appendix §B.19.

### A.2   FURTHER THEORETICAL RESULTS

**Special case with closed-form expression.**   For specific $N$-function, the generalized Sobolev IPM can yield a closed-form expression for computation.

**Proposition A.7** (Closed-form expression). *For $N$-function $\Phi(t) = \frac{(p-1)^{p-1}}{p^p}t^p$ with $1 < p < \infty$, the generalized Sobolev IPM with Musielak regularization has a closed-form expression as follows:*

$$\widehat{\mathcal{GS}}_\Phi(\mu, \nu) = \left[\int_{\mathbb{G}} \hat{w}(x)^{1-p} \, |\mu(\Lambda(x)) - \nu(\Lambda(x))|^p \, \lambda(\mathrm{d}x)\right]^{\frac{1}{p}}. \qquad (30)$$

The proof is placed in Appendix §B.20.

### A.3 DISCRETE CASE FOR REGULARIZED GENERALIZED SOBOLEV IPM WITH POPULAR $N$-FUNCTIONS

Following Theorem 3.6, the discrete case of the regularized generalized Sobolev IPM $\widehat{\mathcal{GS}}_\Phi$ is

$$\widehat{\mathcal{GS}}_\Phi(\mu, \nu) = \inf_{k>0} \frac{1}{k} \left( 1 + \sum_{e \in E} \left[ \int_0^1 \bar{w}_t(e) \, \Phi\left( \frac{k|\bar{h}(e)|}{\bar{w}_t(e)} \right) w_e \mathrm{d}t \right] \right), \tag{31}$$

where $\bar{h}(e) := \mu(\gamma_e) - \nu(\gamma_e)$ for every edge $e \in E$, and $\bar{w}_t(e) := \frac{w_e}{\lambda(\mathbb{G})}t + 1 + \frac{\lambda(\gamma_e)}{\lambda(\mathbb{G})}$ for all edge $e \in E$ and $t \in [0, 1]$.

We consider popular practical $N$-functions $\Phi$ for the GST and the OW: $\Phi_1(t) = \exp(t) - t - 1$, and $\Phi_2(t) = \exp(t^2) - 1$. We also examine the limit case: $\Phi_0(t) = t$.

For each edge $e \in E$, we want to compute

$$\mathcal{A}_\Phi(e) := \int_0^1 \left[ \frac{w_e}{\lambda(\mathbb{G})}t + 1 + \frac{\lambda(\gamma_e)}{\lambda(\mathbb{G})} \right] \Phi\left( \frac{k|\bar{h}(e)|}{\frac{w_e}{\lambda(\mathbb{G})}t + 1 + \frac{\lambda(\gamma_e)}{\lambda(\mathbb{G})}} \right) w_e \mathrm{d}t \tag{32}$$

For simplicity, let denote the exponential integral function for $z > 0$ as follow:

$$Ei(z) = \int_{-\infty}^z \frac{\exp(t)}{t} \mathrm{d}t, \tag{33}$$

and notice that

$$\frac{\mathrm{d}}{\mathrm{d}z} Ei(z) = \frac{\exp(z)}{z}, \quad z \neq 0, \tag{34}$$

and $Ei(z) = -E_1(-z)$, where $E_1(z) = \int_z^\infty \frac{\exp(-t)}{t} \mathrm{d}t$.

**For the limit case:** $\Phi_0(t) = t$. We have

$$\mathcal{A}_{\Phi_0}(e) := \int_0^1 k|\bar{h}(e)|w_e \mathrm{d}t = k|\bar{h}(e)|w_e. \tag{35}$$

**For $N$-function:** $\Phi_1(t) = \exp(t) - t - 1$. For $a > 0, b > 0$, we consider

$$A_1 := \beta \int_0^1 (at + b) \, \Phi_1\left( \frac{\alpha}{at + b} \right) \mathrm{d}t. \tag{36}$$

See Appendix §B.21 for the detailed computation of $A_1$ where its results are summarized as follow:

$$A_1 = \frac{\beta}{2a} \left[ -\alpha^2 \left[ Ei\left( \frac{\alpha}{a+b} \right) - Ei\left( \frac{\alpha}{b} \right) \right] + \alpha \left[ (a+b)\exp\left( \frac{\alpha}{a+b} \right) - b\exp\left( \frac{\alpha}{b} \right) \right] \right.$$
$$\left. + \left[ (a+b)^2 \exp\left( \frac{\alpha}{a+b} \right) - b^2 \exp\left( \frac{\alpha}{b} \right) \right] - 2\alpha a - \left[ (a+b)^2 - b^2 \right] \right]. \tag{37}$$

Therefore, we obtain

$$\mathcal{A}_{\Phi_1} = A_1, \tag{38}$$

where $\beta = w_e, a = \frac{w_e}{\lambda(\mathbb{G})}, b = 1 + \frac{\lambda(\gamma_e)}{\lambda(\mathbb{G})}, \alpha = k|\bar{h}(e)|$.

Additionally, the first-order derivative of $A_1$ w.r.t. $k$ is as follows:

$$\frac{\mathrm{d}}{\mathrm{d}k} A_1 = \frac{\beta|\bar{h}(e)|}{a} \left\{ -\alpha \left[ Ei\left( \frac{\alpha}{a+b} \right) - Ei\left( \frac{\alpha}{b} \right) \right] + (a+b)\exp\left( \frac{\alpha}{a+b} \right) - b\exp\left( \frac{\alpha}{b} \right) - a \right\}. \tag{39}$$

Moreover, its second-order derivative of $A_1$ w.r.t. $k$ is as follows:

$$\frac{\mathrm{d}^2}{\mathrm{d}k^2} A_1 = -\frac{\beta|\bar{h}(e)|^2}{a} \left[ Ei\left( \frac{\alpha}{a+b} \right) - Ei\left( \frac{\alpha}{b} \right) \right]. \tag{40}$$

**For $N$-function:** $\Phi_2(t) = \exp(t^2) - 1$. For $a > 0, b > 0$, we consider

$$A_2 := \beta \int_0^1 (at + b)\, \Phi_2\left(\frac{\alpha}{at + b}\right) \mathrm{d}t. \tag{41}$$

See Appendix §B.22 for the detailed computation of $A_2$ where its result is summarized as follow:

$$A_2 = \frac{\beta}{2a}\left[ -\alpha^2\left[Ei\left(\frac{\alpha^2}{(a+b)^2}\right) - Ei\left(\frac{\alpha^2}{b^2}\right)\right]\right.$$

$$\left. + \left[(a+b)^2 \exp\left(\frac{\alpha^2}{(a+b)^2}\right) - b^2 \exp\left(\frac{\alpha^2}{b^2}\right)\right] - \left[(a+b)^2 - b^2\right]\right]. \tag{42}$$

Therefore, we get

$$\mathcal{A}_{\Phi_2} = A_2, \tag{43}$$

where $\beta = w_e, a = \frac{w_e}{\lambda(\mathbb{G})}, b = 1 + \frac{\lambda(\gamma_e)}{\lambda(\mathbb{G})}, \alpha = k|\bar{h}(e)|$.

Additionally, the first-order derivative of $A_2$ w.r.t. $k$ is as follows:

$$\frac{\mathrm{d}A_2}{\mathrm{d}k} = -\frac{\beta\alpha|\bar{h}(e)|}{a}\left[Ei\left(\frac{\alpha^2}{(a+b)^2}\right) - Ei\left(\frac{\alpha^2}{b^2}\right)\right]. \tag{44}$$

Moreover, its second-order derivative of $A_2$ w.r.t. $k$ is as follows:

$$\frac{\mathrm{d}^2}{\mathrm{d}k^2}A_2 = -\frac{\beta|\bar{h}(e)|^2}{a}\left[Ei\left(\frac{\alpha^2}{(a+b)^2}\right) - Ei\left(\frac{\alpha^2}{b^2}\right) + 2\exp(\alpha^2/(a+b)^2) - 2\exp(\alpha^2/b^2)\right]. \tag{45}$$

## B  PROOFS

In this section, we give detailed proofs for the theoretical results in the main manuscript and additional results in Appendix §A.

### B.1  PROOF FOR THEOREM 3.2

*Proof.* We first derive the lower bound as follows:

$$\begin{aligned}\|f\|_{WL_\Phi^1} &= \|f\|_{L_\Phi} + \|f'\|_{L_\Phi} \\ &\geq \|f'\|_{L_\Phi}. \end{aligned} \tag{46}$$

Additionally, for any scalar $k \geq 1$, we have

$$\int_{\mathbb{G}} \Phi\left(k\frac{|f'(x)|}{t}\right)\lambda(\mathrm{d}x) \geq \int_{\mathbb{G}} k\, \Phi\left(\frac{|f'(x)|}{t}\right)\lambda(\mathrm{d}x) \tag{47}$$

$$\geq \int_{\mathbb{G}} \frac{k}{2}\left(1 + \frac{\lambda(\Lambda(x))}{\lambda(\mathbb{G})}\right)\Phi\left(\frac{|f'(x)|}{t}\right)\lambda(\mathrm{d}x) \tag{48}$$

$$= \int_{\mathbb{G}} \frac{k}{2}\hat{w}(x)\, \Phi\left(\frac{|f'(x)|}{t}\right)\lambda(\mathrm{d}x).$$

The first inequality in (47) is due to Lemma A.5, and the second inequality in (48) comes from the property of the length measure $\lambda$, i.e., $\lambda(\Lambda(x)) \leq \lambda(\mathbb{G})$ for all $x \in \mathbb{G}$.

Therefore, we obtain

$$\left\{t > 0 \mid \int_{\mathbb{G}} \Phi\left(k\frac{|f'(x)|}{t}\right)\lambda(\mathrm{d}x) \leq 1\right\} \subseteq \left\{t > 0 \mid \int_{\mathbb{G}} \frac{k}{2}\hat{w}(x)\, \Phi\left(\frac{|f'(x)|}{t}\right)\lambda(\mathrm{d}x) \leq 1\right\}.$$

Notice that the infimum of a set is smaller than or equal to the infimum of its subset. Consequently, we obtain

$$k\,\|f'\|_{L_\Phi} \geq \|f'\|_{L_\Phi^{k\hat{w}/2}}.$$

Therefore, we have shown that

$$\|f'\|_{L_\Phi} \geq \frac{1}{k} \|f'\|_{L_\Phi^{k\hat{w}/2}}, \quad \text{for every } k \geq 1. \tag{49}$$

By choosing $k = 2$ in (49) and combining with the inequality in (46), we obtain the lower bound as follows:

$$\|f\|_{WL_\Phi^1} \geq \frac{1}{2} \|f'\|_{L_\Phi^{\hat{w}}}.$$

Next, let weight function $w_1(x) := \frac{\lambda(\Lambda(x))}{\lambda(\mathbb{G})}$ for all $x \in \mathbb{G}$, we then derive the upper bound as follows:

$$\begin{aligned}
\|f\|_{WL_\Phi^1} &= \|f\|_{L_\Phi} + \|f'\|_{L_\Phi} \\
&\leq \lambda(\mathbb{G}) \|f'\|_{L_\Phi^{w_1}} + \|f'\|_{L_\Phi} \tag{50} \\
&\leq \lambda(\mathbb{G}) \|f'\|_{L_\Phi^{\hat{w}}} + \|f'\|_{L_\Phi^{\hat{w}}} \tag{51} \\
&= (1 + \lambda(\mathbb{G})) \|f'\|_{L_\Phi^{\hat{w}}},
\end{aligned}$$

where the first inequality in (50) is obtained by using Theorem A.3 with $w_0(x) := \lambda(\Lambda(x))$, and the second inequality (51) is obtained by using Lemma A.4.

Hence, the proof is complete. ∎

### B.2 PROOF FOR THEOREM 3.5

*Proof.* Consider a critic function $f \in WL_{\Psi,0}^1(\mathbb{G}, \lambda)$. Then by Definition 3.1, we have

$$f(x) = f(z_0) + \int_{[z_0,x]} f'(y)\lambda(\mathrm{d}y), \quad \forall x \in \mathbb{G}. \tag{52}$$

Using (52), leveraging the indicator function of the shortest path $[z_0, x]$ (denoted as $\mathbf{1}_{[z_0,x]}$), and notice that $\mu(\mathbb{G}) = 1$, we get

$$\begin{aligned}
\int_{\mathbb{G}} f(x)\mu(\mathrm{d}x) &= \int_{\mathbb{G}} f(z_0)\mu(\mathrm{d}x) + \int_{\mathbb{G}} \int_{[z_0,x]} f'(y)\lambda(\mathrm{d}y)\mu(\mathrm{d}x) \\
&= f(z_0) + \int_{\mathbb{G}} \int_{\mathbb{G}} \mathbf{1}_{[z_0,x]}(y) f'(y)\lambda(\mathrm{d}y)\mu(\mathrm{d}x).
\end{aligned}$$

Then, applying Fubini's theorem to interchange the order of integration in the above last integral, we obtain

$$\begin{aligned}
\int_{\mathbb{G}} f(x)\mu(\mathrm{d}x) &= f(z_0) + \int_{\mathbb{G}} \int_{\mathbb{G}} \mathbf{1}_{[z_0,x]}(y) f'(y)\mu(\mathrm{d}x)\lambda(\mathrm{d}y) \\
&= f(z_0) + \int_{\mathbb{G}} \left( \int_{\mathbb{G}} \mathbf{1}_{[z_0,x]}(y) \mu(\mathrm{d}x) \right) f'(y)\lambda(\mathrm{d}y).
\end{aligned}$$

Using the definition of $\Lambda(y)$ in Equation (1), we can rewrite it as

$$\int_{\mathbb{G}} f(x)\mu(\mathrm{d}x) = f(z_0) + \int_{\mathbb{G}} f'(y)\mu(\Lambda(y)) \lambda(\mathrm{d}y).$$

By exactly the same arguments, we also have

$$\int_{\mathbb{G}} f(x)\nu(\mathrm{d}x) = f(z_0) + \int_{\mathbb{G}} f'(y)\nu(\Lambda(y)) \lambda(\mathrm{d}y).$$

Consequently, the regularized generalized Sobolev IPM in Equation (12) can be reformulated as

$$\widehat{\mathcal{GS}}_\Phi(\mu, \nu) = \sup_{f \in \mathcal{B}_\Psi^{\hat{w}}} \left| \int_{\mathbb{G}} f'(x) \big[ \mu(\Lambda(x)) - \nu(\Lambda(x)) \big] \lambda(\mathrm{d}x) \right|, \tag{53}$$

where we recall that $\mathcal{B}_\Psi^{\hat{w}} = \left\{ f \in WL^1_{\Psi,0}(\mathbb{G},\lambda) : \|f'\|_{L_\Psi^{\hat{w}}} \leq 1 \right\}$ (see Equation (193)).

Observe that, we have on one hand

$$\{f' :\ f \in \mathcal{B}_\Psi^{\hat{w}}\} \subset \{g \in L_\Psi(\mathbb{G},\lambda) :\ \|g\|_{L_\Psi^{\hat{w}}} \leq 1\}.$$

On the other hand, for any $g \in L_\Psi(\mathbb{G},\lambda)$, we have $g = f'$ with $f(x) := \int_{[z_0,x]} g(y)\lambda(\mathrm{d}y) \in WL^1_{\Psi,0}(\mathbb{G},\lambda)$.

Therefore, we conclude that

$$\{f' :\ f \in \mathcal{B}_\Psi^{\hat{w}}\} = \{g \in L_\Psi(\mathbb{G},\lambda) :\ \|g\|_{L_\Psi^{\hat{w}}} \leq 1\}. \tag{54}$$

Consequently, if we let $\hat{f}(x) := \frac{\mu(\Lambda(x)) - \nu(\Lambda(x))}{\hat{w}(x)}$ for $x \in \mathbb{G}$, then Equation (53) can be recast as

$$\widehat{\mathcal{GS}}_\Phi(\mu,\nu) = \sup_{g \in L_\Psi(\mathbb{G},\lambda):\|g\|_{L_\Psi^{\hat{w}}} \leq 1} \left| \int_\mathbb{G} \hat{w}(x)\hat{f}(x)g(x)\lambda(\mathrm{d}x) \right| \tag{55}$$

Additionally, $\Phi, \Psi$ are a pair of complementary $N$-functions. Therefore, we obtain from (55) and (Musielak, 2006, §13.20)[20] that

$$\widehat{\mathcal{GS}}_\Phi(\mu,\nu) = \|\hat{f}\|_{\Phi,\hat{w}}, \tag{56}$$

where $\|\hat{f}\|_{\Phi,\hat{w}}$ is the Musileak-Orlicz norm with weight function $\hat{w}$, define by (see (Musielak, 2006, §13.11)[21])

$$\|\hat{f}\|_{\Phi,\hat{w}} := \left\{ \int_\mathbb{G} \hat{w}(x)\hat{f}(x)g(x)\lambda(\mathrm{d}x) :\ \int_\mathbb{G} \hat{w}(x)\Psi(|g(x)|)\lambda(\mathrm{d}x) \leq 1 \right\}. \tag{57}$$

By applying (Krasnoselskii & Rutickii, 1961, Theorem 10.5, §10.8)[22], we have

$$\|\hat{f}\|_{\Phi,\hat{w}} = \inf_{k>0} \frac{1}{k}\left(1 + \int_\mathbb{G} \hat{w}(x)\,\Phi\left(k\left|\hat{f}(x)\right|\right)\lambda(\mathrm{d}x)\right).$$

This together with (56) yields

$$\begin{aligned}
\widehat{\mathcal{GS}}_\Phi(\mu,\nu) &= \inf_{k>0}\frac{1}{k}\left(1 + \int_\mathbb{G} \hat{w}(x)\,\Phi\left(k\left|\hat{f}(x)\right|\right)\lambda(\mathrm{d}x)\right)\\
&= \inf_{k>0}\frac{1}{k}\left(1 + \int_\mathbb{G} \hat{w}(x)\,\Phi\left(k\left|\frac{\mu(\Lambda(x)) - \nu(\Lambda(x))}{\hat{w}(x)}\right|\right)\lambda(\mathrm{d}x)\right)\\
&= \inf_{k>0}\frac{1}{k}\left(1 + \int_\mathbb{G} \hat{w}(x)\,\Phi\left(\frac{k}{\hat{w}(x)}\left|\mu(\Lambda(x)) - \nu(\Lambda(x))\right|\right)\lambda(\mathrm{d}x)\right).
\end{aligned}$$

This completes the proof of the theorem.

∎

### B.3 PROOF FOR THEOREM 3.6

*Proof.* We consider the length measure on graph $\mathbb{G}$ for $\lambda$. Thus, we have $\lambda(\{x\}) = 0$ for all $x \in \mathbb{G}$. Consequently, we have

$$\widehat{\mathcal{GS}}_\Phi(\mu,\nu) = \inf_{k>0}\frac{1}{k}\left(1 + \sum_{e=\langle u,v\rangle \in E}\int_{(u,v)} \hat{w}(x)\,\Phi\left(\frac{k}{\hat{w}(x)}\left|\mu(\Lambda(x)) - \nu(\Lambda(x))\right|\right)\lambda(\mathrm{d}x)\right). \tag{58}$$

---

[20]Also see (Rao & Ren, 1991, Proposition 10, §3.4).

[21]Also see (Rao & Ren, 1991, Definition 2, §3.3), (Harjulehto & Hästö, 2019, Definition 3.4.5).

[22]See also (Rao & Ren, 1991, Theorem 13, §3.3).

Additionally, we consider input probability measures $\mu, \nu$ supported on nodes in $V$ of graph $\mathbb{G}$. Thus, for all edge $e = \langle u, v \rangle \in E$, and any point $x \in (u, v)$, we have

$$\mu(\Lambda(x)) - \nu(\Lambda(x)) = \mu(\Lambda(x) \setminus (u, v)) - \nu(\Lambda(x) \setminus (u, v)). \tag{59}$$

Moreover, let us consider edge $e = \langle u, v \rangle \in E$. Then for any $x \in (u, v)$, we have $y \in \mathbb{G} \setminus (u, v)$ belongs to $\Lambda(x)$ if and only if $y \in \gamma_e$ where we recall that $\Lambda(x)$ and $\gamma_e$ are defined in Equation (1). Thus, we have

$$\Lambda(x) \setminus (u, v) = \gamma_e, \qquad \forall x \in (u, v). \tag{60}$$

Using Equations (59) and (60), we can rewrite Equation (67) as

$$\widehat{\mathcal{GS}}_\Phi(\mu, \nu) = \inf_{k > 0} \frac{1}{k} \left( 1 + \sum_{e = \langle u, v \rangle \in E} \int_{(u, v)} \hat{w}(x) \, \Phi\left( \frac{k}{\hat{w}(x)} |\mu(\gamma_e) - \nu(\gamma_e)| \right) \lambda(\mathrm{d}x) \right). \tag{61}$$

We next want to compute the integral in (69) for each edge $\langle u, v \rangle \in E$.

For this, recall that $\hat{w}(x) = 1 + \frac{\lambda(\Lambda(x))}{\lambda(\mathbb{G})}, \forall x \in \mathbb{G}$ (see Equation (9)). Without loss of generality, assume that $d_\mathbb{G}(z_0, u) \leq d_\mathbb{G}(z_0, v)$, i.e., among two nodes $u, v$ of the edge $e$, node $v$ is farther away from the root node $z_0$ than node $u$.

Notice that for $x \in (u, v)$, we can write $x = v + t(u - v)$ for $t \in (0, 1)$. With this change of variable, we have

$$\int_{(u,v)} \left[ 1 + \frac{\lambda(\Lambda(x))}{\lambda(\mathbb{G})} \right] \Phi\left( \frac{k}{1 + \frac{\lambda(\Lambda(x))}{\lambda(\mathbb{G})}} |\mu(\gamma_e) - \nu(\gamma_e)| \right) \lambda(\mathrm{d}x) \tag{62}$$

$$= \int_0^1 \left[ 1 + \frac{\lambda(\Lambda(v + t(u - v)))}{\lambda(\mathbb{G})} \right] \Phi\left( \frac{k}{1 + \frac{\lambda(\Lambda(v+t(u-v)))}{\lambda(\mathbb{G})}} |\mu(\gamma_e) - \nu(\gamma_e)| \right) w_e \mathrm{d}t. \tag{63}$$

Moreover, we have

$$\lambda(\Lambda(v + t(u - v))) = \lambda(\Lambda(v)) + \lambda([v, v + t(u - v)]) = \lambda(\gamma_e) + w_e t.$$

Therefore,

$$\int_{(u,v)} \left[ 1 + \frac{\lambda(\Lambda(x))}{\lambda(\mathbb{G})} \right] \Phi\left( \frac{k}{1 + \frac{\lambda(\Lambda(x))}{\lambda(\mathbb{G})}} |\mu(\gamma_e) - \nu(\gamma_e)| \right) \lambda(\mathrm{d}x) \tag{64}$$

$$= \int_0^1 \left[ 1 + \frac{\lambda(\gamma_e) + w_e t}{\lambda(\mathbb{G})} \right] \Phi\left( \frac{k}{1 + \frac{\lambda(\gamma_e) + w_e t}{\lambda(\mathbb{G})}} |\mu(\gamma_e) - \nu(\gamma_e)| \right) w_e \mathrm{d}t \tag{65}$$

$$= \int_0^1 \left[ \frac{w_e}{\lambda(\mathbb{G})} t + 1 + \frac{\lambda(\gamma_e)}{\lambda(\mathbb{G})} \right] \Phi\left( \frac{k |\mu(\gamma_e) - \nu(\gamma_e)|}{\frac{w_e}{\lambda(\mathbb{G})} t + 1 + \frac{\lambda(\gamma_e)}{\lambda(\mathbb{G})}} \right) w_e \mathrm{d}t \tag{66}$$

Hence, the proof is complete.

∎

### B.4 Proof for Proposition 3.7

*Proof.* For the length measure $\lambda$ on graph $\mathbb{G}$, we have $\lambda(\{x\}) = 0$ for all $x \in \mathbb{G}$. Consequently, for $N$-function $\Phi(t) = \frac{(p-1)^{p-1}}{p^p} t^p$ with $1 < p < \infty$, from the closed-form expression of the generalized Sobolev IPM with Musielak regularization in Proposition A.7, we obtain

$$\widehat{\mathcal{GS}}_\Phi(\mu, \nu)^p = \sum_{e = \langle u, v \rangle \in E} \int_{(u,v)} \hat{w}(x)^{1-p} |\mu(\Lambda(x)) - \nu(\Lambda(x))|^p \lambda(\mathrm{d}x). \tag{67}$$

Additionally, for input measures $\mu, \nu$ supported on nodes in $V$ of graph $\mathbb{G}$, then for all edge $e = \langle u, v \rangle \in E$, and any point $x \in (u, v)$, we have

$$\mu(\Lambda(x)) - \nu(\Lambda(x)) = \mu(\Lambda(x) \setminus (u, v)) - \nu(\Lambda(x) \setminus (u, v)).$$

Therefore, from Equation (67), we obtain

$$\widehat{\mathcal{GS}}_\Phi(\mu, \nu)^p = \sum_{e = \langle u, v \rangle \in E} \int_{(u,v)} \hat{w}(x)^{1-p} \left| \mu(\Lambda(x) \setminus (u, v)) - \nu(\Lambda(x) \setminus (u, v)) \right|^p \lambda(\mathrm{d}x). \quad (68)$$

Moreover, consider edge $e = \langle u, v \rangle \in E$, for any $x \in (u, v)$, then we have $y \in \mathbb{G} \setminus (u, v)$ belongs to $\Lambda(x)$ if and only if $y \in \gamma_e$.[23] Thus, we have

$$\Lambda(x) \setminus (u, v) = \gamma_e, \qquad \forall x \in (u, v).$$

Thus, from Equation (68), we obtain

$$\widehat{\mathcal{GS}}_\Phi(\mu, \nu)^p = \sum_{e = \langle u, v \rangle \in E} |\mu(\gamma_e) - \nu(\gamma_e)|^p \int_{(u,v)} \hat{w}(x)^{1-p} \lambda(\mathrm{d}x). \quad (69)$$

From Equation (3), for any $x$ in $\mathbb{G}$, we have

$$\hat{w}(x) = 1 + \frac{\lambda(\Lambda(x))}{\lambda(\mathbb{G})}. \quad (70)$$

Without loss of generality, for any edge $e = \langle u, v \rangle \in E$, assume that $d_{\mathbb{G}}(z_0, u) \leq d_{\mathbb{G}}(z_0, v)$, i.e., among two nodes $u, v$ of the edge $e$, node $v$ is farther away from the root node $z_0$ than node $u$.

Observe that for $x \in (u, v)$, then $x = v + t(u - v)$ for $t \in (0, 1)$. Using this change of variable, we obtain

$$\int_{(u,v)} \left[ 1 + \frac{\lambda(\Lambda(x))}{\lambda(\mathbb{G})} \right]^{1-p} \lambda(\mathrm{d}x) = \int_0^1 \left[ 1 + \lambda(\mathbb{G})^{-1} \lambda(\Lambda(v + t(u - v))) \right]^{1-p} w_e \mathrm{d}t$$

Additionally, notice that

$$\lambda(\Lambda(v + t(u - v))) = \lambda(\Lambda(v)) + \lambda([v, v + t(u - v)]) = \lambda(\gamma_e) + w_e t.$$

Thus, we obtain

$$\int_{(u,v)} \left[ 1 + \frac{\lambda(\Lambda(x))}{\lambda(\mathbb{G})} \right]^{1-p} \lambda(\mathrm{d}x) = \int_0^1 \left[ 1 + \lambda(\mathbb{G})^{-1} \lambda(\gamma_e) + \lambda(\mathbb{G})^{-1} w_e t \right]^{1-p} w_e \mathrm{d}t.$$

Furthermore, the last integral can be computed easily depending on the case $p = 2$ or $p \neq 2$. Consequently, we obtain

$$\int_{(u,v)} \left[ 1 + \frac{\lambda(\Lambda(x))}{\lambda(\mathbb{G})} \right]^{1-p} \lambda(\mathrm{d}x) = \begin{cases} \lambda(\mathbb{G}) \log \left( 1 + \frac{w_e}{\lambda(\mathbb{G}) + \lambda(\gamma_e)} \right) & \text{if } p = 2, \\ \frac{(\lambda(\mathbb{G}) + \lambda(\gamma_e) + w_e)^{2-p} - (\lambda(\mathbb{G}) + \lambda(\gamma_e))^{2-p}}{(2-p)\lambda(\mathbb{G})^{1-p}} & \text{otherwise.} \end{cases}$$

Thus, we have $\int_{(u,v)} \left[ 1 + \frac{\lambda(\Lambda(x))}{\lambda(\mathbb{G})} \right]^{1-p} \lambda(\mathrm{d}x) = \beta_e$ (see Equation (16)).

Consequently, from Equation (69), we obtain

$$\widehat{\mathcal{GS}}_\Phi(\mu, \nu) = \left( \sum_{e \in E} \beta_e |\mu(\gamma_e) - \nu(\gamma_e)|^p \right)^{\frac{1}{p}}. \quad (71)$$

Hence, the proof is completed.

$\blacksquare$

---

[23] See Equation (1) for the definitions of $\Lambda(x)$ and $\gamma_e$.

## B.5 PROOF FOR LEMMA 4.1

*Proof.* We will prove that the regularized generalized Sobolev IPM $\widehat{\mathcal{GS}}_\Phi$ satisfies: (i) nonnegativity, (ii) indiscernibility, (iii) symmetry, and (iv) triangle inequality.

**(i) Nonnegativity.** By choosing $f = 0$ in Definition 3.4, we have that $\widehat{\mathcal{GS}}_\Phi(\mu, \nu) \geq 0$ for every pair of probability measures $(\mu, \nu)$ in $\mathcal{P}(\mathbb{G}) \times \mathcal{P}(\mathbb{G})$. Therefore, the regularized generalized Sobolev IPM $\widehat{\mathcal{GS}}_\Phi$ is nonnegative.

**(ii) Indiscernibility.** Assume that $\widehat{\mathcal{GS}}_\Phi(\mu, \nu) = 0$, then we have

$$\int_{\mathbb{G}} f(x)\mu(\mathrm{d}x) - \int_{\mathbb{G}} f(x)\nu(\mathrm{d}x) = 0, \tag{72}$$

for all $f \in WL^1_{\Psi,0}(\mathbb{G}, \lambda)$ satisfying the constraint $\|f'\|_{L^{\hat{w}}_\Psi} \leq 1$.

Let $g \in WL^1_{\Psi,0}(\mathbb{G}, \lambda)$ be any nonconstant function. Then $\alpha := \|g'\|_{L_\Psi} > 0$. Then by choosing $f := \frac{g}{\alpha}$, we have $f \in WL^1_{\Psi,0}(\mathbb{G}, \lambda)$ with $\|f'\|_{L_\Psi} = \|\frac{g'}{\alpha}\|_{L_\Psi} = \frac{1}{\alpha}\|g'\|_{L_\Psi} = 1$ where we use Lemma A.6 for the second equation. Hence, it follows from (72) that

$$\int_{\mathbb{G}} \frac{g(x)}{\alpha}\mu(\mathrm{d}x) - \int_{\mathbb{G}} \frac{g(x)}{\alpha}\nu(\mathrm{d}x) = 0,$$

which implies that

$$\int_{\mathbb{G}} g(x)\mu(\mathrm{d}x) = \int_{\mathbb{G}} g(x)\nu(\mathrm{d}x). \tag{73}$$

Thus, we have shown that (73) holds true for every nonconstant function $g \in WL^1_{\Psi,0}(\mathbb{G}, \lambda)$. Additionally, Equation (73) is also obviously true for any constant function $g$. Therefore, we obtain

$$\int_{\mathbb{G}} g(x)\mu(\mathrm{d}x) = \int_{\mathbb{G}} g(x)\nu(\mathrm{d}x),$$

for every $g \in WL^1_{\Psi,0}(\mathbb{G}, \lambda)$, which gives $\mu = \nu$ as desired.

**(iii) Symmetry.** This property is obvious from Definition 3.4 as the value $\widehat{\mathcal{GS}}_\Phi(\mu, \nu)$ is unchanged when the role of $\mu$ and $\nu$ is interchanged, i.e.,

$$\widehat{\mathcal{GS}}_\Phi(\mu, \nu) = \widehat{\mathcal{GS}}_\Phi(\nu, \mu).$$

**(iv) Triangle inequality.** Let $\mu, \nu, \sigma$ be probability measures in $\mathcal{P}(\mathbb{G})$, then for any function $f \in WL^1_{\Psi,0}(\mathbb{G}, \lambda)$ satisfying the constraint $\|f'\|_{L^{\hat{w}}_\Psi} \leq 1$, we have

$$\left| \int_{\mathbb{G}} f(x)\mu(\mathrm{d}x) - \int_{\mathbb{G}} f(x)\nu(\mathrm{d}x) \right|$$
$$= \left| \left[ \int_{\mathbb{G}} f(x)\mu(\mathrm{d}x) - \int_{\mathbb{G}} f(x)\sigma(\mathrm{d}x) \right] + \left[ \int_{\mathbb{G}} f(x)\sigma(\mathrm{d}x) - \int_{\mathbb{G}} f(x)\nu(\mathrm{d}x) \right] \right|$$
$$\leq \left| \int_{\mathbb{G}} f(x)\mu(\mathrm{d}x) - \int_{\mathbb{G}} f(x)\sigma(\mathrm{d}x) \right| + \left| \int_{\mathbb{G}} f(x)\sigma(\mathrm{d}x) - \int_{\mathbb{G}} f(x)\nu(\mathrm{d}x) \right|$$
$$\leq \widehat{\mathcal{GS}}_\Phi(\mu, \sigma) + \widehat{\mathcal{GS}}_\Phi(\sigma, \nu).$$

By taking the supremum over $f$, this implies that

$$\widehat{\mathcal{GS}}_\Phi(\mu, \nu) \leq \widehat{\mathcal{GS}}_\Phi(\mu, \sigma) + \widehat{\mathcal{GS}}_\Phi(\sigma, \nu).$$

Due to these above properties, we conclude that the regularized generalized Sobolev IPM $\widehat{\mathcal{GS}}_\Phi$ is a metric on the space $\mathcal{P}(\mathbb{G})$ of probability measures on graph $\mathbb{G}$. ∎

### B.6 PROOF FOR PROPOSITION 4.2

$$\widehat{\mathcal{GS}}_\Phi(\mu,\nu) := \sup_{f \in \mathcal{B}_\Psi^{\hat w}} \left| \int_\mathbb{G} f(x)\mu(\mathrm{d}x) - \int_\mathbb{G} f(y)\nu(\mathrm{d}y) \right|. \tag{74}$$

$$\mathcal{B}_\Psi^{\hat w} := \left\{ f \in WL_{\Psi,0}^1(\mathbb{G},\lambda), \|f'\|_{L_\Psi^{\hat w}} \leq 1 \right\}. \tag{75}$$

*Proof.* Let $\Psi_1$ and $\Psi_2$ be respectively the complementary functions of $\Phi_1$ and $\Phi_2$, consider an arbitrary positive scale $t \in \mathbb{R}_+$, and notice that $\Phi_1 \leq \Phi_2$, then we have:

$$at - \Phi_1(a) \quad \geq \quad at - \Phi_2(a) \ \text{ for every } a \in \mathbb{R}_+,$$
$$\Rightarrow \sup_{a \geq 0} (at - \Phi_1(a)) \quad \geq \quad \sup_{a \geq 0} (at - \Phi_2(a)).$$

This implies that
$$\Psi_1(t) \geq \Psi_2(t), \ \text{ for all } t \in \mathbb{R}_+.$$

It follows that $L_{\Psi_1}^{\hat w}(\mathbb{G},\lambda) \subset L_{\Psi_2}^{\hat w}(\mathbb{G},\lambda)$ and $WL_{\Psi_1,0}^1(\mathbb{G},\lambda) \subset WL_{\Psi_2,0}^1(\mathbb{G},\lambda)$. Moreover, for any fixed Orlicz function $f'$ and any number $t > 0$, we have

$$\int_\mathbb{G} \hat w(x)\, \Psi_1\left( \frac{|f'(x)|}{t} \right) \lambda(\mathrm{d}x) \geq \int_\mathbb{G} \hat w(x)\, \Psi_2\left( \frac{|f'(x)|}{t} \right) \lambda(\mathrm{d}x).$$

Consequently, we obtain

$$\left\{ t > 0 \mid \int_\mathbb{G} \hat w(x)\, \Psi_1\left( \frac{|f'(x)|}{t} \right) \lambda(\mathrm{d}x) \leq 1 \right\} \subset \left\{ t > 0 \mid \int_\mathbb{G} \hat w(x)\, \Psi_2\left( \frac{|f'(x)|}{t} \right) \lambda(\mathrm{d}x) \leq 1 \right\}.$$

Since the infimum of a set is smaller than or equal to the infimum of its subset, we deduce that

$$\|f'\|_{L_{\Psi_1}^{\hat w}} \geq \|f'\|_{L_{\Psi_2}^{\hat w}}.$$

It follows from this and $WL_{\Psi_1,0}^1(\mathbb{G},\lambda) \subset WL_{\Psi_2,0}^1(\mathbb{G},\lambda)$ that

$$\left\{ f \mid f \in WL_{\Psi_1,0}^1(\mathbb{G},\lambda),\, \|f'\|_{L_{\Psi_1}^{\hat w}} \leq 1 \right\} \subset \left\{ f \mid f \in WL_{\Psi_2,0}^1(\mathbb{G},\lambda),\, \|f'\|_{L_{\Psi_2}^{\hat w}} \leq 1 \right\}.$$

Since the supremum of a set is larger than or equal to the supremum of its subset, we conclude that

$$\widehat{\mathcal{GS}}_{\Phi_1}(\mu,\nu) \leq \widehat{\mathcal{GS}}_{\Phi_2}(\mu,\nu),$$

for any input measures $\mu,\nu$ in $\mathcal{P}(\mathbb{G})$. Therefore, the proof is complete. $\blacksquare$

### B.7 PROOF FOR PROPOSITION 4.3

*Proof.* Given a positive scalar $c > 0$, a pair of complementary $N$-functions $\Phi, \Psi$, and let $\mathcal{B}_{\Psi,c}^{\hat w} := \left\{ f \in WL_{\Psi,0}^1(\mathbb{G},\lambda) : \|f'\|_{L_\Psi^{\hat w}} \leq \frac{1}{c} \right\}$. We define the IPM distance w.r.t. $\mathcal{B}_{\Psi,c}^{\hat w}$ as follows

$$\widetilde{\mathcal{GS}}_{\Phi,c}(\mu,\nu) := \sup_{f \in \mathcal{B}_{\Psi,c}^{\hat w}} \left| \int_\mathbb{G} f(x)\mu(\mathrm{d}x) - \int_\mathbb{G} f(y)\nu(\mathrm{d}y) \right|. \tag{76}$$

By exploiting the graph structure for the IPM objective function and applying a similar reasoning as in the proof of identity (53) in §B.2, we can rewrite Equation (76) as

$$\widetilde{\mathcal{GS}}_{\Phi,c}(\mu,\nu) = \sup_{f \in \mathcal{B}_{\Psi,c}^{\hat w}} \left| \int_\mathbb{G} f'(x) \big[ \mu(\Lambda(x)) - \nu(\Lambda(x)) \big] \lambda(\mathrm{d}x) \right|. \tag{77}$$

Additionally, by using a similar reasoning as in the proof of (54) in §B.2, we have

$$\{ f' : f \in \mathcal{B}_{\Psi,c}^{\hat w} \} = \left\{ g \in L_\Psi(\mathbb{G},\lambda) : \|g\|_{L_\Psi^{\hat w}} \leq \frac{1}{c} \right\}. \tag{78}$$

Let $\tilde{f}(x) := \frac{\mu(\Lambda(x)) - \nu(\Lambda(x))}{c\,\hat{w}(x)}$ for $x \in \mathbb{G}$. Then by using (78), Equation (77) can be recasted as

$$
\widetilde{\mathcal{GS}}_{\Phi,c}(\mu,\nu) = \sup_{g \in L_\Psi(\mathbb{G},\lambda):\, \|g\|_{L_\Psi^{\hat{w}}} \le \frac{1}{c}} \left| \int_{\mathbb{G}} \hat{w}(x)\tilde{f}(x)[c\,g(x)]\lambda(\mathrm{d}x) \right|
$$

$$
= \sup_{\tilde{g} \in L_\Psi(\mathbb{G},\lambda):\, \|\tilde{g}\|_{L_\Psi^{\hat{w}}} \le 1} \left| \int_{\mathbb{G}} \hat{w}(x)\tilde{f}(x)\tilde{g}(x)\lambda(\mathrm{d}x) \right|
$$

$$
= \left\| \tilde{f} \right\|_{\Phi,\hat{w}} \tag{79}
$$

$$
= \inf_{k>0} \frac{1}{k}\left(1 + \int_{\mathbb{G}} \hat{w}(x)\,\Phi\left(k\left|\tilde{f}(x)\right|\right)\lambda(\mathrm{d}x)\right) \tag{80}
$$

$$
= \inf_{k>0} \frac{1}{k}\left(1 + \int_{\mathbb{G}} \hat{w}(x)\,\Phi\left(k\left|\frac{\mu(\Lambda(x)) - \nu(\Lambda(x))}{c\,\hat{w}(x)}\right|\right)\lambda(\mathrm{d}x)\right)
$$

$$
= \frac{1}{c}\inf_{(k/c)>0} \frac{1}{(k/c)}\left(1 + \int_{\mathbb{G}} \hat{w}(x)\,\Phi\left(\frac{(k/c)}{\hat{w}(x)}\left|\mu(\Lambda(x)) - \nu(\Lambda(x))\right|\right)\lambda(\mathrm{d}x)\right)
$$

$$
= \frac{1}{c}\,\widehat{\mathcal{GS}}_\Phi(\mu,\nu), \tag{81}
$$

where the third equality in (79) following from (Musielak, 2006, §13.20) and Equation (57), the fourth equality in (80) following from (Krasnoselskii & Rutickii, 1961, Theorem 10.5, §10.8), and the last equality in (81) following from Theorem 3.5.

Additionally, notice that from Theorem 3.2, we have

$$
c_1 \|f'\|_{L_\Psi^{\hat{w}}} \le \|f\|_{WL_\Psi^1} \le c_2 \|f'\|_{L_\Psi^{\hat{w}}}, \tag{82}
$$

where $c_1 = 1/2$, and $c_2 = 1 + \lambda(\mathbb{G})$.

This implies that

$$
\mathcal{B}_{\Psi,c_1}^{\hat{w}} \supseteq \mathcal{B}_\Psi \supseteq \mathcal{B}_{\Psi,c_2}^{\hat{w}}. \tag{83}
$$

Therefore, for probability measures $\mu, \nu \in \mathcal{P}(\mathbb{G})$, we have

$$
\widetilde{\mathcal{GS}}_{\Phi,c_1}(\mu,\nu) \ge \mathcal{GS}_\Phi(\mu,\nu) \ge \widetilde{\mathcal{GS}}_{\Phi,c_2}(\mu,\nu). \tag{84}
$$

It follows from Equations (81) and (84) that

$$
\frac{1}{c_1}\,\widehat{\mathcal{GS}}_\Phi(\mu,\nu) \ge \mathcal{GS}_\Phi(\mu,\nu) \ge \frac{1}{c_2}\,\widehat{\mathcal{GS}}_\Phi(\mu,\nu). \tag{85}
$$

Consequently, we obtain

$$
c_1\,\mathcal{GS}_\Phi(\mu,\nu) \le \widehat{\mathcal{GS}}_\Phi(\mu,\nu) \le c_2\,\mathcal{GS}_\Phi(\mu,\nu). \tag{86}
$$

Hence, we have

$$
\frac{1}{2}\,\mathcal{GS}_\Phi(\mu,\nu) \le \widehat{\mathcal{GS}}_\Phi(\mu,\nu) \le (1 + \lambda(\mathbb{G}))\,\mathcal{GS}_\Phi(\mu,\nu). \tag{87}
$$

The proof is complete.

∎

### B.8 Proof for Proposition 4.4

*Proof.* Following Equation (55) in the proof of Theorem 3.5 in §B.2, we have

$$
\widehat{\mathcal{GS}}_\Phi(\mu,\nu) = \sup_{g \in L_\Psi(\mathbb{G},\lambda):\, \|g\|_{L_\Psi^{\hat{w}}} \le 1} \left| \int_{\mathbb{G}} \hat{w}(x)\hat{f}(x)g(x)\lambda(\mathrm{d}x) \right|, \tag{88}
$$

where $\hat{f}(x) = \frac{\mu(\Lambda(x)) - \nu(\Lambda(x))}{\hat{w}(x)}$ for $x \in \mathbb{G}$.

Additionally, following (Le et al., 2024, Remark A.1), for $\Phi(t) = \frac{(p-1)^{p-1}}{p^p} t^p$ with $1 < p < \infty$, we have its complementary function $\Psi(t) = t^q$ where $q$ is the conjugate of $p$, i.e., $1/q + 1/p = 1$.

Consequently, we have $L_\Psi = L^q$, and $L_\Psi^{\hat{w}} = L_{\hat{w}}^q$, where $L_{\hat{w}}^q$ is the weighted $L^q$ space with weight function $\hat{w}$. Therefore, we can rewrite Equation (88) as follows:

$$\widehat{\mathcal{GS}}_\Phi(\mu, \nu) = \sup_{g \in L^q(\mathbb{G}, \lambda): \|g\|_{L_{\hat{w}}^q} \leq 1} \left| \int_{\mathbb{G}} \hat{w}(x) \hat{f}(x) g(x) \lambda(\mathrm{d}x) \right| \tag{89}$$

$$= \left\| \hat{f} \right\|_{L_{\hat{w}}^p} \tag{90}$$

$$= \left[ \int_{\mathbb{G}} \hat{w}(x) \left| \hat{f}(x) \right|^p \lambda(\mathrm{d}x) \right]^{\frac{1}{p}} \tag{91}$$

$$= \left[ \int_{\mathbb{G}} \hat{w}(x)^{1-p} \left| \mu(\Lambda(x)) - \nu(\Lambda(x)) \right|^p \lambda(\mathrm{d}x) \right]^{\frac{1}{p}}. \tag{92}$$

Moreover, following (Le et al., 2025, Theorem 3.4), the closed-form of the regularized $p$-order Sobolev IPM is as follows:

$$\hat{\mathcal{S}}_p(\mu, \nu) = \left[ \int_{\mathbb{G}} \hat{w}_{\hat{\mathcal{S}}}(x)^{1-p} \left| \mu(\Lambda(x)) - \nu(\Lambda(x)) \right|^p \lambda(\mathrm{d}x) \right]^{\frac{1}{p}}, \tag{93}$$

where the weight function $\hat{w}_{\hat{\mathcal{S}}}(x) = 1 + \lambda(\Lambda(x))$ for all $x \in \mathbb{G}$ (Le et al., 2025, Equation (5)).

Recall that $\hat{w}(x) = 1 + \frac{\lambda(\Lambda(x))}{\lambda(\mathbb{G})}$ for all $x \in \mathbb{G}$. For any $x \in \mathbb{G}$, we have

$$\min(1, \lambda(\mathbb{G})^{-1})(1 + \lambda(\Lambda(x))) \leq 1 + \frac{\lambda(\Lambda(x))}{\lambda(\mathbb{G})} \leq \max(1, \lambda(\mathbb{G})^{-1})(1 + \lambda(\Lambda(x))).$$

Then, for $1 < p < \infty$, we obtain

$$\min(1, \lambda(\mathbb{G})^{-1})^{1-p} \hat{w}_{\mathcal{S}}(x)^{1-p} \geq \hat{w}(x)^{1-p} \geq \max(1, \lambda(\mathbb{G})^{-1})^{1-p} \hat{w}_{\mathcal{S}}(x). \tag{94}$$

Therefore, from Equations (92), (93), (94), we get

$$\hat{c}_1 \hat{\mathcal{S}}_p(\mu, \nu) \leq \widehat{\mathcal{GS}}_\Phi(\mu, \nu) \leq \hat{c}_2 \hat{\mathcal{S}}_p(\mu, \nu), \tag{95}$$

where $\hat{c}_1 = \max(1, \lambda(\mathbb{G})^{-1})^{\frac{1-p}{p}}$ and $\hat{c}_2 = \min(1, \lambda(\mathbb{G})^{-1})^{\frac{1-p}{p}}$.

Hence, the proof is complete. ∎

### B.9 PROOF FOR PROPOSITION 4.5

*Proof.* Following Proposition 4.4, we have

$$\max(1, \lambda(\mathbb{G})^{-1})^{\frac{1-p}{p}} \hat{\mathcal{S}}_p(\mu, \nu) \leq \widehat{\mathcal{GS}}_\Phi(\mu, \nu) \leq \min(1, \lambda(\mathbb{G})^{-1})^{\frac{1-p}{p}} \hat{\mathcal{S}}_p(\mu, \nu). \tag{96}$$

Additionally, following (Le et al., 2025, Theorem 4.2), we have

$$\left[ \frac{\min(1, \lambda(\mathbb{G})^{p-1})}{1 + \lambda(\mathbb{G})^p} \right]^{\frac{1}{p}} \mathcal{S}_p(\mu, \nu) \leq \hat{\mathcal{S}}_p(\mu, \nu) \leq \left[ \max(1, \lambda(\mathbb{G})^{p-1}) \right]^{\frac{1}{p}} \mathcal{S}_p(\mu, \nu) \tag{97}$$

Therefore, from Equations (96), (97), we obtain

$$c_1 \mathcal{S}_p(\mu, \nu) \leq \widehat{\mathcal{GS}}_\Phi(\mu, \nu) \leq c_2 \mathcal{S}_p(\mu, \nu), \tag{98}$$

where $c_1 = \left[ \frac{\min(1, \lambda(\mathbb{G})^{p-1}) \max(1, \lambda(\mathbb{G})^{-1})^{1-p}}{1 + \lambda(\mathbb{G})^p} \right]^{\frac{1}{p}}$; $c_2 = \left[ \min(1, \lambda(\mathbb{G})^{-1})^{1-p} \max(1, \lambda(\mathbb{G})^{p-1}) \right]^{\frac{1}{p}}$.

Hence, the proof is complete. ∎

### B.10 PROOF FOR PROPOSITION 4.6

*Proof.* From Equation (9), for any $x \in \mathbb{G}$, we have

$$1 \leq \hat{w}(x) \leq 2. \tag{99}$$

Then, for any $t > 0$, we have

$$\int_{\mathbb{G}} \Psi\left(\frac{|f'(x)|}{t}\right) \lambda(\mathrm{d}x) \leq \int_{\mathbb{G}} \hat{w}(x) \Psi\left(\frac{|f'(x)|}{t}\right) \lambda(\mathrm{d}x) \leq 2 \int_{\mathbb{G}} \Psi\left(\frac{|f'(x)|}{t}\right) \lambda(\mathrm{d}x). \tag{100}$$

Consequently, we obtain

$$\left\{ t > 0 \mid \int_{\mathbb{G}} \Psi\left(\frac{|f'(x)|}{t}\right) \lambda(\mathrm{d}x) \leq 1 \right\} \tag{101}$$

$$\supseteq \left\{ t > 0 \mid \int_{\mathbb{G}} \hat{w}(x) \Psi\left(\frac{|f'(x)|}{t}\right) \lambda(\mathrm{d}x) \leq 1 \right\}$$

$$\supseteq \left\{ t > 0 \mid 2 \int_{\mathbb{G}} \Psi\left(\frac{|f'(x)|}{t}\right) \lambda(\mathrm{d}x) \leq 1 \right\}.$$

Additionally, observe that by following Lemma A.5, for any $t > 0$, we have

$$2 \int_{\mathbb{G}} \Psi\left(\frac{|f'(x)|}{t}\right) \lambda(\mathrm{d}x) \leq \int_{\mathbb{G}} \Psi\left(2 \frac{|f'(x)|}{t}\right) \lambda(\mathrm{d}x) \tag{102}$$

Consequently, we have

$$\left\{ t > 0 \mid 2 \int_{\mathbb{G}} \Psi\left(\frac{|f'(x)|}{t}\right) \lambda(\mathrm{d}x) \leq 1 \right\} \supseteq \left\{ t > 0 \mid \int_{\mathbb{G}} \Psi\left(2 \frac{|f'(x)|}{t}\right) \lambda(\mathrm{d}x) \leq 1 \right\}. \tag{103}$$

Notice that the infimum of a set is smaller than or equal to the infimum of its subset. Consequently, from Equations (101), and (103), we obtain

$$\|f'\|_{L_\Psi} \leq \|f'\|_{L_\Psi^{\hat{w}}} \leq \inf\left\{ t > 0 \mid \int_{\mathbb{G}} \Psi\left(2 \frac{|f'(x)|}{t}\right) \lambda(\mathrm{d}x) \leq 1 \right\}. \tag{104}$$

Moreover, we also have

$$\inf\left\{ t > 0 \mid \int_{\mathbb{G}} \Psi\left(2 \frac{|f'(x)|}{t}\right) \lambda(\mathrm{d}x) \leq 1 \right\} = 2 \inf\left\{ (t/2) > 0 \mid \int_{\mathbb{G}} \Psi\left(\frac{|f'(x)|}{(t/2)}\right) \lambda(\mathrm{d}x) \leq 1 \right\}$$

$$= 2 \|f'\|_{L_\Psi}. \tag{105}$$

Thus, we obtain

$$\|f'\|_{L_\Psi} \leq \|f'\|_{L_\Psi^{\hat{w}}} \leq 2 \|f'\|_{L_\Psi}. \tag{106}$$

With the technical assumption $f(z_0) = 0$, then $WL_\Psi^1$ is equal to $WL_{\Psi,0}^1$ (as assumed for GSI-M throughout our work). Thus, we have

$$\left\{ f \in WL_\Psi^1, \|f'\|_{L_\Psi} \leq 1 \right\} \supseteq \left\{ f \in WL_{\Psi,0}^1, \|f'\|_{L_\Psi^{\hat{w}}} \leq 1 \right\} \supseteq \left\{ f \in WL_\Psi^1, \|f'\|_{L_\Psi} \leq 1/2 \right\}. \tag{107}$$

Additionally, for a positive scalar $c > 0$, a pair of complementary $N$-function $\Phi, \Psi$, let consider

$$\widetilde{\mathcal{GST}}_{\Phi,c}(\mu, \nu) := \sup_{f \in WL_\Psi^1, \|f'\|_{L_\Psi} \leq 1/c} \left| \int_{\mathbb{G}} f(x)\mu(\mathrm{d}x) - \int_{\mathbb{G}} f(x)\nu(\mathrm{d}x) \right|. \tag{108}$$

Following (Le et al., 2024, Equations (12), (13) in §A.1), we can rewrite $\widetilde{\mathcal{GST}}_{\Phi,c}$ as follows:

$$\widetilde{\mathcal{GST}}_{\Phi,c}(\mu, \nu) = \sup_{g \in L_\Psi(\mathbb{G}, \lambda): \|g\|_{L_\Psi} \leq 1/c} \left| \int_{\mathbb{G}} g(x)h(x) \lambda(\mathrm{d}x) \right|, \tag{109}$$

where $h(x) := \mu(\Lambda(x)) - \nu(\Lambda(x))$ for all $x \in \mathbb{G}$. Consequently, let $\tilde{g} = cg$, we have

$$\widetilde{\mathcal{GST}}_{\Phi,c}(\mu, \nu) = \sup_{\tilde{g} \in L_\Psi(\mathbb{G}, \lambda): \|\tilde{g}\|_{L_\Psi} \leq 1} \left| \int_{\mathbb{G}} \tilde{g}(x) \frac{h(x)}{c} \lambda(\mathrm{d}x) \right| \tag{110}$$

By applying (Rao & Ren, 1991, Proposition 10, pp.81), we obtain

$$\widetilde{\mathcal{GST}}_{\Phi,c}(\mu,\nu) = \left\|\frac{h}{c}\right\|_{\Phi} \tag{111}$$

where $\left\|\frac{h}{c}\right\|_{\Phi}$ is the Orlicz norm (Rao & Ren, 1991, Definition 2, pp.58), defined as

$$\left\|\frac{h}{c}\right\|_{\Phi} := \sup\left\{\int_{\mathbb{G}}\left|\frac{1}{c}h(x)g(x)\right|\lambda(\mathrm{d}x) : \int_{\mathbb{G}}\Psi(|g(x)|)\lambda(\mathrm{d}x) \le 1\right\}. \tag{112}$$

Then, by applying (Rao & Ren, 1991, Theorem 13, pp.69), we have

$$\left\|\frac{h}{c}\right\|_{\Phi} = \inf_{k>0}\frac{1}{k}\left(1 + \int_{\mathbb{G}}\Phi\left(\frac{k}{c}|h(x)|\right)\lambda(\mathrm{d}x)\right).$$

Thus, together with (111), we obtain

$$\widetilde{\mathcal{GST}}_{\Phi,c}(\mu,\nu) = \frac{1}{c}\inf_{(k/c)>0}\frac{1}{(k/c)}\left(1 + \int_{\mathbb{G}}\Phi((k/c)|h(x)|)\lambda(\mathrm{d}x)\right) = \frac{1}{c}\mathcal{GST}_{\Phi}(\mu,\nu). \tag{113}$$

Since the supremum of a set is larger than or equal to the supremum of its subset, then for $\mu,\nu \in \mathcal{P}(\mathbb{G})$, from Equations (107), (108), (113), and consider $c = 2$, then we obtain

$$\mathcal{GST}_{\Phi}(\mu,\nu) \ge \widehat{\mathcal{GS}}(\mu,\nu) \ge \frac{1}{2}\mathcal{GST}_{\Phi}(\mu,\nu). \tag{114}$$

Hence, the proof is complete. ∎

### B.11 PROOF FOR PROPOSITION 4.7

*Proof.* For $N$-function $\Phi(t) = \frac{(p-1)^{p-1}}{p^p}t^p$ with $1 < p < \infty$, following (Le et al., 2024, Proposition 4.4), we have

$$\mathcal{GST}_{\Phi}(\mu,\nu) = \mathcal{ST}_p(\mu,\nu). \tag{115}$$

Additionally, following Proposition 4.6, we have

$$\frac{1}{2}\mathcal{GST}_{\Phi}(\mu,\nu) \le \widehat{\mathcal{GS}}_{\Phi}(\mu,\nu) \le \mathcal{GST}_{\Phi}(\mu,\nu). \tag{116}$$

Thus, from Equations (115), and (116), we obtain

$$\frac{1}{2}\mathcal{ST}_p(\mu,\nu) \le \widehat{\mathcal{GS}}_{\Phi}(\mu,\nu) \le \mathcal{ST}_p(\mu,\nu). \tag{117}$$

Hence, the proof is complete. ∎

### B.12 PROOF FOR PROPOSITION 4.8

*Proof.* For the limit case $\Phi(t) := t$, and graph $\mathbb{G}$ is a tree, then following (Le et al., 2024, Remark 4.5 and Proposition 4.6), we have

$$\mathcal{GST}_{\Phi}(\mu,\nu) = \mathcal{OW}(\mu,\nu). \tag{118}$$

Additionally, following Proposition 4.6, we have

$$\frac{1}{2}\mathcal{GST}_{\Phi}(\mu,\nu) \le \widehat{\mathcal{GS}}_{\Phi}(\mu,\nu) \le \mathcal{GST}_{\Phi}(\mu,\nu). \tag{119}$$

Thus, from Equations (118), and (119), we obtain

$$\frac{1}{2}\mathcal{OW}(\mu,\nu) \le \widehat{\mathcal{GS}}_{\Phi}(\mu,\nu) \le \mathcal{OW}(\mu,\nu). \tag{120}$$

Hence, the proof is complete.

∎

### B.13 PROOF FOR PROPOSITION 4.9

*Proof.* For the limit case $\Phi(t) := t$, and notice that $\lim_{p \to 1^+} \frac{(p-1)^{p-1}}{p^p} t^p = t$ (Le et al., 2024, §A.7), then following (Le et al., 2024, Proposition 4.4), by taking the limit $p \to 1^+$, the closed-form of generalized Sobolev transport (GST) is equal to the 1-order Sobolev transport (ST), i.e.,

$$\mathcal{GST}_\Phi(\mu, \nu) = \mathcal{ST}_1(\mu, \nu). \tag{121}$$

Additionally, suppose that graph $\mathbb{G}$ is a tree, then by following Le et al. (2022), the 1-order ST is in turn equal to the 1-order Wasserstein.

$$\mathcal{ST}_1(\mu, \nu) = \mathcal{W}_1(\mu, \nu). \tag{122}$$

Therefore, from Equations (121), and (122), we obtain

$$\mathcal{GST}_\Phi(\mu, \nu) = \mathcal{W}_1(\mu, \nu). \tag{123}$$

Moreover, following Proposition 4.6, we have

$$\frac{1}{2} \mathcal{GST}_\Phi(\mu, \nu) \leq \widehat{\mathcal{GS}}_\Phi(\mu, \nu) \leq \mathcal{GST}_\Phi(\mu, \nu). \tag{124}$$

Thus, from Equations (123), and (124), we obtain

$$\frac{1}{2} \mathcal{W}_1(\mu, \nu) \leq \widehat{\mathcal{GS}}_\Phi(\mu, \nu) \leq \mathcal{W}_1(\mu, \nu). \tag{125}$$

Hence, the proof is complete.

∎

### B.14 PROOF FOR LEMMA A.1

*Proof.* Consider an $N$-function $\Phi$, $f \in WL^1_{\Phi,0}(\mathbb{G}, \lambda)$, a nonnegative weight function $\hat{w}$, and $t > 0$. Then we have

$$\int_{\mathbb{G}} \hat{w}(x) \, \Phi\left(\frac{1}{t} |f(x)|\right) \lambda(\mathrm{d}x) = \int_{\mathbb{G}} \hat{w}(x) \, \Phi\left(\frac{1}{t} \left|\int_{\mathbb{G}} \mathbf{1}_{[z_0, x]}(y) f'(y) \lambda(\mathrm{d}y)\right|\right) \lambda(\mathrm{d}x)$$

$$= \int_{\mathbb{G}} \hat{w}(x) \, \Phi\left(\frac{\lambda(\mathbb{G})}{t} \left|\frac{1}{\lambda(\mathbb{G})} \int_{\mathbb{G}} \mathbf{1}_{[z_0, x]}(y) f'(y) \lambda(\mathrm{d}y)\right|\right) \lambda(\mathrm{d}x). \tag{126}$$

For $\alpha > 0$, let $\bar{\Phi}_\alpha(s) := \Phi(\alpha s)$ for $s > 0$. Then $\bar{\Phi}_\alpha$ is also a convex, non-decreasing function. Using this function, we can rewrite Equation (126) as follows:

$$\int_{\mathbb{G}} \hat{w}(x) \, \Phi\left(\frac{1}{t} |f(x)|\right) \lambda(\mathrm{d}x) = \int_{\mathbb{G}} \hat{w}(x) \, \bar{\Phi}_{\frac{\lambda(\mathbb{G})}{t}} \left(\left|\frac{1}{\lambda(\mathbb{G})} \int_{\mathbb{G}} \mathbf{1}_{[z_0, x]}(y) f'(y) \lambda(\mathrm{d}y)\right|\right) \lambda(\mathrm{d}x).$$

Then, by applying Jensen's inequality, we have

$$\int_{\mathbb{G}} \hat{w}(x) \, \Phi\left(\frac{1}{t} |f(x)|\right) \lambda(\mathrm{d}x) \leq \frac{1}{\lambda(\mathbb{G})} \int_{\mathbb{G}} \int_{\mathbb{G}} \hat{w}(x) \, \bar{\Phi}_{\frac{\lambda(\mathbb{G})}{t}} \left(\left|\mathbf{1}_{[z_0, x]}(y) f'(y)\right|\right) \lambda(\mathrm{d}y) \lambda(\mathrm{d}x)$$

$$= \frac{1}{\lambda(\mathbb{G})} \int_{\mathbb{G}} \int_{\mathbb{G}} \hat{w}(x) \, \Phi\left(\frac{\lambda(\mathbb{G})}{t} \left|\mathbf{1}_{[z_0, x]}(y) f'(y)\right|\right) \lambda(\mathrm{d}y) \lambda(\mathrm{d}x).$$

Due to $\Phi(0) = 0$, we can rewrite the above expression as

$$\int_{\mathbb{G}} \hat{w}(x) \, \Phi\left(\frac{1}{t} |f(x)|\right) \lambda(\mathrm{d}x) \leq \frac{1}{\lambda(\mathbb{G})} \int_{\mathbb{G}} \hat{w}(x) \left(\int_{[z_0, x]} \Phi\left(\frac{\lambda(\mathbb{G})}{t} |f'(y)|\right) \lambda(\mathrm{d}y)\right) \lambda(\mathrm{d}x)$$

$$= \frac{1}{\lambda(\mathbb{G})} \int_{\mathbb{G}} \hat{w}(x) \left(\int_{\mathbb{G}} \mathbf{1}_{[z_0, x]}(y) \, \Phi\left(\frac{\lambda(\mathbb{G})}{t} |f'(y)|\right) \lambda(\mathrm{d}y)\right) \lambda(\mathrm{d}x).$$

Applying Fubini's theorem, we can interchange the order of the integrations. As a consequence, we obtain

$$\int_{\mathbb{G}} \hat{w}(x)\,\Phi\left(\frac{1}{t}\,|f(x)|\right)\lambda(\mathrm{d}x) \le \frac{1}{\lambda(\mathbb{G})}\int_{\mathbb{G}}\left(\int_{\mathbb{G}}\hat{w}(x)\mathbf{1}_{[z_0,x]}(y)\lambda(\mathrm{d}x)\right)\Phi\left(\frac{\lambda(\mathbb{G})}{t}\,|f'(y)|\right)\lambda(\mathrm{d}y)$$

$$= \frac{1}{\lambda(\mathbb{G})}\int_{\mathbb{G}}\Phi\left(\frac{\lambda(\mathbb{G})}{t}\,|f'(y)|\right)\bar{\lambda}_{\hat{w}}(\Lambda(y))\lambda(\mathrm{d}y).$$

Hence, the proof is complete. ∎

### B.15 Proof for Lemma A.2

*Proof.* The conclusion of the lemma is trivial if $\|f\|_{L_\Phi} = 0$. Thus we only need to consider the case $\|f\|_{L_\Phi} > 0$ in the following proof. As a consequence and due to $f(z_0) = 0$, there must be $x \in \mathbb{G}$ such that $|f'(x)| > 0$, i.e. the set $\{x \in \mathbb{G} : |f'(x)| > 0\}$ is nonempty.

We will prove the result by contradiction. Specifically, suppose by contradiction that $0 < t < \frac{\|f\|_{L_\Phi}}{\lambda(\mathbb{G})}$. Then as $|f'(x)| = 0$ implies $\Phi\left(\frac{|f'(x)|}{t}\right) = 0$ and as $\Phi$ is strictly increasing, we have

$$\int_{\mathbb{G}} w_0(x)\,\Phi\left(\frac{|f'(x)|}{t}\right)\lambda(\mathrm{d}x) \quad = \int_{\mathbb{G},|f'|>0} w_0(x)\,\Phi\left(\frac{|f'(x)|}{t}\right)\lambda(\mathrm{d}x)$$

$$> \int_{\mathbb{G},|f'|>0} w_0(x)\,\Phi\left(\frac{\lambda(\mathbb{G})}{\|f\|_{L_\Phi}}\,|f'(x)|\right)\lambda(\mathrm{d}x)$$

$$= \int_{\mathbb{G}} w_0(x)\,\Phi\left(\frac{\lambda(\mathbb{G})}{\|f\|_{L_\Phi}}\,|f'(x)|\right)\lambda(\mathrm{d}x). \tag{127}$$

On the other hand, since

$$\|f\|_{L_\Phi} = \inf\left\{t > 0 \mid \int_{\mathbb{G}}\Phi\left(\frac{|f(x)|}{t}\right)\lambda(\mathrm{d}x) \le 1\right\}$$

we deduce that

$$\int_{\mathbb{G}}\Phi\left(\frac{|f(x)|}{\|f\|_{L_\Phi}}\right)\lambda(\mathrm{d}x) = 1.$$

Applying Lemma A.1 for $\hat{w} = 1$ and $t = \|f\|_{L_\Phi}$ to the above left hand side, we have

$$1 = \int_{\mathbb{G}}\Phi\left(\frac{|f(x)|}{\|f\|_{L_\Phi}}\right)\lambda(\mathrm{d}x) \quad \le \frac{1}{\lambda(\mathbb{G})}\int_{\mathbb{G}}\Phi\left(\frac{\lambda(\mathbb{G})}{\|f\|_{L_\Phi}}\,|f'(y)|\right)\lambda(\Lambda(y))\lambda(\mathrm{d}y)$$

$$\le \frac{1}{\lambda(\mathbb{G})}\int_{\mathbb{G}}\Phi\left(\frac{\lambda(\mathbb{G})}{\|f\|_{L_\Phi}}\,|f'(y)|\right)w_0(y)\lambda(\mathrm{d}y).$$

Thus, we obtain

$$\int_{\mathbb{G}}\Phi\left(\frac{\lambda(\mathbb{G})}{\|f\|_{L_\Phi}}\,|f'(x)|\right)w_0(x)\lambda(\mathrm{d}y) \ge \lambda(\mathbb{G}).$$

This together with the inequality in (127) yields

$$\int_{\mathbb{G}} w_0(x)\,\Phi\left(\frac{|f'(x)|}{t}\right)\lambda(\mathrm{d}x) > \lambda(\mathbb{G}),$$

which contradicts the assumption. Hence, the proof is complete. ∎

### B.16 PROOF FOR THEOREM A.3

*Proof.* We consider the Musielak norm with weight function $\frac{w_0}{\lambda(\mathbb{G})}$ for gradient function $f'$

$$\|f'\|_{L_\Phi^{w_0/\lambda(\mathbb{G})}} = \inf\left\{t > 0 \mid \int_\mathbb{G} \frac{w_0(x)}{\lambda(\mathbb{G})}\,\Phi\left(\frac{|f'(x)|}{t}\right)\lambda(\mathrm{d}x) \leq 1\right\}.$$

In particular,

$$\int_\mathbb{G} w_0(x)\,\Phi\left(\frac{|f'(x)|}{\|f'\|_{L_\Phi^{w_0/\lambda(\mathbb{G})}}}\right)\lambda(\mathrm{d}x) \leq \lambda(\mathbb{G}).$$

Therefore, by applying Lemma A.2 with $t := \|f\|_{L_\Phi^{w_0/\lambda(\mathbb{G})}}$, we conclude that

$$\|f\|_{L_\Phi} \leq \lambda(\mathbb{G})\,\|f'\|_{L_\Phi^{w_0/\lambda(\mathbb{G})}}.$$

Hence, the proof is complete. ∎

### B.17 PROOF FOR LEMMA A.4

*Proof.* For any $N$-function $\Phi$, any Borel measurable function $f$ on $\mathbb{G}$, and scalar $t > 0$, we have

$$\int_\mathbb{G} w_1(x)\,\Phi\left(\frac{|f(x)|}{t}\right)\lambda(\mathrm{d}x) \geq \int_\mathbb{G} w_2(x)\,\Phi\left(\frac{|f(x)|}{t}\right)\lambda(\mathrm{d}x).$$

Therefore, we obtain

$$\left\{t > 0 \mid \int_\mathbb{G} w_1(x)\,\Phi\left(\frac{|f(x)|}{t}\right)\lambda(\mathrm{d}x) \leq 1\right\} \subseteq \left\{t > 0 \mid \int_\mathbb{G} w_2(x)\,\Phi\left(\frac{|f(x)|}{t}\right)\lambda(\mathrm{d}x) \leq 1\right\}.$$

Notice that the infimum of a set is smaller than or equal to the infimum of its subset. As a consequence, we obtain

$$\|f\|_{L_\Phi^{w_1}} \geq \|f\|_{L_\Phi^{w_2}}.$$

The proof is complete. ∎

### B.18 PROOF FOR LEMMA A.5

*Proof.* Notice that $\Phi$ is a convex function and $\Phi(0) = 0$. Therefore, for any $0 < s \leq t$, we have

$$\begin{aligned}
\Phi(s) &= \Phi\left(\frac{s}{t}t + \left(1 - \frac{s}{t}\right)0\right)\\
&\leq \frac{s}{t}\Phi(t) + \left(1 - \frac{s}{t}\right)\Phi(0) = \frac{s}{t}\Phi(t),
\end{aligned}$$

which yields $\frac{\Phi(s)}{s} \leq \frac{\Phi(t)}{t}$. Thus, the function $t \mapsto \frac{\Phi(t)}{t}$ is nondecreasing on $(0, +\infty)$.

Consequently, since $k \geq 1$, we get

$$\frac{\Phi(t)}{t} \leq \frac{\Phi(kt)}{kt}.$$

That is, $k\Phi(t) \leq \Phi(kt)$. The proof is complete. ∎

### B.19 PROOF FOR LEMMA A.6

Following the definition of Musielak norm (Equation (3)), we have

$$\begin{aligned}
\|kf\|_{L_\Phi^{\hat{w}}} &= \inf\left\{t > 0 \mid \int_\mathbb{G} \hat{w}(x)\,\Phi\left(\frac{|kf(x)|}{t}\right)\lambda(\mathrm{d}x) \leq 1\right\} && (128)\\
&= k\inf\left\{\frac{t}{k} > 0 \mid \int_\mathbb{G} \hat{w}(x)\,\Phi\left(\frac{|f(x)|}{t/k}\right)\lambda(\mathrm{d}x) \leq 1\right\} && (129)\\
&= k\|f\|_{L_\Phi^{\hat{w}}}. && (130)
\end{aligned}$$

The proof is complete.

## B.20 PROOF FOR PROPOSITION A.7

*Proof.* Following (Le et al., 2024, Remark A.1), for $\Phi(t) = \frac{(p-1)^{p-1}}{p^p} t^p$ with $1 < p < \infty$, we have its complementary $N$-function $\Psi(t) = t^q$ where $q$ is the conjugate of $p$, i.e., $1/q + 1/p = 1$. Consequently, we have $L_\Psi = L^q$, and $L_\Psi^{\hat{w}} = L_{\hat{w}}^q$, where $L_{\hat{w}}^q$ is the weighted $L^q$ space with weight function $\hat{w}$.

Additionally, let $\hat{f}(x) = \frac{\mu(\Lambda(x)) - \nu(\Lambda(x))}{\hat{w}(x)}$ for $x \in \mathbb{G}$, by following Equation (55) in the proof of Theorem 3.5 in §B.2, we have

$$\widehat{\mathcal{GS}}_\Phi(\mu, \nu) = \sup_{g \in L_\Psi(\mathbb{G}, \lambda): \|g\|_{L_\Psi^{\hat{w}}} \leq 1} \left| \int_\mathbb{G} \hat{w}(x) \hat{f}(x) g(x) \lambda(\mathrm{d}x) \right| \tag{131}$$

$$= \sup_{g \in L^q(\mathbb{G}, \lambda): \|g\|_{L_{\hat{w}}^q} \leq 1} \left| \int_\mathbb{G} \hat{w}(x) \hat{f}(x) g(x) \lambda(\mathrm{d}x) \right| \tag{132}$$

$$= \left\| \hat{f} \right\|_{L_{\hat{w}}^p} \tag{133}$$

$$= \left[ \int_\mathbb{G} \hat{w}(x) \left| \hat{f}(x) \right|^p \lambda(\mathrm{d}x) \right]^{\frac{1}{p}} \tag{134}$$

$$= \left[ \int_\mathbb{G} \hat{w}(x)^{1-p} \left| \mu(\Lambda(x)) - \nu(\Lambda(x)) \right|^p \lambda(\mathrm{d}x) \right]^{\frac{1}{p}}. \tag{135}$$

Hence, the proof is complete.

∎

## B.21 COMPUTE $A_1$ FOR $\mathcal{A}_{\Phi_1}$ IN $\widehat{\mathcal{GS}}_{\Phi_1}$ (§A.3)

For $a > 0, b > 0$, we consider the term

$$A_1 := \beta \int_0^1 (at + b) \Phi_1 \left( \frac{\alpha}{at+b} \right) \mathrm{d}t. \tag{136}$$

Set $u = at + b$, so $\mathrm{d}u = a\mathrm{d}t$ and $u \in [b, a+b]$, then we can rewrite

$$A_1 = \frac{\beta}{a} \int_b^{a+b} [u \exp(\alpha/u) - \alpha - u] \, \mathrm{d}u. \tag{137}$$

Next, we want to compute

$$B_{\Phi_1} = \int_b^{a+b} u \exp(\alpha/u) \mathrm{d}u. \tag{138}$$

First, we use the substitution by setting $v = \alpha/u$. Then $u = \alpha/v$ and $\mathrm{d}u = -\alpha v^{-2}\mathrm{d}v$.

The integrand becomes

$$\int u \exp(\alpha/u) \mathrm{d}u = \int \frac{\alpha}{v} \exp(v)[-\alpha v^{-2}\mathrm{d}v] = -\alpha^2 \int \exp(v) v^{-3} \mathrm{d}v. \tag{139}$$

Second, we use two integrations by parts to compute:

$$B_1 := \int \exp(v) v^{-3} \mathrm{d}v. \tag{140}$$

Applying the integration by part, we get

$$B_1 = \frac{1}{2} \int \exp(v) v^{-2} \mathrm{d}v - \frac{1}{2} \exp(v) v^{-2}. \tag{141}$$

Applying the integration by part for the integral in the right hand side, we obtain

$$\int \exp(v)v^{-2}\mathrm{d}v = \int \exp(v)v^{-1}\mathrm{d}v - \exp(v)v^{-1}. \tag{142}$$

Then, from Equations (140), (141), and (142), we obtain

$$B_1 = \frac{1}{2}\left(Ei(v) - \exp(v)v^{-1}\right) - \frac{1}{2}\exp(v)v^{-2} + C \tag{143}$$

$$= \frac{1}{2}Ei(v) - \frac{1}{2}\exp(v)v^{-1} - \frac{1}{2}\exp(v)v^{-2} + C. \tag{144}$$

Thus, from Equations (171), and (143), and recall that $v = \alpha/u$, we have

$$\int u\exp(\alpha/u)\mathrm{d}u = -\alpha^2 B_1 \tag{145}$$

$$= -\frac{\alpha^2}{2}Ei(\alpha/u) + \frac{\alpha^2}{2}\frac{\exp(\alpha/u)}{\alpha/u} + \frac{\alpha^2}{2}\frac{\exp(\alpha/u)}{(\alpha/u)^2} + C \tag{146}$$

$$= \frac{1}{2}\left[u^2\exp(\alpha/u) + \alpha u\exp(\alpha/u) - \alpha^2 Ei(\alpha/u)\right] + C. \tag{147}$$

Thus, an antiderivative for the integrand of $A_1$ is

$$F_{A_1} := \frac{\beta}{a}\int\left[u\exp(\alpha/u) - \alpha - u\right]\mathrm{d}u \tag{148}$$

$$= \frac{\beta}{2a}\left(u^2\exp(\alpha/u) + \alpha u\exp(\alpha/u) - \alpha^2 Ei(\alpha/u) - 2\alpha u - u^2\right) + C. \tag{149}$$

Hence, we have

$$A_1 = F_{A_1}(a+b) - F_{A_1}(b). \tag{150}$$

Or, we obtain

$$A_1 = \frac{\beta}{2a}\left[-\alpha^2\left[Ei\left(\frac{\alpha}{a+b}\right) - Ei\left(\frac{\alpha}{b}\right)\right] + \alpha\left[(a+b)\exp\left(\frac{\alpha}{a+b}\right) - b\exp\left(\frac{\alpha}{b}\right)\right]\right.$$

$$\left. + \left[(a+b)^2\exp\left(\frac{\alpha}{a+b}\right) - b^2\exp\left(\frac{\alpha}{b}\right)\right] - 2\alpha a - \left[(a+b)^2 - b^2\right]\right]. \tag{151}$$

Therefore, we have

$$\mathcal{A}_{\Phi_1} = A_1, \tag{152}$$

where $\beta = w_e, a = \frac{w_e}{\lambda(\mathbb{G})}, b = 1 + \frac{\lambda(\gamma_e)}{\lambda(\mathbb{G})}, \alpha = k|\bar{h}(e)|$.

Additionally, we next compute the derivative of $A_1$ w.r.t. $k$.

For simplicity, let

$$S_1(\alpha) := -\alpha^2\left[Ei\left(\frac{\alpha}{a+b}\right) - Ei\left(\frac{\alpha}{b}\right)\right] + \alpha\left[(a+b)\exp\left(\frac{\alpha}{a+b}\right) - b\exp\left(\frac{\alpha}{b}\right)\right]$$

$$+ \left[(a+b)^2\exp\left(\frac{\alpha}{a+b}\right) - b^2\exp\left(\frac{\alpha}{b}\right)\right] - 2\alpha a - \left[(a+b)^2 - b^2\right], \tag{153}$$

then we have

$$A_1 = \frac{\beta}{2a}S_1(\alpha). \tag{154}$$

Thus, we have

$$\frac{\mathrm{d}A_1}{\mathrm{d}k} = \frac{\mathrm{d}A_1}{\mathrm{d}\alpha}\frac{\mathrm{d}\alpha}{\mathrm{d}k}. \tag{155}$$

Notice that $\frac{\mathrm{d}}{\mathrm{d}z} Ei(z) = \frac{\exp(z)}{z}$, and use the chain rule, we have

For the first term of $S_1$, let $S_{11}(\alpha) := -\alpha^2 \left[ Ei\left(\frac{\alpha}{a+b}\right) - Ei\left(\frac{\alpha}{b}\right) \right]$, we have

$$\frac{\mathrm{d}S_{11}}{\mathrm{d}\alpha} = -2\alpha \left[ Ei\left(\frac{\alpha}{a+b}\right) - Ei\left(\frac{\alpha}{b}\right) \right] - \alpha \left[ \exp(\alpha/(a+b)) - \exp(\alpha/b) \right]. \tag{156}$$

For second term of $S_1$, let $S_{12}(\alpha) := \alpha \left[ (a+b)\exp\left(\frac{\alpha}{a+b}\right) - b\exp\left(\frac{\alpha}{b}\right) \right]$, we have

$$\frac{\mathrm{d}S_{12}}{\mathrm{d}\alpha} = (a+b)\exp\left(\frac{\alpha}{a+b}\right) - b\exp\left(\frac{\alpha}{b}\right) + \alpha \left[ \exp(\alpha/(a+b)) - \exp(\alpha/b) \right]. \tag{157}$$

For the third term of $S_1$, denote $S_{13}(\alpha) := (a+b)^2 \exp\left(\frac{\alpha}{a+b}\right) - b^2 \exp\left(\frac{\alpha}{b}\right)$, we have

$$\frac{\mathrm{d}S_{13}}{\mathrm{d}\alpha} = (a+b)\exp\left(\frac{\alpha}{a+b}\right) - b\exp\left(\frac{\alpha}{b}\right) \tag{158}$$

For the fourth term of $S_1$, we have $\frac{\mathrm{d}}{\mathrm{d}\alpha}(-2\alpha a) = -2a$.

The last term of $S_1$ is constant w.r.t. $\alpha$.

Therefore, we obtain

$$\frac{\mathrm{d}}{\mathrm{d}k} A_1 = \frac{\beta|\bar{h}(e)|}{a} \left\{ -\alpha \left[ Ei\left(\frac{\alpha}{a+b}\right) - Ei\left(\frac{\alpha}{b}\right) \right] + (a+b)\exp\left(\frac{\alpha}{a+b}\right) - b\exp\left(\frac{\alpha}{b}\right) - a \right\}. \tag{159}$$

Furthermore, we next compute the second-order derivative of $A_1$ w.r.t. $k$.

$$\frac{\mathrm{d}^2}{\mathrm{d}k^2} A_1 = \frac{\beta|\bar{h}(e)|}{a} \frac{\mathrm{d}}{\mathrm{d}k} Q_1(\alpha), \tag{160}$$

where we denote

$$Q_1(\alpha) := -\alpha \left[ Ei\left(\frac{\alpha}{a+b}\right) - Ei\left(\frac{\alpha}{b}\right) \right] + (a+b)\exp\left(\frac{\alpha}{a+b}\right) - b\exp\left(\frac{\alpha}{b}\right) - a. \tag{161}$$

By using the chain rule, we have

$$\frac{\mathrm{d}^2}{\mathrm{d}k^2} A_1 = \frac{\beta|\bar{h}(e)|^2}{a} \frac{\mathrm{d}}{\mathrm{d}\alpha} Q_1(\alpha), \tag{162}$$

For the first term of $Q_1$, denote $Q_{11}(\alpha) := -\alpha \left[ Ei\left(\frac{\alpha}{a+b}\right) - Ei\left(\frac{\alpha}{b}\right) \right]$, we have

$$\frac{\mathrm{d}Q_{11}}{\mathrm{d}\alpha} = -\left[ Ei\left(\frac{\alpha}{a+b}\right) - Ei\left(\frac{\alpha}{b}\right) \right] - \alpha \left[ \frac{\exp(\alpha/(a+b))}{\alpha} - \frac{\exp(\alpha/b)}{\alpha} \right] \tag{163}$$

$$= -\left[ Ei\left(\frac{\alpha}{a+b}\right) - Ei\left(\frac{\alpha}{b}\right) \right] - \left[ \exp(\alpha/(a+b)) - \exp(\alpha/b) \right]. \tag{164}$$

For the second term of $Q_1$, we have

$$\frac{\mathrm{d}}{\mathrm{d}\alpha} \left[ (a+b)\exp\left(\frac{\alpha}{a+b}\right) \right] = \exp\left(\frac{\alpha}{a+b}\right). \tag{165}$$

For the third term of $Q_1$, we have

$$\frac{\mathrm{d}}{\mathrm{d}\alpha} \left[ -b\exp(\alpha/b) \right] = -\exp(\alpha/b). \tag{166}$$

The last term of $Q_1$ is a constant w.r.t. $\alpha$.

Therefore, we obtain

$$\frac{\mathrm{d}^2}{\mathrm{d}k^2} A_1 = -\frac{\beta|\bar{h}(e)|^2}{a} \left[ Ei\left(\frac{\alpha}{a+b}\right) - Ei\left(\frac{\alpha}{b}\right) \right]. \tag{167}$$

Hence, we complete the detailed derivation for $A_1$.

## B.22 COMPUTE $A_2$ FOR $\mathcal{A}_{\Phi_2}$ IN $\widehat{\mathcal{GS}}_{\Phi_2}$ (§A.3)

For $a > 0, b > 0$, we consider

$$A_2 := \beta \int_0^1 (at+b)\,\Phi_2\left(\frac{\alpha}{at+b}\right)\,\mathrm{d}t. \tag{168}$$

Set $u = at + b$, so $\mathrm{d}u = a\mathrm{d}t$ and $u \in [b, a+b]$, then we can rewrite

$$A_2 = \frac{\beta}{a}\int_b^{a+b}\left[u\exp(\alpha^2/u^2) - u\right]\mathrm{d}u. \tag{169}$$

Next, we want to compute

$$B_{\Phi_2} = \int_b^{a+b} u\exp(\alpha^2/u^2)\mathrm{d}u. \tag{170}$$

First, we use the substitution by setting $v = \alpha^2/u^2$, so $v > 0$ on the interval. Then $\mathrm{d}v = -2\alpha^2 u^{-3}\mathrm{d}u$, or $\mathrm{d}u = -\frac{u^3}{2\alpha^2}\mathrm{d}v$.

The integrand becomes

$$\int u\exp(\alpha^2/u^2)\mathrm{d}u = -\int \frac{1}{2\alpha^2}u^4\exp(v)\mathrm{d}v = -\frac{\alpha^2}{2}\int\exp(v)v^{-2}\mathrm{d}v, \tag{171}$$

since we have $u^4 = \alpha^4 v^{-2}$.

Second, following Equation (142), we have

$$B_2 := \int \exp(v)v^{-2}\mathrm{d}v = Ei(v) - \exp(v)v^{-1} + C. \tag{172}$$

Thus, we obtain

$$\int u\exp(\alpha^2/u^2)\mathrm{d}u \quad = -\frac{\alpha^2}{2}\left(Ei(v) - \exp(v)v^{-1}\right) + C \tag{173}$$

$$= \frac{1}{2}\left[u^2\exp(\alpha^2/u^2) - \alpha^2 Ei(\alpha^2/u^2)\right] + C, \tag{174}$$

since $v = \alpha^2/u^2$, and consequently $\exp(v)v^{-1} = (u^2/\alpha^2)\exp(\alpha^2/u^2)$.

Thus, an antiderivative for the integrand of $A_2$ is

$$F_{A_2} \quad := \frac{\beta}{a}\int\left[u\exp(\alpha^2/u^2) - u\right]\mathrm{d}u \tag{175}$$

$$= \frac{\beta}{a}\left(\frac{1}{2}\left[u^2\exp(\alpha^2/u^2) - \alpha^2 Ei(\alpha^2/u^2)\right] - \frac{u^2}{2}\right) + C. \tag{176}$$

Hence, we have

$$A_2 = F_{A_2}(a+b) - F_{A_2}(b). \tag{177}$$

Or, we obtain

$$A_2 = \frac{\beta}{2a}\left[-\alpha^2\left[Ei\left(\frac{\alpha^2}{(a+b)^2}\right) - Ei\left(\frac{\alpha^2}{b^2}\right)\right]\right.$$

$$\left. + \left[(a+b)^2\exp\left(\frac{\alpha^2}{(a+b)^2}\right) - b^2\exp\left(\frac{\alpha^2}{b^2}\right)\right] - \left[(a+b)^2 - b^2\right]\right]. \tag{178}$$

Therefore, we have

$$\mathcal{A}_{\Phi_2} = A_2, \tag{179}$$

where $\beta = w_e, a = \frac{w_e}{\lambda(\mathbb{G})}, b = 1 + \frac{\lambda(\gamma_e)}{\lambda(\mathbb{G})}, \alpha = k|\bar{h}(e)|$.

Additionally, we next compute the derivative of $A_2$ w.r.t. $k$.

For simplicity, let

$$S_2(\alpha) := -\alpha^2 \left[ Ei\left( \frac{\alpha^2}{(a+b)^2} \right) - Ei\left( \frac{\alpha^2}{b^2} \right) \right]$$

$$+ \left[ (a+b)^2 \exp\left( \frac{\alpha^2}{(a+b)^2} \right) - b^2 \exp\left( \frac{\alpha^2}{b^2} \right) \right] - \left[ (a+b)^2 - b^2 \right], \qquad (180)$$

then we have

$$A_2 = \frac{\beta}{2a} S_2(\alpha). \qquad (181)$$

Thus, we have

$$\frac{\mathrm{d}A_2}{\mathrm{d}k} = \frac{\mathrm{d}A_2}{\mathrm{d}\alpha} \frac{\mathrm{d}\alpha}{\mathrm{d}k}. \qquad (182)$$

Notice that $\frac{\mathrm{d}}{\mathrm{d}z} Ei(z) = \frac{\exp(z)}{z}$, and use the chain rule, we have

$$\frac{\mathrm{d}S_2}{\mathrm{d}\alpha} = -2\alpha \left[ Ei\left( \frac{\alpha^2}{(a+b)^2} \right) - Ei\left( \frac{\alpha^2}{b^2} \right) \right] - \alpha^2 \left[ \frac{2}{\alpha} \exp(\alpha^2/(a+b)^2) - \frac{2}{\alpha} \exp(\alpha^2/b^2) \right]$$

$$+ 2\alpha \left[ \exp(\alpha^2/(a+b)^2) - \exp(\alpha^2/b^2) \right]$$

$$= -2\alpha \left[ Ei\left( \frac{\alpha^2}{(a+b)^2} \right) - Ei\left( \frac{\alpha^2}{b^2} \right) \right]. \qquad (183)$$

Therefore, we obtain

$$\frac{\mathrm{d}A_2}{\mathrm{d}k} = -\frac{\beta\alpha|\bar{h}(e)|}{a} \left[ Ei\left( \frac{\alpha^2}{(a+b)^2} \right) - Ei\left( \frac{\alpha^2}{b^2} \right) \right]. \qquad (184)$$

Furthermore, we next compute the second-order derivative of $A_2$ w.r.t. $k$.

$$\frac{\mathrm{d}^2}{\mathrm{d}k^2} A_2 = -\frac{\beta|\bar{h}(e)|}{a} \frac{\mathrm{d}}{\mathrm{d}k} Q_2(\alpha), \qquad (185)$$

where we denote

$$Q_2(\alpha) := \alpha \left[ Ei\left( \frac{\alpha^2}{(a+b)^2} \right) - Ei\left( \frac{\alpha^2}{b^2} \right) \right]. \qquad (186)$$

By using the chain rule, we have

$$\frac{\mathrm{d}^2}{\mathrm{d}k^2} A_2 = -\frac{\beta|\bar{h}(e)|}{a} \left\{ |\bar{h}(e)| \left[ Ei\left( \frac{\alpha^2}{(a+b)^2} \right) - Ei\left( \frac{\alpha^2}{b^2} \right) \right] \right.$$

$$\left. + \alpha \left[ \frac{2}{\alpha} \exp(\alpha^2/(a+b)^2) - \frac{2}{\alpha} \exp(\alpha^2/b^2) \right] |\bar{h}(e)| \right\}. \qquad (187)$$

Therefore, we obtain

$$\frac{\mathrm{d}^2}{\mathrm{d}k^2} A_2 = -\frac{\beta|\bar{h}(e)|^2}{a} \left[ Ei\left( \frac{\alpha^2}{(a+b)^2} \right) - Ei\left( \frac{\alpha^2}{b^2} \right) + 2\exp(\alpha^2/(a+b)^2) - 2\exp(\alpha^2/b^2) \right]. \qquad (188)$$

Hence, we complete the detailed derivation for $A_2$.

## C   FURTHER EXPERIMENTAL DETAILS AND EMPIRICAL RESULTS

### C.1   FURTHER EXPERIMENTAL DETAILS

We summarize the number of pairs of probability measures for each dataset which is required to evaluate for kernelized SVM in Table 1.

**Implementation notes for $\gamma_e$ in GSI-M.**

• **Preprocessing procedure for $\gamma_e$.** Following (Le et al., 2025), we precompute $\gamma_e$ for all edge $e$ in the given graph $\mathbb{G}$. It only needs once for such preprocessing procedure since it does not depend on input measures, but only the graph structure itself. More concretely, we apply the Dijkstra algorithm to recompute the shortest paths from root node $z_0$ to all other input supports (or vertices) with complexity $\mathcal{O}(|E| + |V| \log |V|)$ where we write $|\cdot|$ for the set cardinality. Then, we can evaluate $\gamma_e$ for each edge $e$ in $E$.

• **Sparsity of $\gamma_e$.** As observed in (Le et al., 2025), for any support of input measure $\mu$, its mass is contributed to $\mu(\gamma_e)$ if and only if $e \subseteq [z_0, x]$ (Le et al., 2022). Therefore, let $\mathrm{supp}(\mu)$ be the set of supports of measure $\mu$, and define $E_{\mu,\nu} \subseteq E$ as follows

$$E_{\mu,\nu} := \{e \in E \mid \exists z \in (\mathrm{supp}(\mu) \cup \mathrm{supp}(\nu)), e \subseteq [z_0, z]\}.$$

Additionally, note that $\Phi(0) = 0$ for all $N$-function $\Phi$. Then, in fact, we can remove all edges $e \in E \setminus E_{\mu,\nu}$ in the summation in Equation (14) for the univariate optimization problem for computing GSI-M.

Table 1: The number of pairs of probability measures on datasets for kernelized SVM.

| Datasets | #pairs |
|----------|--------:|
| TWITTER | 4394432 |
| RECIPE | 8687560 |
| CLASSIC | 22890777 |
| AMAZON | 29117200 |
| Orbit | 11373250 |
| MPEG7 | 18130 |

## C.2 FURTHER EMPIRICAL RESULTS

We provide further results, corresponding to the empirical results in §6 for graph $\mathbb{G}_{\mathrm{Sqrt}}$.

**Computational comparison.** We illustrate corresponding results for computational comparison on graph $\mathbb{G}_{\mathrm{Sqrt}}$ with 1K nodes and 32K edges in Figure 4. The computation of GSI-M is also several-order faster than OW, and comparable to GST. More concretely, $\widehat{\mathcal{GS}}$ is $100\times, 6800\times, 2900\times$ faster than OW for $\Phi_0, \Phi_1, \Phi_2$ respectively. For $N$-functions $\Phi_1$ and $\Phi_2$, GSI-M takes less than 23 *seconds* while OW needs at least 19 *hours*, and up to 34 *hours* for the computation.

**Document classification.** Figure 5 illustrates corresponding results for document classification on graph $\mathbb{G}_{\mathrm{Sqrt}}$ with 10K nodes and about 1M edges.

**TDA.** Figure 6 shows corresponding results for TDA on graph $\mathbb{G}_{\mathrm{Sqrt}}$ with 10K nodes and about 1M edges.

# D BRIEF REVIEWS

In this section, to ease the readers, we give a brief review for related notions which we have used in the development of our proposal. For completeness, we also summarize some notions reviewed in (Le et al., 2024; 2025).

## D.1 GRAPH ILLUSTRATION

We illustrate graph notions, reviewed in §2, in Figure 7.

## D.2 A REVIEW ON FUNCTIONAL SPACES

We describe a short review on the $L^p$ space, the weighted $L^p$ space.

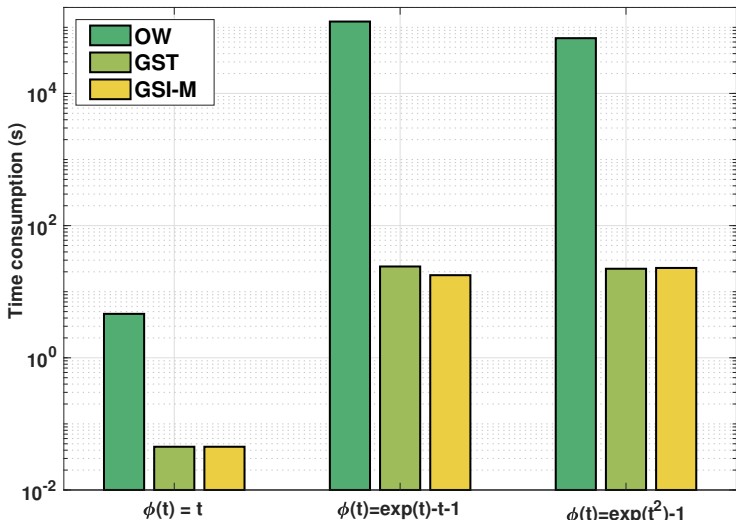

Figure 4: Time consumption for GSI-M, GST and OW on $\mathbb{G}_{\text{Sqrt}}$ with 1K nodes and 32K edges.

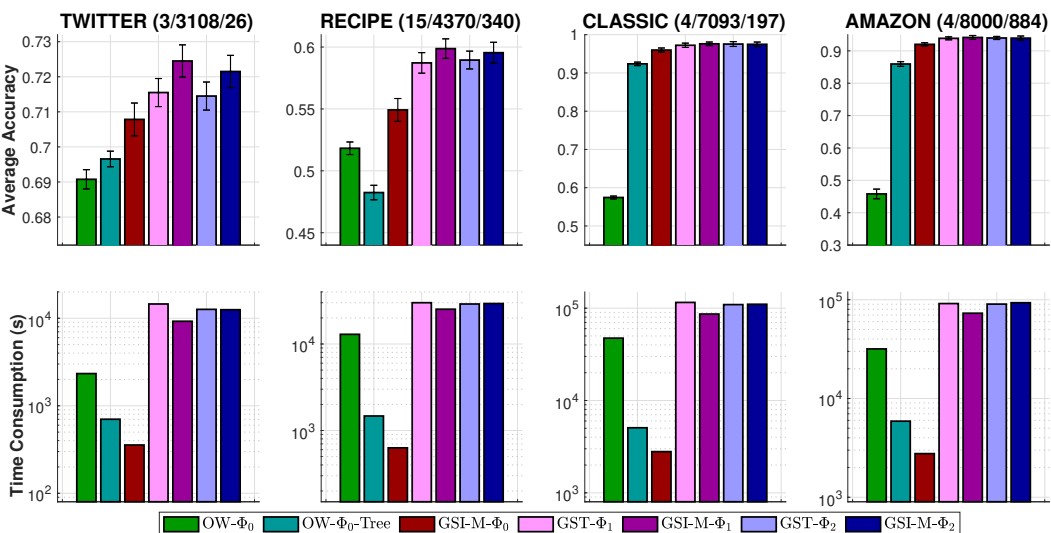

Figure 5: SVM results and time consumption for kernel matrices with graph $\mathbb{G}_{\text{Sqrt}}$.

$L^p$ **space.** For a nonnegative Borel measure $\lambda$ on $\mathbb{G}$, let $L^p(\mathbb{G}, \lambda)$ be the space of all Borel measurable functions $f : \mathbb{G} \to \mathbb{R}$ s.t. $\int_{\mathbb{G}} |f(y)|^p \lambda(\mathrm{d}y) < \infty$. For $p = \infty$, we instead assume that $f$ is bounded $\lambda$-a.e. Then, $L^p(\mathbb{G}, \lambda)$ is a normed space with the norm being defined as follows:

$$\|f\|_{L^p} := \left( \int_{\mathbb{G}} |f(y)|^p \lambda(\mathrm{d}y) \right)^{\frac{1}{p}} \quad \text{for } 1 \le p < \infty.$$

On the other hand, for $p = \infty$, then we have

$$\|f\|_{L^\infty} := \inf \left\{ t \in \mathbb{R} : |f(x)| \le t \text{ for } \lambda\text{-a.e. } x \in \mathbb{G} \right\}.$$

Additionally, functions $f_1, f_2 \in L^p(\mathbb{G}, \lambda)$ are considered to be the same if $f_1(x) = f_2(x)$ for $\lambda$-a.e. $x \in \mathbb{G}$.

$L^p_{\hat{w}}$ **space.** For a nonnegative Borel measure $\lambda$ on $\mathbb{G}$, and a positive weight function $\hat{w}$ on $G$, let $L^p_{\hat{w}}(\mathbb{G}, \lambda)$ be the space of all Borel measurable functions $f : \mathbb{G} \to \mathbb{R}$ s.t. $\int_{\mathbb{G}} \hat{w}(x)|f(x)|^p \lambda(\mathrm{d}x) < \infty$. For $p = \infty$, we instead assume that $f$ is bounded $\hat{w}\lambda$-a.e. Then, $L^p_{\hat{w}}(\mathbb{G}, \lambda)$ is a normed space with

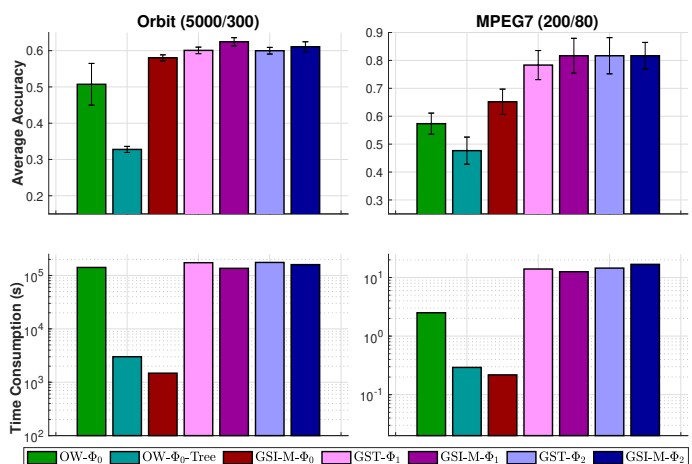

Figure 6: SVM results and time consumption for kernel matrices with graph $\mathbb{G}_{\text{Sqrt}}$.

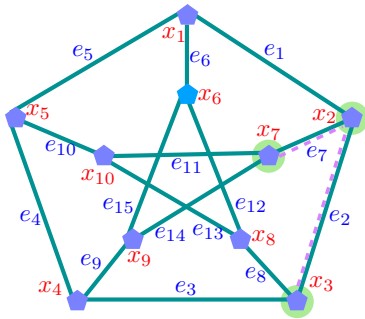

Figure 7: A geodetic graph with 10 nodes $\{x_1, x_2, \ldots, x_{10}\}$ and 15 edges $\{e_1, e_2, \ldots, e_{15}\}$, and each edge length equals to 1, i.e., $w_{e_j} = 1, \forall j$. For any $x_i, x_j$, there is a unique shortest path between them, with a length 2. Therefore, it satisfies the uniqueness property of the shortest paths. Let $x_1$ be the unique-path root node (i.e., $z_0 = x_1$), and subgraph $\widetilde{\mathbb{G}}$ containing 3 nodes $\{x_2, x_3, x_7\}$ and 2 edges $\{e_2, e_7\}$, then we have $\Lambda(x_2) = \gamma(e_1) = \widetilde{\mathbb{G}}$.

the norm being defined as follows:

$$\|f\|_{L_{\hat{w}}^p} := \left( \int_{\mathbb{G}} \hat{w}(x)|f(x)|^p \lambda(\mathrm{d}x) \right)^{\frac{1}{p}} \text{ for } 1 \le p < \infty.$$

For the case $p = \infty$, as $\hat{w}(x) > 0$ for every $x \in \mathbb{G}$, we have

$$\|f\|_{L_{\hat{w}}^\infty} := \inf \{t \in \mathbb{R} : |f(x)| \le t \text{ for } (\hat{w}\lambda)\text{-a.e. } x \in \mathbb{G}\}$$
$$= \inf \{t \in \mathbb{R} : |f(x)| \le t \text{ for } \lambda\text{-a.e. } x \in \mathbb{G}\}$$
$$= \|f\|_{L^\infty}.$$

### D.3 A REVIEW ON SOBOLEV TRANSPORT

In this section, we provide a brief review on the Sobolev transport (ST) (Le et al., 2022) for graph-based measures, and the length measure on a graph.

**Definition D.1** (Graph-based Sobolev space (Le et al., 2022))**.** Let $\lambda$ be a nonnegative Borel measure on $\mathbb{G}$, and let $1 \le p \le \infty$. A continuous function $f : \mathbb{G} \to \mathbb{R}$ is said to belong to the Sobolev space $W^{1,p}(\mathbb{G}, \lambda)$ if there exists a function $h \in L^p(\mathbb{G}, \lambda)$ satisfying

$$f(x) - f(z_0) = \int_{[z_0, x]} h(y) \, \lambda(\mathrm{d}y) \quad \forall x \in \mathbb{G}. \tag{189}$$

Such function $h$ is unique in $L^p(\mathbb{G}, \lambda)$ and is called the graph derivative of $f$ w.r.t. the measure $\lambda$. The graph derivative of $f \in W^{1,p}(\mathbb{G}, \lambda)$ is denoted as $f' \in L^p(\mathbb{G}, \lambda)$.

**Sobolev transport (ST) (Le et al., 2022).** For probability measures $\mu, \nu \in \mathcal{P}(\mathbb{G})$, and $1 \le p \le \infty$, the $p$-order Sobolev transport (ST) (Le et al., 2022, Definition 3.2) is defined as

$$\mathcal{ST}_p(\mu, \nu) := \begin{cases} \sup \left[ \int_{\mathbb{G}} f(x)\mu(\mathrm{d}x) - \int_{\mathbb{G}} f(x)\nu(\mathrm{d}x) \right] \\ \text{s.t. } f \in W^{1,p'}(\mathbb{G}, \lambda),\ \|f'\|_{L^{p'}(\mathbb{G}, \lambda)} \le 1, \end{cases} \tag{190}$$

where we write $f'$ for the generalized graph derivative of $f$, and $W^{1,p'}(\mathbb{G}, \lambda)$ for the graph-based Sobolev space on $\mathbb{G}$.

ST is a scalable variant of optimal transport (OT) for probability measures supported on a graph (i.e., not OT itself) since it relaxes the Lipschitz constriant for the critic function as in $1$-order Wasserstein by considering this Lipschitz constraint in a Sobolev space. Additionally, Le et al. (2022, Corollary 4.3) showed that $1$-order Sobolev transport is identical to $1$-order Wasserstein *when the graph is a tree*. Furthermore, Le et al. (2022, §4.1) admitted that the exact relationship between $p$-order Sobolev transport and $p$-order Wasserstein when $p > 1$ is *unknown*.

**Proposition D.2** (Closed-form expression of Sobolev transport (Le et al., 2022))**.** *Let $\lambda$ be any nonnegative Borel measure on $\mathbb{G}$, and let $1 \le p < \infty$. Then, we have*

$$\mathcal{ST}_p(\mu, \nu) = \left( \int_{\mathbb{G}} |\mu(\Lambda(x)) - \nu(\Lambda(x))|^p\, \lambda(\mathrm{d}x) \right)^{\frac{1}{p}},$$

*where $\Lambda(x)$ is the subset of $\mathbb{G}$ defined by Equation* (1)*.*

**Definition D.3** (Length measure (Le et al., 2022))**.** *Let $\lambda^*$ be the unique Borel measure on $\mathbb{G}$ s.t. the restriction of $\lambda^*$ on any edge is the length measure of that edge. That is, $\lambda^*$ satisfies:*

    i) *For any edge $e$ connecting two nodes $u$ and $v$, we have $\lambda^*(\langle x, y\rangle) = (t - s)w_e$ whenever $x = (1-s)u + sv$ and $y = (1-t)u + tv$ for $s, t \in [0, 1)$ with $s \le t$. Here, recall that $\langle x, y\rangle$ is the line segment in $e$ connecting $x$ and $y$.*

    ii) *For any Borel set $F \subset \mathbb{G}$, we have*

$$\lambda^*(F) = \sum_{e \in E} \lambda^*(F \cap e).$$

**Lemma D.4** ($\lambda^*$ is the length measure on graph (Le et al., 2022))**.** *Suppose that $\mathbb{G}$ has no short cuts, namely, any edge $e$ is a shortest path connecting its two end-points. Then, $\lambda^*$ is a length measure in the sense that*

$$\lambda^*([x, y]) = d_{\mathbb{G}}(x, y)$$

*for any shortest path $[x, y]$ connecting $x, y$. Particularly, $\lambda^*$ has no atom in the sense that $\lambda^*(\{x\}) = 0$ for every $x \in \mathbb{G}$.*

## D.4 A REVIEW ON SOBOLEV IPM AND ITS SCALABLE REGULARIZED APPROACH

We give a brief review on the Sobolev norm (Adams & Fournier, 2003), Sobolev IPM, and its scalable regularized approach (Le et al., 2025) for graph-based measures.

**Sobolev norm.** $W^{1,p}(\mathbb{G}, \lambda)$ is a normed space (reviewed in §D.3), with the Sobolev norm (Adams & Fournier, 2003, §3.1) being defined as

$$\|f\|_{W^{1,p}} := \left( \|f\|_{L^p}^p + \|f'\|_{L^p}^p \right)^{\frac{1}{p}}. \tag{191}$$

Additionally, let $W_0^{1,p}(\mathbb{G}, \lambda)$ be the subspace consisting of all functions $f$ in $W^{1,p}(\mathbb{G}, \lambda)$ satisfying $f(z_0) = 0$. Denote $\mathcal{B}_p := \left\{ f \in W_0^{1,p}(\mathbb{G}, \lambda) : \|f\|_{W^{1,p}} \le 1 \right\}$ as the unit ball in the Sobolev space.

**Sobolev IPM.** Sobolev IPM for graph-based measures is an instance of the IPM where its critic function belongs to the graph-based Sobolev space, and is constrained within the unit ball of that space. More concretely, given a nonnegative Borel measure $\lambda$ on $\mathbb{G}$, an exponent $1 \le p \le \infty$ and its conjugate $p'$, the Sobolev IPM between measures $\mu, \nu \in \mathcal{P}(\mathbb{G})$ is defined as follows:

$$\mathcal{S}_p(\mu, \nu) := \sup_{f \in \mathcal{B}_{p'}} \left| \int_{\mathbb{G}} f(x)\mu(\mathrm{d}x) - \int_{\mathbb{G}} f(y)\nu(\mathrm{d}y) \right|. \tag{192}$$

Note that one should distinguish Sobolev IPM problem from Sobolev transport (Le et al., 2022), generalized Sobolev transport (GST) (Le et al., 2024), and Wasserstein problems. In fact, for Sobolev IPM, the critic function is constrained within a unit ball of Sobolev norm, which involves both critic function and its gradient while Sobolev transport, GST and Wasserstein only constraint on gradient of critic function (i.e., Lipschitz constraint).

**Scalable regularized Sobolev IPM (Le et al., 2025).** For weight function $\hat{w}_{\hat{\mathcal{S}}}(x) = 1 + \lambda(\Lambda(x))$, for all $x \in \mathbb{G}$ . Let $\mathcal{B}(p', \hat{w}_{\hat{\mathcal{S}}})$ be defined as follows:

$$\mathcal{B}(p', \hat{w}_{\hat{\mathcal{S}}}) := \left\{ f \in W_0^{1,p'}(\mathbb{G}, \lambda) : \|f'\|_{L_{\hat{w}_{\hat{\mathcal{S}}}}^{p'}} \le 1 \right\}. \tag{193}$$

Then, the regularized Sobolev IPM is defined as

**Definition D.5** (Regularized Sobolev IPM on graph (Le et al., 2025))**.** Let $\lambda$ be a nonnegative Borel measure on $\mathbb{G}$ and $1 \le p \le \infty$. Then for any given probability measures $\mu, \nu \in \mathcal{P}(\mathbb{G})$, the regularized Sobolev IPM is defined as

$$\hat{\mathcal{S}}_p(\mu, \nu) := \sup_{f \in \mathcal{B}(p', \hat{w}_{\hat{\mathcal{S}}})} \left| \int_{\mathbb{G}} f(x)\mu(\mathrm{d}x) - \int_{\mathbb{G}} f(y)\nu(\mathrm{d}y) \right|. \tag{194}$$

Note that it is *unknown* whether the regularization approach in Le et al. (2025) with weight function $\hat{w}_{\hat{\mathcal{S}}}$ can be extended for general cases beyond the $L^p$ geometric structure, e.g., Orlicz geometric structure.

### D.5 A Review on IPM and Wasserstein Distance

We provide a short review on IPM and Wasserstein distance for probability measures.

**IPM.** Integral probability metrics (IPM) for probability measures $\mu, \nu$ are defined as follows:

$$\gamma_{\mathcal{F}}(\mu, \nu) = \sup_{f \in \mathcal{F}} \left| \int_{\lambda} f(x)\mu(\mathrm{d}x) - \int_{\lambda} f(y)\nu(\mathrm{d}y) \right|. \tag{195}$$

**Special case: 1-Wasserstein distance (dual formulation).** The 1-Wasserstein distance is a special case of IPM. In particular, for $\mathcal{F} = \mathcal{F}_W := \{ f : |f(x) - f(y)| \le d_{\mathbb{G}}(x, y) \}$ where $d_{\mathbb{G}}$ is the graph metric on graph $\mathbb{G}$, then IPM is equal to the 1-Wasserstein distance with ground cost $d_{\mathbb{G}}$

$$\mathcal{W}(\mu, \nu) = \sup_{f \in \mathcal{F}_W} \left| \int_{\mathbb{G}} f(x)\mu(\mathrm{d}x) - \int_{\mathbb{G}} f(y)\nu(\mathrm{d}y) \right|. \tag{196}$$

*p*-**Wasserstein distance (primal formulation).** Let $1 \le p < \infty$, for probability measures $\mu$ and $\nu$ on $\mathbb{G}$, then, the $p$-Wasserstein distance is defined as follows:

$$\mathcal{W}_p(\mu, \nu)^p = \inf_{\pi \in \Pi(\mu, \nu)} \int_{\mathbb{G} \times \mathbb{G}} d_{\mathbb{G}}(x, y)^p \pi(\mathrm{d}x, \mathrm{d}y),$$

where $\Pi(\mu, \nu) := \left\{ \pi \in \mathcal{P}(\mathbb{G} \times \mathbb{G}) : \pi_1 = \mu, \pi_2 = \nu \right\}$; $\pi_1, \pi_2$ are the first and second marginals of $\pi$ respectively.

We would like to remark that to our knowledge, there is *no* closed-form expression for optimal transport (OT) problem for probability measures on a graph in general. However, when the graph is a tree, OT admits closed-form expression, a.k.a., tree-Wasserstein (Le et al., 2019).

### D.6 ORLICZ FUNCTIONS

We provide a brief review on Orlicz functions as summarized in (Le et al., 2024) for completeness. For comprehensive studies on Orlicz functions, see (Adams & Fournier, 2003; Rao & Ren, 1991).

**Popular examples of $N$-functions.** Some popular examples for $N$-functions (Adams & Fournier, 2003, §8.2) in the literature are as follows:

1. $\Phi(t) = t^p$ with $1 < p < \infty$.

2. $\Phi(t) = \exp(t) - t - 1$.

3. $\Phi(t) = \exp(t^p) - 1$ with $1 < p < \infty$.

4. $\Phi(t) = (1 + t) \log(1 + t) - t$.

**Complementary function.** For $N$-function $\Phi$, its complementary function $\Psi : \mathbb{R}_+ \to \mathbb{R}_+$ (Adams & Fournier, 2003, §8.3) is the $N$-function, defined as follows:

$$\Psi(t) = \sup \left[ at - \Phi(a) \mid a \geq 0 \right], \quad \text{for } t \geq 0. \tag{197}$$

**Popular examples of complementary pairs of $N$-functions.** Some popular complementary pairs of $N$-functions (Adams & Fournier, 2003, §8.3), (Rao & Ren, 1991, §2.2) are as follows:

1. $\Phi(t) = \frac{t^p}{p}$ and $\Psi(t) = \frac{t^q}{q}$ where $q$ is the conjugate of $p$, i.e., $\frac{1}{p} + \frac{1}{q} = 1$ and $1 < p < \infty$.

2. $\Phi(t) = \exp(t) - t - 1$ and $\Psi(t) = (1 + t) \log(1 + t) - t$.

3. For the $N$-function $\Phi(t) = \exp(t^p) - 1$ with $1 < p < \infty$, its complementary $N$-function yields an explicit for, but not simple (Rao & Ren, 1991, §2.2), see (Le et al., 2024, §A.8) for the details.

**Young inequality.** Let $\Phi, \Psi$ be a pair of complementary $N$-functions, then we have

$$st \leq \Psi(s) + \Phi(t).$$

**Orlicz norm.** Together with the Luxemburg norm, the Orlicz norm (Rao & Ren, 1991, §3.3, Definition 2) is a popular norm for $L_\Phi(\mathbb{G}, \lambda)$ in the literature, defined as

$$\|f\|_\Phi := \sup \left\{ \int_\mathbb{G} |f(x)g(x)| \lambda(\mathrm{d}x) \ \Big| \ \int_\mathbb{G} \Psi(|g(x)|) \lambda(\mathrm{d}x) \leq 1 \right\}, \tag{198}$$

where $\Psi$ is the complementary $N$-function of $\Phi$.

**Computation for Orlicz norm.** Following (Rao & Ren, 1991, §3.3, Theorem 13), the Orlicz norm can be recast as follows:

$$\|f\|_\Phi = \inf_{k > 0} \frac{1}{k} \left( 1 + \int_\mathbb{G} \Phi(k \, |f(x)|) \lambda(\mathrm{d}x) \right).$$

Therefore, one can use any second-order method, e.g., fmincon solver in MATLAB (with trust region reflective algorithm), to solve the *univariate* optimization problem for Orlicz norm computation.

**Equivalence (Adams & Fournier, 2003, §8.17) (Musielak, 2006, §13.11).** The Luxemburg norm is equivalent to the Orlicz norm. In fact, we have

$$\|f\|_{L_\Phi} \leq \|f\|_\Phi \leq 2 \|f\|_{L_\Phi}. \tag{199}$$

**Connection between $L^p$ and $L_\Phi$ functional spaces.** When the convex function $\Phi(t) = t^p$, for $1 < p < \infty$, we have

$$L^p(\mathbb{G}, \lambda) = L_\Phi(\mathbb{G}, \lambda).$$

**Generalized Hölder inequality.** Let $\Phi, \Psi$ be a pair of complementary $N$-functions, then generalized Hölder inequality w.r.t. Luxemburg norm (Adams & Fournier, 2003, §8.11) is as follows:

$$\left| \int_{\mathbb{G}} f(x)g(x)\lambda(dx) \right| \leq 2 \left\| f \right\|_{L_\Phi} \left\| g \right\|_{L_\Psi}. \tag{200}$$

Additionally, we have the generalized Hölder inequality w.r.t. Luxemburg norm and Orlicz norm (Musielak, 2006, §13.13) is as follows:

$$\left| \int_{\mathbb{G}} f(x)g(x)\lambda(dx) \right| \leq \left\| f \right\|_{L_\Phi} \left\| g \right\|_{\Psi}. \tag{201}$$

### D.7 GENERALIZED SOBOLEV TRANSPORT (GST)

We briefly review generalized Sobolev transport (GST) (Le et al., 2024) for graph-based measures.

**Generalized Sobolev transport (GST) (Le et al., 2024).** Let $\Phi$ be an $N$-function and $\lambda$ be a nonnegative Borel measure on $\mathbb{G}$. For probability measures $\mu, \nu$ on a graph $\mathbb{G}$, the generalized Sobolev transport (GST) is defined as follows:

$$\mathcal{GS}_\Phi(\mu, \nu) := \begin{cases} \sup & \left| \int_{\mathbb{G}} f(x)\mu(\mathrm{d}x) - \int_{\mathbb{G}} f(x)\nu(\mathrm{d}x) \right| \\ \text{s.t.} & f \in WL^1_\Psi(\mathbb{G}, \lambda), \left\| f' \right\|_{L_\Psi} \leq 1, \end{cases}$$

where $\Psi$ is the complementary function of $\Phi$ (see Equation (197)).

Note that GST is a scalable variant of Orlicz-Wasserstein (OW) for graph-based probability measures (i.e., not OW itself). Moreover, Le et al. (2024, Proposition 4.6) showed that GST is equal to OW *when the graph is a tree, and $\Phi(t) = t$ (i.e., the limite case of $N$-function).*

## E   FURTHER DISCUSSIONS

For completeness, we recall important discussions on the underlying graph in (Le et al., 2022; 2024; 2025), since they are also applied and/or adapted to the proposed generalized Sobolev IPM with Musielak regularization.

**Measures on a graph (Le et al., 2025).** We reemphasize that in this work we consider the Sobolev IPM problem between *two input probability measures* supported on the *same* graph, which is also explored in (Le et al., 2025). Such measures supported on a graph metric space are also considered in OT problem, explored in (Le et al., 2022; 2024). Our work generalizes Sobolev IPM, and we also derive a novel regularization for the generalized Sobolev IPM, which admits an efficient algorithmic approach (i.e., simply solving a univariate optimization problem for its computation).

The generalized Sobolev IPM with Musielak regularization (GSI-M) is for *input probability measures*, i.e., to compute distance between two probability measures, on the *same* graph. We further distinguish the considered problem to the following related problems:

• **Compute distance between two (different) input graphs.** Petric Maretic et al. (2019); Dong & Sawin (2020) compute OT problem between *two input graphs*, where their goals are to compute distance between such two input graphs. Therefore, they are essentially different to our considered problem which computes distance between *two input probability measures* supported on the *same* graph.

• **Graph kernels between two (different) input graphs.** Graph kernels are functions between *two input graphs* to measure their similarity. See Borgwardt et al. (2020) for a comprehensive review on graph kernels. Obviously, this research direction is different to our considered kernels for SVM which are built upon the GSI-M used for measuring similarity between *two input probability measures* on the *same* graph.

**Path length for points in graph $\mathbb{G}$ (Le et al., 2022).** We can canonically measure a path length connecting any two points $x, y \in \mathbb{G}$ where $x, y$ are not necessary to be nodes in $V$ of graph $\mathbb{G}$.

Consider the edge $e = \langle u, v \rangle$ connecting two nodes $u, v \in V$, for $x, y \in \mathbb{R}^n$ and $x, y \in e$, we have

$$x = (1 - s)u + sv,$$
$$y = (1 - t)u + tv,$$

for some scalars $t, s \in [0, 1]$. Therefore, the length of the path $[x, y]$ along edge $e$ (i.e., the line segment $\langle x, y \rangle$) is equal to $|t - s|w_e$. As a result, the length for an arbitrary path in $\mathbb{G}$ can be similarly defined by breaking down into pieces over edges and summing over their corresponding lengths (Le et al., 2022).

**Extension to measures supported on $\mathbb{G}$.** Similar to the regularized Sobolev IPM (Le et al., 2025), the discrete case of the GSI-M in Equation (14) can be easily extended for measures with finite supports on $\mathbb{G}$ (i.e., supports of the input measures may not be nodes in $V$, but possibly points on edges in $E$) by using the same strategy to measure a path length for support data points in graph $\mathbb{G}$. More precisely, we break down edges containing supports into pieces and sum over their corresponding values instead of the sum over edges.

**The assumption of uniqueness property of the shortest paths on $\mathbb{G}$.** As discussed in (Le et al., 2022; 2024; 2025), note that $w_e \in \mathbb{R}_+$ for any edge $e \in E$ in graph $\mathbb{G}$., it is almost surely that every node in $V$ can be regarded as unique-path root node since with a high probability, lengths of paths connecting any two nodes in graph $\mathbb{G}$ are different.

Additionally, for some special graph, e.g., a grid of nodes, there is *no* unique-path root node for such graph. However, by perturbing each node, and/or perturbing lengths of edges if $\mathbb{G}$ is a non-physical graph, with a small deviation, we can obtain a graph satisfying the unique-path root node assumption.

Besides that, for input probability measures with full supports in graph $\mathbb{G}$, or at least full supports in any cycle in graph $\mathbb{G}$, then it exists a special support data point where there are multiple shortest paths from the root node to it. In this case, we simply choose one fixed shortest path among them for this support data point (or we can add a virtual edge from the root node to this support data point where the edge length is deducted by a small deviation). In many practical applications (e.g., document classification and TDA in our experiments), one can neglect this special case since input probability measures have a finite number of supports.

**The generalized Sobolev IPM with Musielak regularization (GSI-M).** Similar to regularized Sobolev IPM (Le et al., 2025), we assume that the graph metric space is given. The question of adaptively learning an optimal graph metric structure from given data is left for future work for further investigation.

**The graphs $\mathbb{G}_{\textbf{Log}}$ and $\mathbb{G}_{\textbf{Sqrt}}$ (Le et al., 2022).** For an efficient and fast computation, we apply the farthest-point clustering method to cluster supports of measures into at most $M$ clusters.[24] Then, let the set of vertices $V$ be the set of centroids of these clusters, i.e., graph vertices. For edges, in graph $\mathbb{G}_{\text{Log}}$, we randomly choose $(M \log M)$ edges; and $M^{3/2}$ edges for graph $\mathbb{G}_{\text{Sqrt}}$. We further denote the set of those randomly sampled edges as $\tilde{E}$.

For each edge $e$, its corresponding edge length (i.e., edge weight) $w_e$ is computed by the Euclidean distance between the two corresponding nodes of edge $e$. Let $n_c$ be the number of connected components in the graph $\tilde{\mathbb{G}}(V, \tilde{E})$. Then, we randomly add $(n_c - 1)$ more edges between these $n_c$ connected components to construct a connected graph $\mathbb{G}$ from $\tilde{\mathbb{G}}$. Let $E_c$ be the set of these $(n_c - 1)$ added edges and denote set $E = \tilde{E} \cup E_c$, then $\mathbb{G}(V, E)$ is the constructed graph.

**Datasets.** For the datasets in our experiments (i.e., TWITTER, RECIPE, CLASSIC, AMAZON for document datasets, and Orbit, MPEG7 for TDA), one can contact the authors of Sobolev transport (Le et al., 2022) to access to them.

**Computational devices.** We run all of our experiments on commodity hardware.

---

[24]$M$ is the input number of clusters for the clustering method. Consequently, the clustering result has at most $M$ clusters, depending on input data.

**Hyperparamter validation.** We use the same strategy as in (Le et al., 2025). For validation, we further randomly split *the training set* into $70\%/30\%$ for validation-training and validation with $10$ repeats to choose hyper-parameters in experiments.

**Further discussion on hyperparameters.** The performance of the generalized Sobolev IPM with Musielak regularization (GSI-M) basically depends on the choice of the $N$-function $\Phi$, which is much similar to how kernel functions impact performance in kernel-dependent machine learning framework. In our experiments with $N$-functions $\Phi_1$ and $\Phi_2$, performances with $N$-function $\Phi_1$ is slightly more favorable than those with $N$-function $\Phi_2$, except in MPEG7 dataset with graph $\mathbb{G}_{\mathsf{Sqrt}}$.

Determining the optimal $N$-function $\Phi$ for the generalized Sobolev IPM with Musielak regularization (GSI-M) in a given task is an open problem that warrants further investigation. We leave it for future work. As an interim solution, cross-validation can be used to select $N$-function $\Phi$ from a set of candidate functions.

**The number of pairs in training and test for kernelized SVM (Le et al., 2025).** Let $N_{tr}, N_{te}$ be the number of measures for training and test respectively. For the kernelized SVM training, the number of pairs which we need to evaluate distances is $(N_{tr}-1) \times \frac{N_{tr}}{2}$. On the test phase, the number of pairs which we need to evaluate distances is $N_{tr} \times N_{te}$. Therefore, for each run, the number of pairs which we require to evaluate distances for both training and test is totally $N_{tr} \times \left(\frac{N_{tr}-1}{2} + N_{te}\right)$. See Table 1 for the number of pairs we need to evaluate distances for kernelized SVM in experiments.

**Large language models (LLM) for writing.** LLM is only used for word choice to aid the writing.

