# OpenReview forum: "Generalized Sobolev IPM for Graph-Based Measures"
_ICLR.cc/2026/Conference — Submitted to ICLR 2026_

### Official Review · Reviewer_BDM1 · 2025-10-27

**Soundness:** 3
**Presentation:** 2
**Contribution:** 2
**Rating:** 2
**Confidence:** 2

**Summary:**

The paper aims to generalize the Sobolev IPM [1] using the idea of an Orlicz geometric structure, which has previously been used to extend the Wasserstein distance on L_p spaces to the Orlicz–Wasserstein (OW) distance. By assuming an underlying graph structure, the computation of the proposed generalized Sobolev IPM reduces to a simple univariate optimization problem, which is significantly faster than OW.

**Strengths:**

The main computational advantage of the proposed method lies in the assumption that the compared probability measures are supported on the same graph. Under this assumption, the computation of the optimal transport plan, required in Wasserstein or OW distances, can be avoided, similar to the case where a closed-form solution exists for the Wasserstein distance in the one-dimensional setting. This advantage has also been extensively exploited in several prior works [1, 2, 3].

**Weaknesses:**

- At first glance, I thought this paper extends [1], in the sense that the Sobolev IPM in [1] is defined on an L_p geometric structure, while the proposed method generalizes it by replacing the L_p structure with an Orlicz geometric one. However, I found that the method called generalized Sobolev transport (GST, [2]) also employs an Orlicz geometric structure to generalize L_p. Moreover, my understanding is that the proposed method is essentially a weighted version of [2], that is, it introduces weights into the definition of the Orlicz–Sobolev space, resulting in what the authors call the generalized Sobolev IPM with Musielak regularization (GSI-M) in Eq. (11). I cannot see why [1] is cited in the abstract but not [2], since [2] appears to be more closely related.
- regarding the weighting functions, [2] fixes the weights as w(x) = 1+\lambda(\Lmabda(x)), while in this paper, the weights are defined as w(x) = 1+\lambda(\Lmabda(x))/\lambda(G). A question arises: is it possible to use other weight functions, such as user-defined functions. Furthermore, as the proposed method is a weighted variant of GST [2], the paper should include a more careful discussion on the choice and impact of the weighting functions.
- In the experiments, the authors compare OW (without graph structure), GST [2], and the proposed GSI. It appears that GST and GSI achieve comparable performance in terms of both accuracy and running time. In other words, as mentioned above, although the proposed GSI is the weighted version of GST [2], the paper fails to clearly explain what benefits the weighting scheme of GSI .
- Overall, the paper is not written in a way that is accessible to those who are not familiar with this research line. The proposed method seems to be a weighted variant of [2], but the motivation and practical advantages of introducing this specific weighting scheme remain unclear.

[1] Tam et al., Scalable Sobolev IPM for Probability Measures on a Graph, ICML2025.

[2] Tam et al., Generalized Sobolev Transport for Probability Measures on a Graph, ICML2024

[3] Tam et al., Sobolev transport: A scalable metric for probability measures with graph metric, AISTATS2022

**Questions:**

See the weaknesses.

---

> ### Author Response · Authors · 2025-11-13
> **Rebuttal by Authors [1/3]**
>
> Dear Reviewer BDM1,
>
> Thank you for your valuable feedback. Below are the answers for your questions and comments.
>
> ---
>
> **(1) [...]The main computational advantage of the proposed method lies in the assumption that the compared probability measures are supported on the same graph. Under this assumption, the computation of the optimal transport plan, required in Wasserstein or OW distances, can be avoided, similar to the case where a closed-form solution exists for the Wasserstein distance in the one-dimensional setting. This advantage has also been extensively exploited in several prior works [1, 2, 3][...]**
>
> **[...]regarding the weighting functions, [2] fixes the weights as w(x) = 1+\lambda(\Lmabda(x)), while in this paper, the weights are defined as w(x) = 1+\lambda(\Lmabda(x))/\lambda(G). A question arises: is it possible to use other weight functions, such as user-defined functions. Furthermore, as the proposed method is a weighted variant of GST [2], the paper should include a more careful discussion on the choice and impact of the weighting functions[...]**
>
> → We clarify **10** following points which the Reviewer may overlook and/or misunderstand:
>
> + **[1]** To our knowledge, in general, Wasserstein does **NOT** yield a closed-form expression for probability measures supported on the **same graph**. However, **when the graph is a tree**, then Wasserstein admits a closed-form expression for fast computation.
>
> + **[2]** **We are not aware of how the computation of the optimal transport plan, required in Wasserstein or OW distance can be avoided for probability measures supported on the same graph, as the Reviewer mentioned**. It will be very helpful for the discussions if the Reviewer could **kindly share references for such important results**.
>
> + **[3]** To our knowledge, for the mentioned Sobolev transport for probability measures supported on the same graph (Le et al., 2022), it yields a closed-form expression for fast computation. However, Sobolev transport is only a variant of Wasserstein (**not Wasserstein itself**) for probability measures supported on the same graph. Please see Section 3 in (Le et al., 2022) where they mentioned that instead of using the Lipschitz constraint for the critic as in $1$-Wasserstein, they relax it by considering the constraint in a Sobolev space. Additionally, in their Corollary 4.3 (Le et al., 2022), $1$-order Sobolev transport is identical to $1$-Wasserstein when **the graph is a tree**. Furthermore, in Section 4.1, Le et al., (2022) admitted that the exact relationship between $p$-order Sobolev transport and $p$-Wasserstein when $p > 1$ is **unknown**.
>
> + **[4]** To our knowledge, for the mentioned Sobolev IPM for probability measures supported on the same graph (Le et al., 2025), **it is essentially different to the Sobolev transport and Wasserstein**. For Sobolev IPM, the critic function is constrained within a unit ball of Sobolev norm, which **involves both critic function and its gradient** while Sobolev transport and Wasserstein only constraint on **gradient of critic function** (i.e., Lipschitz constraint).
>
> + **[5]** To our knowledge, for the mentioned generalized Sobolev transport (GST) for probability measures supported on the same graph (Le et al., 2024), GST is a scalable variant of OW (**not OW itself**). Moreover, following the Proposition 4.6 in Le et al., (2024), GST is equal to OW when **the graph is a tree**, and ${\Phi}(t) = t$ (the limit case of $N$-function).
>
> + **[6]** To our knowledge, for Sobolev IPM, Le et al. (2025) exploit the **specific properties of $L^p$ geometry structure** (e.g., closed-form expression of the $L^p$-norm within their proof) to establish the equivalence between the Sobolev norm and weighted $L^p$-norm for their weight function. Consequently, such a result may **not** be directly extended to general $N$-function as in our Theorem 3.2. It is also **unknown** whether the result in Le et al. (2025) remains valid for general $N$-functions, beyond their considered $L^p$ geometric structure, or not.
>
> + **[7]** To our knowledge, the main theoretical result in Theorem 3.2 is **novel**. It is in fact nontrivial, and clearly beyond existing results in Le et al. (2025) as well as in the literature. Unlike the $L^p$ geometric structure used in Le et al. (2025), there is **no** closed-form expression for the Orlicz norm (Equation (2) in line 115-116) within the Orlicz-Sobolev norm (Equation (7) in line 152-153) for the generalized Sobolev IPM.
>
> + **[8]** We emphasize that **the equivalence result in Theorem 3.2 holds true for our considered weight function** in Equation (9) in line 181-183 (see the rigorous proof in Appendix B.1 for Theorem 3.2 where it requires several auxiliary results presented in Appendix A.1, in fact, beyond results in Le et al. (2025)). Note that such the equivalence result may **not** still hold true for any given general weight function.
>
> (see the following Official Comment for the remaining answer)

---

> ### Author Response · Authors · 2025-11-13
> **Rebuttal by Authors [2/3]**
>
> (continue the answer for **(1)**)
>
> + **[9]** Exploiting the graph structure may help to derive a tractable computation for the summation over the graph (see Equation (13) in Theorem 3.4 in line 212-215 for the summation over the graph in computation). However, we emphasize that efficient computation for the **original Sobolev IPM and its generalization** for probability measures supported on the same graph are still **open problems**. Our finding results are limited as an efficient **regularization** approach for generalized Sobolev IPM only. Above all, our **novel theoretical results in Theorem 3.2** play the **key** role for the proposed regularization approach for the generalized Sobolev IPM which allows us to develop an efficient algorithmic approach for its computation.
>
> + **[10]** For the connection between our proposed GSI-M and generalized Sobolev transport (Le et al., 2024), please see the Proposition 4.6 in line 296-300, and further discussion in line 333-341.
>
> Please also see our answer for **(1)** for the Reviewer **LiDo**.
>
>
> Finally, we **strongly encourage the Reviewer to carefully revisit Theorem 3.2 and its rigorous proof in Appendix B.1** in reevaluating the main theoretical contribution of our work. **Theorem 3.2 is the cornerstone of our proposed regularization approach**, which leads to an efficient algorithmic framework for generalized Sobolev IPM problem in computation for practical applications.
>
> ---
>
> **(2) [...] At first glance, I thought this paper extends [1], in the sense that the Sobolev IPM in [1] is defined on an L_p geometric structure, while the proposed method generalizes it by replacing the L_p structure with an Orlicz geometric one. However, I found that the method called generalized Sobolev transport (GST, [2]) also employs an Orlicz geometric structure to generalize L_p. Moreover, my understanding is that the proposed method is essentially a weighted version of [2], that is, it introduces weights into the definition of the Orlicz–Sobolev space, resulting in what the authors call the generalized Sobolev IPM with Musielak regularization (GSI-M) in Eq. (11). I cannot see why [1] is cited in the abstract but not [2], since [2] appears to be more closely related.[...]**
>
> → We clarify that we study **Sobolev IPM** for graph-based measures (**NOT the generalized Sobolev transport (GST) problem**) in this work, please see line 11-13, and line 44-46. Note that **our considered Sobolev IPM problem in this work is essentially different to Sobolev transport and GST problem**. In fact, Sobolev IPM constrains the critic function within the Sobolev norm, **involving both the critic function and its gradient**, while Sobolev transport and GST only have constraints on **a gradient of the critic function**. Again, we respectfully emphasize that we do **NOT** study GST problem in this work, nor aim to provide a weighted variant for GST as in your review.
>
> Please also see the answer for your questions and comments in **(1)**.
>
> We reemphasize that our **novel theoretical result in Theorem 3.2** plays the **key** role for the proposed regularization approach for the generalized Sobolev IPM, which allows us to develop an efficient algorithmic approach for its computation. Additionally, we also **theoretically establish the connection between our proposed approach GSI-M and generalized Sobolev transport (GST) (Le et al., 2024)** in the Proposition 4.6 in line 296-300, and provide a further discussion in line 333-341.
>
> We also reemphasize that **the equivalence result in Theorem 3.2 holds true for our considered weight function** in Equation (9) in line 181-183 (see the rigorous proof in Appendix B.1 for Theorem 3.2 where it requires several auxiliary results presented in Appendix A.1, in fact, beyond results in Le et al. (2025)). Note that such the equivalence result may **not** still hold true for any given general weight function.
>
> ---

---

> ### Author Response · Authors · 2025-11-13
> **Rebuttal by Authors [3/3]**
>
> **(3) [...] In the experiments, the authors compare OW (without graph structure), GST [2], and the proposed GSI. It appears that GST and GSI achieve comparable performance in terms of both accuracy and running time. In other words, as mentioned above, although the proposed GSI is the weighted version of GST [2], the paper fails to clearly explain what benefits the weighting scheme of GSI .[...]**
>
> **[...] Overall, the paper is not written in a way that is accessible to those who are not familiar with this research line. The proposed method seems to be a weighted variant of [2], but the motivation and practical advantages of introducing this specific weighting scheme remain unclear.[...]**
>
> → Again, we reemphasize that we study **Sobolev IPM** for graph-based measures (**NOT the generalized Sobolev transport (GST) problem**) in this work, please see line 11-13, and line 44-46. We also clarify that we do **NOT** study GST problem in this work, nor aim to provide a weighted variant for GST as in your review. Above all, we clarify that **our considered Sobolev IPM problem in this work is essentially different to Sobolev transport and GST problem**.
>
> Please also see the answer for your questions and comments in **(1) and (2)**, where we clarify our considered problem.
>
> + Additionally, we **theoretically derive the connection of our proposed GSI-M and GST** (Le et al., 2024) in Proposition 4.6 in line 296-300, and provide a further discussion in line 333-341.
>
> + Furthermore, we note **the important role of our main theoretical result in Theorem 3.2**, which enables our proposed regularization approach for the generalized Sobolev IPM problem, leading to an efficient algorithmic approach for its computation.
>
> + Notably, we also reemphasize that **the equivalence result in Theorem 3.2 holds true for our considered weight function** in Equation (9) in line 181-183 (see the rigorous proof in Appendix B.1 for Theorem 3.2 where it requires several auxiliary results presented in Appendix A.1, in fact, beyond results in Le et al. (2025)). Note that such the equivalence result may **not** still hold true for any given general weight function.
>
> For the empirical results, they illustrate that GSI-M with $N$-function $\Phi_1, \Phi_2$ **compare favorably** to GSI-M with the limit case $\Phi_0$, **especially in RECIPE and MPEG7** (see Figures 2 and 3; as well as Figures 5 and 6 in the Appendix). Additionally, performances of GSI-M **compare favorably** to those of GST with the same $N$-function $\Phi_1, \Phi_2$, **especially in TWITTER and RECIPE** (again, see Figures 2 and 3; as well as Figures 5 and 6 in the Appendix). These results can serve as initial evidence on the advantages of Orlicz geometric structure in GSI-M for practical applications. Additionally, the empirical results also **well support our claimed contributions** in line 70-82 (we believe that there are **no overclaimed contributions** in our work). We agree that it is still an on-going research direction with a great potential for further exploration.
>
> We emphasize that efficient computation for the **original Sobolev IPM and its generalization** are still **open problems**. Our finding results are limited as an efficient **regularization** approach for generalized Sobolev IPM only.
>
> We clarify that there is **no assumption on any prior knowledge on the Sobolev IPM** for readers. Moreover, to ease for the readers, we briefly review related works and notions to our proposals in Appendix D.
>
> We also note that all methods (OW, GST, GSI-M) are evaluated in the same setting, i.e., comparing probability measures supported on the same graph metric space.
>
> ---
>
> **Concluding remarks.** We would be grateful if you could confirm whether our clarifications and responses adequately address your concerns. If they do, we kindly ask that you consider increasing your rating. We are also happy to address any additional questions or concerns you might have.
>
> ---

---

> ### Author Response · Authors · 2025-11-20
> **Any Questions from the Reviewer BDM1 on Our Rebuttal?**
>
> We would like to thank the Reviewer again for your thoughtful review and valuable feedback.
>
> We would appreciate it if you could let us know whether our responses have addressed your concerns, and whether you still have any other questions about our rebuttal.
>
> We would be happy to do any follow-up discussion or address any additional comments.

---

> ### Author Response · Authors · 2025-11-24
> **Kind Reminder: Response of Author Rebuttal for Paper 17917**
>
> Dear Reviewer BDM1,
>
> We sincerely appreciate the time you have taken to provide feedback on our work, which has helped us greatly improve its clarity, among other attributes in the revision.
>
> This is a gentle reminder that our rebuttal has been available, and we have already updated the revision in the Openreview.
>
> We would appreciate it if you could let us know whether our responses and our revision have addressed your concerns, and whether you still have any other questions about our rebuttal and the revision.
>
> We would be more than happy to do any follow-up discussion or address any additional comments.
>
> Sincerely,
>
> The Authors

---

### Official Review · Reviewer_Zzvv · 2025-11-01

**Soundness:** 3
**Presentation:** 3
**Contribution:** 2
**Rating:** 6
**Confidence:** 4

**Summary:**

This paper introduces Generalized Sobolev IPM (GSI) with Musielak regularization for measuring distances between probability measures on graph metric spaces. Building on Sobolev Transport (ST) and Generalized Sobolev Transport (GST), the authors develop theoretical connections between GSI and other transport distances including Orlicz-Wasserstein. They prove metric properties (Theorem 3.4) and establish computational efficiency improvements. The paper demonstrates applications in document classification and TDA, showing competitive performance with faster computation times compared to existing methods.

**Strengths:**

**Theoretical innovation and significance:** The paper makes a significant contribution by generalizing Sobolev IPM through Orlicz geometry, creating meaningful connections between integral probability metrics and transport distances on graphs. The rigorous proofs establishing relationships between GSI-M and GST (Proposition 4.6: $\frac{1}{2} \text{GST}\_{\Phi}(\mu,\nu) \leq \hat{GS}\_{\Phi}(\mu,\nu) \leq \text{GST}_{\Phi}(\mu,\nu)$) provide valuable theoretical insights that advance beyond prior work (Le et al., 2022, 2024).

**Clarity and practical relevance:** The presentation is exceptionally clear, with well-structured theoretical development that makes complex concepts accessible. The discrete case formulation (Theorem 3.5) provides practical computational methods, and the paper effectively demonstrates applications in document classification and TDA where computational efficiency matters, addressing real-world limitations of existing approaches.

**Weaknesses:**

**Limited experimental validation with key baselines:** While the paper cites Fused Gromov-Wasserstein (FGW) and Fused Partial Gromov-Wasserstein (FPGW) (Bai et al., 2025; Brogat-Motte et al., 2022), it lacks direct comparisons with these methods. Given that FGW has become a standard for structured object matching, including these comparisons would significantly strengthen the empirical validation and better position GSI-M within the broader landscape of graph-based distance metrics.

**Narrow experimental scope:** The evaluation focuses primarily on document classification (Orbit) and TDA (MPEG7) datasets but misses opportunities to test on more diverse graph-structured problems. Additional experiments on graph matching tasks or node classification would better demonstrate the versatility of GSI-M across different application domains where graph structure plays a critical role.

**Insufficient parameter sensitivity analysis:** While the paper demonstrates computational advantages, a more thorough ablation study on how the choice of N-function $\Phi$ affects practical performance would help practitioners understand when and why to choose specific configurations. The current experiments don't fully explore how different Φ functions impact results across varying data characteristics.

**Questions:**

1. In Proposition 4.8, you establish $1/2 \text{OW}(\mu,\nu) \leq \hat{GS}_\Phi(\mu,\nu) \leq \text{OW}(\mu,\nu)$ for tree graphs. How does this bound degrade for graphs with cycles, and is there a tight bound based on graph properties like treewidth?

2. Your Equation (14) shows the discrete case formulation. How sensitive is the computational performance to the sparsity pattern of $\gamma_e$, and have you observed cases where non-standard Φ functions provide meaningful advantages over standard Sobolev IPM?

---

> ### Author Response · Authors · 2025-11-13
> **Rebuttal by Authors [1/2]**
>
> Dear Reviewer Zzvv,
>
> Thank you for your valuable feedback. Below are the answers for your questions and comments.
>
> ---
>
> **(1) [...] Limited experimental validation with key baselines: While the paper cites Fused Gromov-Wasserstein (FGW) and Fused Partial Gromov-Wasserstein (FPGW) (Bai et al., 2025; Brogat-Motte et al., 2022), it lacks direct comparisons with these methods. Given that FGW has become a standard for structured object matching, including these comparisons would significantly strengthen the empirical validation and better position GSI-M within the broader landscape of graph-based distance metrics.[...]**
>
> → We clarify that as in line 347-351, we study Sobolev IPM for **two probability measures** supported on a **same graph**, which is also considered in Le et al. (2025). While the Fused Gromov-Wasserstein (FGW) and Fused Partial Gromov-Wasserstein (FPGW) (Bai et al., 2025; Brogat-Motte et al., 2022) are to compute distance/discrepancies between **two input (different) graphs**. Please also see line 2112-2130 in the Appendix for further clarification.
>
> ---
>
> **(2) [...] Narrow experimental scope: The evaluation focuses primarily on document classification (Orbit) and TDA (MPEG7) datasets but misses opportunities to test on more diverse graph-structured problems. Additional experiments on graph matching tasks or node classification would better demonstrate the versatility of GSI-M across different application domains where graph structure plays a critical role.[...]**
>
> → We respectfully emphasize that the generalized Sobolev transport is to compare two probability distributions supported on the **same graph**. We distinguish the considered problem with the research lines on computing either kernels or distances/discrepancies between **two input (different) graphs**, as in line 347-351 as well as in line 212-2130 in the Appendix.
>
> In our experiments, we demonstrate the effectiveness of our approach in document classification on four real-world datasets and topological data analysis (TDA), including orbit recognition for linked twist maps, which is a discrete dynamical system modeling flows in DNA microarrays, and object shape recognition in MPEG7 dataset. These evaluations on document classification and TDA are often used for tasks involving comparing measures on a graph, see Le et al. (2025). We believe that such experimental coverage is rich and diverse enough.
>
> ---
>
> **(3) [...] Insufficient parameter sensitivity analysis: While the paper demonstrates computational advantages, a more thorough ablation study on how the choice of N-function $\Phi$ affects practical performance would help practitioners understand when and why to choose specific configurations. The current experiments don't fully explore how different $\Phi$ functions impact results across varying data characteristics.[...]**
>
> → We clarify that in our experiments, we illustrate the performances for different Orlicz functions $\Phi$, including popular $N$-function $\Phi_1, \Phi_2$ used in the literature, and the limit case $\Phi_0$, please see results in Figures 2 and 3; as well as Figures 5 and 6 in the Appendix.
>
> The performance of the generalized Sobolev IPM with Musielak regularization (GSI-M) basically depends on the choice of $N$-function $\Phi$ (see Definition 3.3 in line 202-208; and Theorem 3.5 in line 223-229), which is much similar to how kernel functions impact performance in kernel-dependent machine learning framework. In our experiments with $N$-functions $\Phi_1, \Phi_2$, performances with $N$-function $\Phi_1$ is slightly more favorable than those with $N$-function $\Phi_2$, except in MPEG dataset with graph $G_{Sqrt}$ (see Figures 2 and 3; as well as Figures 5 and 6 in the Appendix).
>
> We agree with the Reviewer that determining the optimal $N$-function $\Phi$ for the generalized Sobolev IPM with Musielak regularization (GSI-M) in a given task is an important open problem that warrants further investigation. Given that our main focus is on GSI-M, we leave this problem for future work for a careful study. As an interim solution, one may use cross-validation to select $N$-function $\Phi$ from a set of candidate functions.
>
> ---
>
> **(4) [...] In Proposition 4.8, you establish $1/2OW \le \hat{GS} \le OW$ for tree graphs. How does this bound degrade for graphs with cycles, and is there a tight bound based on graph properties like treewidth?[...]**
>
> → We clarify that the result in Proposition 4.8 is **only valid for tree graphs**. To our knowledge, there is no result for the connection between GSI-M and OW for the general graph with cycles yet.
>
> Additionally, the bound in Proposition 4.8 holds true for **any given tree graph**, there is **no** requirement on the treewidth graph.
>
> To our knowledge, this is the first bound established to connect OW and generalized Sobolev IPM with Musielak regularization (GSI-M).
>
> ---

---

> ### Author Response · Authors · 2025-11-13
> **Rebuttal by Authors [2/2]**
>
> **(5) [...] Your Equation (14) shows the discrete case formulation. How sensitive is the computational performance to the sparsity pattern of $\gamma_e$, and have you observed cases where non-standard $\Phi$ functions provide meaningful advantages over standard Sobolev IPM?[...]**
>
> → We clarify that as in line 1791-1798 in the Appendix, as observed in Le et al., (2025), for any support of input measure $\mu$, its mass is contributed to $\mu(\gamma_e)$ if and only if edge $e$ is in the shortest path connecting root node $z_0$ and $x$. Additionally, $\Phi(0) = 0$ for all $N$-function $\Phi$. Therefore, in fact, we can remove all edge in $E \ E_{\mu, \nu}$ (see line 1794-1795 for the definition of $E_{\mu, \nu}$) in the summation in Equation (14)  in line 225-228 for the univariate optimization problem for computing GSI-M.
>
> We emphasize that efficient computation for the **original Sobolev IPM and its generalization** are still **open problems**. Our finding results are limited as an efficient **regularization** approach for generalized Sobolev IPM only.
>
> In our experiments, GSI-M with $N$-function $\Phi_1, \Phi_2$ **compare favorably** to GSI-M with the limit case $\Phi_0$, **especially in RECIPE and MPEG7** (see Figures 2 and 3; as well as Figures 5 and 6 in the Appendix). These results can serve as initial evidence on the advantages of Orlicz geometric structure in GSI-M for practical applications. Additionally, the empirical results also well support our claimed contributions in line 70-82 (we believe that there are no overclaimed contributions in our work). We agree that it is still an on-going research direction with a great potential for further exploration.
>
> ---
>
> **Concluding remarks.** We would be grateful if you could confirm whether our clarifications and responses adequately address your concerns. If they do, we kindly ask that you consider increasing your rating. We are also happy to address any additional questions or concerns you might have.
>
> ---

---

> ### Author Response · Authors · 2025-11-20
> **Any Questions from the Reviewer Zzvv on Our Rebuttal?**
>
> We would like to thank the Reviewer again for your thoughtful review and valuable feedback.
>
> We would appreciate it if you could let us know whether our responses have addressed your concerns, and whether you still have any other questions about our rebuttal.
>
> We would be happy to do any follow-up discussion or address any additional comments.

---

> ### Author Response · Authors · 2025-11-24
> **Kind Reminder: Response of Author Rebuttal for Paper 17917**
>
> Dear Reviewer Zzvv,
>
> We sincerely appreciate the time you have taken to provide feedback on our work, which has helped us greatly improve its clarity, among other attributes in the revision.
>
> This is a gentle reminder that our rebuttal has been available, and we have already updated the revision in the Openreview.
>
> We would appreciate it if you could let us know whether our responses and our revision have addressed your concerns, and whether you still have any other questions about our rebuttal and the revision.
>
> We would be more than happy to do any follow-up discussion or address any additional comments.
>
> Sincerely,
>
> The Authors

---

### Official Review · Reviewer_LiDo · 2025-11-01

**Soundness:** 3
**Presentation:** 2
**Contribution:** 2
**Rating:** 2
**Confidence:** 3

**Summary:**

This paper proposes a generalization of the Sobolev Integral Probability Metric (IPM) for measures on graphs. They extend the standard $L^p$ geometric structure based solution to Orlicz geometric structure using N-functions, termed as Generalized Sobolev IPM (GSI)
Based on an equivalence between Orlicz-Sobolev norm and a weighted Musielak norm, the paper then introduces an efficiently-computable GSI with Musielak regularization.

**Strengths:**

- The extension of Sobolev IPMs from $L^p$ spaces to Orlicz spaces is reasonable.
- The authoors address the computational tractability of this metric.
- The paper shows that the proposed GSI-M is a metric and is equivalent to GSI.
- The empirical results show that GSI-M is computationally more efficient than the related Orlicz-Wasserstein (OW) distance.

**Weaknesses:**

- The novelty of the core technical insight appears largely incremental and derivative. The paper leverages the exact same weight function $\hat{w}(x)$ that was a key finding in Le et al. (2025) to relate the norms. This makes the paper feel like a direct substitution of $L^p$ norms with Orlicz/Musielak norms onto the framework of Le et al. (2025).
- The practical motivation for the generalization is weak. The experiments do not demonstrate a compelling advantage for using the more complex Orlicz functions ($\Phi_1, \Phi_2$) over the limit case ($\Phi_0$). The limit case $\Phi_0$ reduces to the 1-order ST (which has a closed-form) and performs comparably, suggesting the added complexity of the Orlicz structure offers marginal practical benefit for these tasks.
- Section 4, which lists connections to other metrics (ST, GST, OW, OT), reads like an appendix. It provides a long list of propositions without sufficient intuition or discussion of their implications or motivation.
- The paper provides very little background on Sobolev IPMs or their limitations (around Eq. 5), assuming significant prior knowledge. This makes it difficult for the broader community to appreciate the starting point and the motivation for the proposition.
- The paper is not written clearly, and motivation and intuition seems lacking; the thought processes are hard to follow.

**Questions:**

- Could you explicitly clarify the technical novelty compared to Le et al. (2025)? Specifically, is the core contribution the proof that the norm equivalence (Thm 3.2) also holds for Orlicz/Musielak spaces using the same weight function, or is there a more fundamental difference? Would it be easier to reuse prior art in this line of research rather than potentially restating similar results?

- The empirical benefit of the Orlicz generalization (using $\Phi_1, \Phi_2$) over the $L^1$-like $\Phi_0$ case seems minimal. Can you provide evidence of (or hypothesize about) specific applications, graph types, or N-functions where this generalization would provide a clear and significant practical advantage?

- To improve readability, you might consider moving Sec. 4 to the appendix, keeping only the major Theorems. The saved space could then be used to to motivate the problem for a broader audience.

---

> ### Author Response · Authors · 2025-11-13
> **Rebuttal by Authors [1/3]**
>
> Dear Reviewer LiDo,
>
> Thank you for your valuable feedback. Below are the answers for your questions and comments.
>
> ---
>
> **(1) [...]The novelty of the core technical insight appears largely incremental and derivative. The paper leverages the exact same weight function $\hat w(x)$ that was a key finding in Le et al. (2025) to relate the norms[...]**
>
> **[...] Could you explicitly clarify the technical novelty compared to Le et al. (2025)? Specifically, is the core contribution the proof that the norm equivalence (Thm 3.2) also holds for Orlicz/Musielak spaces using the same weight function, or is there a more fundamental difference? Would it be easier to reuse prior art in this line of research rather than potentially restating similar results?[...]**
>
> → We respectfully disagree. We clarify **4** following points which the Reviewer may overlook and/or misunderstand:
>
> + **[1]** Our considered weight function $\hat w(x) := 1 + \frac{\lambda(\Lambda(x))}{\lambda(G)}$ (Equation (9) in line 181-183) is **different** to the weight function used in Le et al. (2025), i.e., $\hat w(x) := 1 + {\lambda(\Lambda(x))}$ in their Equation (5).
>
> + **[2]** To our knowledge, Le et al. (2025) exploit the **specific properties of $L^p$ geometry structure** (e.g., closed-form expression of the $L^p$-norm within their proof) to establish the equivalence between the Sobolev norm and weighted $L^p$-norm for their weight function. Consequently, such a result may **not** be directly extended to general $N$-function as in our Theorem 3.2. It is also **unknown** whether the result in Le et al. (2025) remains valid for general $N$-functions, beyond their considered $L^p$ geometric structure, or not.
>
> + **[3]** To our knowledge, the main theoretical result in Theorem 3.2 is **novel**. It is in fact nontrivial, and clearly beyond existing results in Le et al. (2025) as well as in the literature. Unlike the $L^p$ geometric structure used in Le et al. (2025), there is **no** closed-form expression for the Orlicz norm (Equation (2) in line 115-116) within the Orlicz-Sobolev norm (Equation (7) in line 152-153) for the generalized Sobolev IPM.
>
> + **[4]** We emphasize that the **equivalence result in Theorem 3.2 holds true for our considered weight function** in Equation (9) in line 181-183 (see the rigorous proof in Appendix B.1 for Theorem 3.2 where it requires several auxiliary results presented in Appendix A.1, in fact, beyond results in Le et al. (2025)). Note that such the equivalence result may **not** still hold true for any given general weight function.
>
> Please also see the answer for **(1)** for the Reviewer **BDM1**.
>
> Finally, we **strongly encourage the Reviewer to carefully revisit Theorem 3.2 and its rigorous proof in Appendix B.1** in reevaluating the main theoretical contribution of our work. **Theorem 3.2 is the cornerstone of our proposed regularization approach**, which leads to an efficient algorithmic framework for generalized Sobolev IPM problem in computation for practical applications.
>
> ---

---

> ### Author Response · Authors · 2025-11-13
> **Rebuttal by Authors [2/3]**
>
> **(2) [...] The practical motivation for the generalization is weak. The experiments do not demonstrate a compelling advantage for using the more complex Orlicz functions ($\Phi_1, \Phi_2$) over the limit case ($\Phi_0$). The limit case $\Phi_0$ reduces to the 1-order ST (which has a closed-form) and performs comparably, suggesting the added complexity of the Orlicz structure offers marginal practical benefit for these tasks.[...]**
>
> **[...] The empirical benefit of the Orlicz generalization (using $\Phi_1, \Phi_2$) over the
> $L^1$-like ($\Phi_0$) case seems minimal. Can you provide evidence of (or hypothesize about) specific applications, graph types, or N-functions where this generalization would provide a clear and significant practical advantage?[...]**
>
> → We clarify that the limit case ($\Phi_0$) is **not** an $N$-function due to its linear growth (see footnote 10, line 323). Therefore, it does not incorporate Orlicz geometric structure.
>
> For the advantages of Orlicz geometric structure, it has been exploited in several works, please see line 51-63 (notably, recent finding results in Altschuler and Chewi (2023) and Guha et al. (2023)).
>
> To our knowledge, there is **no efficient algorithmic approach** for generalized Sobolev IPM for practical applications. In this work, **(i)** we generalize Sobolev IPM beyond its coupled $L^p$ geometric structure (by leveraging Orlicz geometric structure). **(ii)** We develop a novel regularization approach (i.e. theoretically ground-based by Theorem 3.2), then derive an efficient algorithmic approach to compute the (regularized) generalized Sobolev IPM for practical applications, even in large-scale settings. Please see line 70-82 for our claimed contribution.
>
> Our empirical results illustrate that GSI-M with $N$-function $\Phi_{1}, \Phi_{2}$ **compare favorably** to GSI-M with the limit case $\Phi_0$, **especially in RECIPE and MPEG7** (see Figures 2 and 3; as well as Figures 5 and 6 in the Appendix). These results can serve as initial evidence on the advantages of Orlicz geometric structure in GSI-M for practical applications. Additionally, the empirical results also **well support our claimed contributions** in line 70-82 (we believe that there are **no overclaimed contributions** in our work). We agree that it is still an on-going research direction with a great potential for further exploration.
>
> We emphasize that efficient computation for the **original Sobolev IPM and its generalization** are still **open problems**. Our finding results are limited as an efficient **regularization** approach for generalized Sobolev IPM only.
>
> ---
>
> **(3) [...] Section 4, which lists connections to other metrics (ST, GST, OW, OT), reads like an appendix. It provides a long list of propositions without sufficient intuition or discussion of their implications or motivation.[...]**
>
> **[...] To improve readability, you might consider moving Sec. 4 to the appendix, keeping only the major Theorems. The saved space could then be used to motivate the problem for a broader audience.[...]**
>
> → We respectfully disagree. In Section 4, we describe the meaning and position of our theoretical results, then rigorously state them mathematically. Additionally, in Section 5, we provide further discussions where **these theoretical results play the key role to position our proposed regularized approach for the generalized Sobolev IPM problem in the big research picture of the literature**. We invite the Reviewer to revisit Section 5 for further discussion, in addition to Section 4.
>
> We emphasize that our finding theoretical results in Section 4 are important, which provides theoretical properties of GSI-M. These finding results are **one of the main contributions** of our work. Therefore, we respectfully think that Section 4 should be in the main manuscript.
>
> ---

---

> ### Author Response · Authors · 2025-11-13
> **Rebuttal by Authors [3/3]**
>
> **(4) [...] The paper provides very little background on Sobolev IPMs or their limitations (around Eq. 5), assuming significant prior knowledge. This makes it difficult for the broader community to appreciate the starting point and the motivation for the proposition.[...]**
>
> **[...] The paper is not written clearly, and motivation and intuition seems lacking; the thought processes are hard to follow.[...]**
>
> → We clarify that in line 44-69, we describe the motivation of our studying Sobolev IPM. We argue that Sobolev IPM is intrinsically coupled with the $L^p$ geometric structure limiting its ability to incorporate alternative structure priors (line 44-50). We invite the Reviewer to revisit Section 1 for the motivation and intuition of our work.
>
> Additionally, in line 123-131, we rigorously review the definition of the Sobolev IPM, which **suffices** to illustrate our argument, i.e., **Sobolev IPM is coupled with $L^p$ geometric structure**.
>
> Thus, we believe there is **no assumption on any prior knowledge on the Sobolev IPM** for readers. Additionally, to ease for the readers, we briefly review related works and notions to our proposals in Appendix D.
>
> ---
>
> **Concluding remarks.** We would be grateful if you could confirm whether our clarifications and responses adequately address your concerns. If they do, we kindly ask that you consider increasing your rating. We are also happy to address any additional questions or concerns you might have.
>
> ---

---

> ### Author Response · Authors · 2025-11-20
> **Any Questions from the Reviewer LiDo on Our Rebuttal?**
>
> We would like to thank the Reviewer again for your thoughtful review and valuable feedback.
>
> We would appreciate it if you could let us know whether our responses have addressed your concerns, and whether you still have any other questions about our rebuttal.
>
> We would be happy to do any follow-up discussion or address any additional comments.

---

> ### Author Response · Authors · 2025-11-24
> **Kind Reminder: Response of Author Rebuttal for Paper 17917**
>
> Dear Reviewer LiDo,
>
> We sincerely appreciate the time you have taken to provide feedback on our work, which has helped us greatly improve its clarity, among other attributes in the revision.
>
> This is a gentle reminder that our rebuttal has been available, and we have already updated the revision in the Openreview.
>
> We would appreciate it if you could let us know whether our responses and our revision have addressed your concerns, and whether you still have any other questions about our rebuttal and the revision.
>
> We would be more than happy to do any follow-up discussion or address any additional comments.
>
> Sincerely,
>
> The Authors

---

### Author Response · Authors · 2025-11-24
**Summary of Revisions**

Dear Area Chairs and the Reviewers,

Following the Reviewers' valuable feedbacks, we have made significant improvements to the manuscript, updated the revised paper on the OpenReview highlighting our changes in violet. We summarize the main changes as follows:

---

+ **(1) The novelty of our theoretical finding results in Theorem 3.2**: we add *Remark 3.3 in line 207-210*, and a paragraph in *line 211-227* to discuss the novelty of the finding results in Theorem 3.2: **(i)** we emphasize the finding result is beyond existing results in Le et al. (2025) as well as the literature; **(ii)** the important role of our weight function in Equation (9) in Theorem 3.2 (such result may **not** exist for any general weight function) [Reviewers **BDM1,  LiDo**]

+ **(2) Our considered problem is the generalized Sobolev IPM (GSI) problem**: **(i)** we distinguish it from Sobolev transport and generalized Sobolev transport problems in *line 51-53*; **(ii)** highlight its computational challenges in *line 74-75*; **(iii)** its coupling challenging with $L^p$ geometric structure in *line 141-145*; **(iv)** clearly specify our brief review for related works and notions in *footnote 7*. [Reviewers **BDM1,  LiDo**]

+ **(3) Empirical results**: **(i)** we clarify that the empirical results well support our claimed contribution in Section 1 in *line 410-411*; **(ii)** we also add more discussions for empirical results: GSI-M with $N$-function $\Phi_1, \Phi_2$ compare favorably than with $\Phi_0$, especially in RECIPE, MPEG7; and GSI-M compares favorably than GST with the same $N$-function, especially in TWITTER and RECIPE [Reviewers **BDM1,  LiDo**]

+ **(4) Relation between the considered generalized Sobolev IPM problem with the generalized Sobolev transport (GST)** We further clarify the relation between our considered GSI problem and GST (Le et al, 2024). Although GSI-M may be regarded as a weighted variant of GST, we emphasize the role of our theoretical finding results in Theorem 3.2 and our weight function in Equation (9). Briefly, Theorem 3.2 is ground-based for our proposed regularization approach, GSI-M, for the considered GSI problem. It is **unknown** whether there is any connection between our considered GSI problem and a weighted variant of GST with any given general weight function [Reviewer **BDM1**]

+ **(5) About Sobolev transport (Le et al., 2022), Sobolev IPM (Le et al., 2025), generalized Sobolev transport (Le et al., 2024)**: **(i)** we add further clarification for the review of Sobolev transport in Appendix D in *line 2009-2014*, especially the relation between Sobolev transport and Wasserstein; **(ii)** we further distinguish the Sobolev IPM problem with Sobolev transport, generalized Sobolev transport, Wasserstein in *line 2059-2063*; **(iii)** we clarify the relation between generalized Sobolev transport and Orlicz-Wasserstein in *line 2077-2080*. [Reviewer **BDM1**]

+ **(6) Roles of theoretical finding results in Section 4**: we add further interpretation for the roles of theoretical finding results in Section 4, and clearly point to its further discussion in Section 5. [Reviewer **LiDo**]

+ **(7) Distinguish our considered GSI problem from kernels/distances/discrepancies for graphs**: we further clarify them in *line 402-403*, and clearly point out its further discussion in *Appendix E* [Reviewer **Zzvv**]

+ **(8) Further discussion on $N$-function hyperparameter**: we further discuss the $N$-function hyperparameter for GSI-M in *line 2272-2280*, in brief, we agree that it is an important open problem, and an interim solution is to use cross-validation to select $N$-function $\Phi$ from a set of candidate functions. [Reviewer **Zzvv**]

---

Again, we thank the Reviewers for constructive critical comments, which helped us to improve the paper.

We would be grateful if you could confirm whether our clarifications and responses adequately address your concerns. We are also happy to address any additional questions or concerns you might have.

---

### Author Response · Authors · 2025-12-02
**Final Remarks for the Rebuttal**

Dear Area Chair and Senior Air Chair,


As the rebuttal period deadline is approaching (without responses yet), we would like to highlight **critical remarks** which we respectfully disagree with the Reviewers’ comments:

---

**[1] Novelty of theoretical finding result in Theorem 3.2** (comparing to existing results in Le et al., 2025)

+ Existing results in Le et al., 2025 for Sobolev IPM are limited for the $L^p$ geometric structure where it is essentially coupled with the $L^p$ geometric structure (e.g., **heavily relying on the closed-form expression of $L^p$-norm**). Thus, it is nontrivial to incorporate structural priors beyond the $L^p$ geometry paradigm.

+ Our work generalizes Sobolev IPM beyond its coupled $L^p$ geometric structure by leveraging the Orlicz geometric structure, based on our novel theoretical finding results in Theorem 3.2. They are in fact **nontrivial and beyond results/techniques** in Le et al. (2025): **(i)** we leverage convex $N$-functions for Orlicz geometric structures (i.e., involving Orlicz-norm / Musielak-norm, which have **NO closed-form expressions** in general, unlike the $L^p$-norm in the $L^p$ geometry paradigm); **(ii)** we have developed several auxiliary results in Appendix A.1 for Orlicz geometric structure, **obviously departing from existing results** in Le et al. (2025), to establish our finding results in Theorem 3.2;  **(iii)** our weight function $\hat w$ (Equation (9)) **mainly stems from our finding results in Theorem A.3** (Appendix A.1), in fact it is **different** to the weight function derived in Le et al. (2025). Last but not least, we note that **our weight function plays a critical role in Theorem 3.2**. For any given general weight functions, one may **NOT** establish such results as in our Theorem 3.2.

+ Furthermore, in Propositions 4.4 and 4.5 (Section 4), we theoretically establish connections between our work and results in Le et al. (2025); and further discuss the relation in Section 5.

---

**[2] The role of our empirical results**

+ Performances of GSI-M with $N$-function $\Phi_1, \Phi_2$ **compare favorably** than those with $\Phi_0$, especially in **RECIPE and MPEG7** datasets.

+ Performances of GSI-M **compare favorably** than GST with the same $N$-function, especially in **TWITTER and RECIPE** datasets.

These empirical results **well-support our claimed contributions** in Section 1. We believe that there are no overclaims.

---

Additionally, we would like to highlight some other **important points** which the Reviewers may overlook or misunderstand:

---

**[3] In this work, we study the Sobolev IPM problem, and generalize it by leveraging Orlicz geometric structure**: we emphasize that in this work, we do **NOT** study Sobolev transport (ST), generalized Sobolev IPM (GST) problems; nor we try to propose a weighted variant of GST.

+ We agree that GSI-M may be regarded as a weighted variant of GST. However, there are **NO** established relations between our generalized Sobolev IPM problem and weighted variant of GST with any given general weight functions in the literature, to our knowledge.

+ **Our proposed GSI-M is based on theoretical finding results in Theorem 3.2., and our weight function plays the cornerstone for our proposed regularization approach (GSI-M)**. Therefore, weighted variant of GST with general weight function may **NOT** be related to our considered generalized Sobolev IPM problem.

+ Additionally, we theoretically establish the connection between our proposed GSI-M with ST/GST in Propositions 4.7 and 4.6 respectively (Section 4). We also give further discussion on their relations in Section 5.

---

**[4] Under this assumption (i.e., "the compared probability measures are supported on the same graph"), the computation of the optimal transport plan, required in Wasserstein or OW distances, can be avoided**

+ To our knowledge, we are not aware of such results yet. Additionally, after carefully reading all the mentioned references (Le et al., 2022, 2024, 2025), we believe that such results are **NOT** in these mentioned references. However, **when the graph is a tree**, then we can obtain such results from these mentioned references (for OW, there is an additional condition, $\Phi(t) = t$).

---

**[5] Distinguish our considered Sobolev IPM problem and its generalized problem (Generalized Sobolev IPM) from kernels/distances/discrepancies for graphs**

+ Our considered problems (Sobolev IPM / Generalized Sobolev IPM) are to compare **two input probability measures** supported on the **same graph**. They are different to kernels/distances/discrepancies for graphs problems where they compare **two input (different) graphs**.

---

We have clarified all of these critical remarks and important points, highlighted our further clarification in violet color in the revised version.

Thank you for your consideration.

Best regards,

The Authors

---

### Meta-Review · Area_Chair_MsJv · 2026-01-06

**Summary:**

This paper proposes a generalized Sobolev IPM for graph-supported measures by extending Sobolev IPM from L_pgeometry to Orlicz geometry, introducing a Musielak-regularized formulation that yields efficient, univariate computation, with theoretical connections to existing IPM and transport distances and empirical results on document classification and TDA.

The reviewers found the technical development careful and the presentation thorough, noting computational efficiency and solid proofs. However, they consistently raised concerns that the core contribution is incremental relative to prior work (notably Le et al.), with limited conceptual novelty, weak motivation for the proposed generalization, and unclear practical benefits beyond existing Sobolev/GST variants. Experimental gains were modest and insufficient to justify the added complexity.

After considering the rebuttal and discussion, these concerns remain. We therefore recommend rejection.

**Reviewer Concerns:**

Please see my summary.

**Reviewer Scores:**

It is difficult to say.  Overall, the authors provided some solid rebuttal, but it's a subjective judgement for the reviewer whether they would like to raise their score.

---

### Decision · Program_Chairs · 2026-01-26

Reject